# The robustness of differentiable causal discovery in misspecified scenarios

**Huiyang Yi**[1,*] **Yanyan He**[2,*] **Duxin Chen**[1,†] **Mingyu Kang**[2], **He Wang**[1], **Wenwu Yu**[1]
[1]School of Mathematics, Southeast University, Nanjing 210096, China
[2]School of Cyber Science and Engineering, Southeast University, Nanjing 210096, China
`yihuiyang@seu.edu.cn, heyy@seu.edu.cn, chendx@seu.edu.cn`
`kangmingyu.china@gmail.com, wanghe91@seu.edu.cn, wwyu@seu.edu.cn`

## Abstract

Causal discovery aims to learn causal relationships between variables from targeted data, making it a fundamental task in machine learning. However, causal discovery algorithms often rely on unverifiable causal assumptions, which are usually difficult to satisfy in real-world data, thereby limiting the broad application of causal discovery in practical scenarios. Inspired by these considerations, this work extensively benchmarks the empirical performance of various mainstream causal discovery algorithms, which assume i.i.d. data, under eight model assumption violations. Our experimental results show that differentiable causal discovery methods exhibit robustness under the metrics of Structural Hamming Distance and Structural Intervention Distance of the inferred graphs in commonly used challenging scenarios, except for scale variation. We also provide the theoretical explanations for the performance of differentiable causal discovery methods. Finally, our work aims to comprehensively benchmark the performance of recent differentiable causal discovery methods under model assumption violations, and provide the standard for reasonable evaluation of causal discovery, as well as to further promote its application in real-world scenarios.

## 1 Introduction

In the realm of modern science, numerous endeavors hinge upon the elucidation of underlying causal mechanisms. However, owing to practical constraints including costs, risks, and ethical implications, the execution of randomized experiments frequently proves unviable. Consequently, mining causal relationships from purely observational data, known as causal discovery, plays a crucial role in addressing causal questions such as intervention and counterfactual (Peters et al., 2017; Spirtes et al., 2001; Pearl, 2009; Pearl & Mackenzie, 2018).

Causal discovery encompasses a comprehensive suite of methodologies, primarily categorized into constraint-based, score-based, functional causal model-driven, and gradient-based approaches. These methods often rely on unverifiable causal assumptions as their foundation (Peters et al., 2017; Vowels et al., 2022). Constraint-based methods, notably PC (Spirtes & Glymour, 1991) and FCI (Spirtes et al., 1995), meticulously reconstruct causal graphs to the Markov equivalence class (MEC) through rigorous statistical independence tests, guided by the faithfulness assumption. Score-based techniques, such as GES (Chickering, 2002), employ a scoring function to quantify the congruence between an equivalence class and observed data, optimally searching the vast landscape of directed acyclic graphs (DAGs) to identify the MEC.

To transcend the limitation of solely identifying MECs from observational data, functional causal model-based methods, exemplified by LiNGAM (Shimizu et al., 2006), leverage precise assumptions regarding the functional class and noise distribution within structural equation models (SEMs), enabling the unambiguous identification of a unique DAG. Recently, Zheng et al. (2018) introduced gradient-based techniques (e.g. NOTEARS (Zheng et al., 2018)), which convert combinatorial acyclic

---

[*]Equal Contribution.
[†]Duxin Chen is the corresponding author.

constraints into smooth equality constraints and solve the optimization by transforming equality-constrained optimization into unconstrained optimization through the augmented Lagrangian method (ALM) (Nemirovsky, 1999). In some literature (Zhang et al., 2023; Liu et al., 2023), gradient-based methods are also referred to as differentiable causal discovery.

Apart from the various assumptions of the methods above, traditional approaches typically rely on causal sufficiency and no measurement error assumptions to simplify the problem (Peters et al., 2017; Zhang et al., 2018). Real-world data often fail to meet all of these assumptions, and these are also impossible to verify adequately (Peters et al., 2017). Although some studies have considered the complexity of real data and developed algorithms targeted at latent confounders (Spirtes et al., 1995; Xie et al., 2020; Salehkaleybar et al., 2020; Cai et al., 2019; Kong et al., 2023; Cai et al., 2023), measurement error (Zhang et al., 2018; Dai et al., 2022), heterogeneity (Huang et al., 2020; Cai et al., 2020; Ghassami et al., 2017; 2018), scale variation (Shimizu et al., 2011; Reisach et al., 2023; Deng et al., 2024), and missing data (Tu et al., 2019a; Gao et al., 2022), the true mechanisms remain unclear when the causal discovery algorithms applied to real data. These specifically designed algorithms also cannot be effectively employed for real data. Therefore, the robustness of causal discovery algorithms in scenarios where model hypotheses are violated is of great importance.

Previous research (Heinze-Deml et al., 2018) mainly evaluated various constraint-based and score-based algorithms under different scenarios, only focusing on linear SEM. The work (Mooij et al., 2016) benchmarked causal discovery for nonlinear additive noise models and information-geometric approaches, limiting to bivariate scenarios. The previous study (Singh et al., 2017) primarily assessed algorithms that use only observational data, a mix of observational and interventional data, and active learning, but their algorithm outputs were restricted to MEC. Also, those works (Glymour et al., 2019; Vowels et al., 2022) reviewed the advancements in traditional causal discovery (constraint-based, score-based, and functional causal model-based) and differentiable causal discovery, respectively, but lacked experimental support. The recent work (Ng et al., 2024) conducted an experimental assessment of the advancements in differentiable causal discovery, illuminating the shortcomings of current methods. However, their evaluation overlooked the ubiquitous violation of model assumptions that characterize real-world applications. Conversely, another work (Montagna et al., 2023) benchmarked the efficacy of traditional causal discovery algorithms, encompassing score-matching techniques, under scenarios where model assumptions were violated. Nevertheless, their analysis did not encompass the recent strides made in differentiable causal discovery, and the misspecified conditions they evaluated were constrained in scope. Given that the application of causal discovery methods to real data inevitably entails the violation of one or more unidentified assumptions, and that algorithms premised on specific assumptions may falter in practical use, the robustness of causal discovery in such misspecified contexts assumes paramount importance.

Our study undertakes an exhaustive empirical evaluation of both established and cutting-edge causal discovery methodologies, comprehensively examining their performance under diverse scenarios with assumption violations. The misspecified scenarios encompassed in our analysis represent the most extensive coverage in the current research landscape. We meticulously evaluate mainstream causal discovery approaches, spanning constraint-based, score-based, functional causal model-based, gradient-based methodologies, among others, ensuring a holistic view of the field. Notably, our work fills a crucial gap in the literature by being the first to assess the performance of gradient-based methods across a wide array of misspecified scenarios. Considering their practical implementation potential, it is important to evaluate their performance. Our contributions can be summarized as follows:

- We conduct extensive large-scale experimental evaluations of twelve prominent causal discovery algorithms across eight pivotal model assumption violation scenarios. Our rigorous research endeavor involves executing over 70,000 experiments on more than 2,400 synthetic datasets, ensuring a comprehensive assessment of the algorithm capabilities.

- We delve into challenging scenarios such as heterogeneity, scale variation, missing data, and mechanism violation, thereby enriching the benchmark data landscape for model assumption violations. This also aims to foster more comprehensive benchmark testing and foster a more rational evaluation framework for future causal discovery algorithms.

- Our analysis of the experimental outcomes offers theoretical insights into the performance of linear differentiable causal discovery methods under certain misspecified scenarios. Recognizing the robustness demonstrated by differentiable methods in these commonly used challenging settings,

we underscore the significant value of further in-depth research into differentiable causal discovery, as it holds the promise of advancing the field in novel and impactful directions.

## 2 BACKGROUND

In this section, we introduce the definition of causal discovery (Section 2.1), functional causal model-based (Section 2.2), score-based (Section 2.3) and gradient-based (Section 2.3) methods. For constraint-based methods, see Appendix A.2 and B.1.

### 2.1 TASK FORMULATION

A structural causal model $\mathcal{M}$ (Pearl, 2009) consists of the set of endogenous variables $X = (X_1, \ldots, X_d) \in \mathbb{R}^d$, exogenous variables $U = (U_1, \ldots, U_d) \in \mathbb{R}^d$, and functional mechanisms $\mathcal{F} = (f_1, \ldots, f_d)$. Each variable $X_i$ is defined by a structural equation:

$$X_i = f_i(X_{pa(X_i)}, U_i), \forall i = 1, \cdots, d, \tag{1}$$

where $X_i$ is the $i$-th node variable, $pa(X_i)$ denote the parents of $X_i$ , $f_i : \mathbb{R}^{|X_{pa(X_i)}|+1} \to \mathbb{R}$ is the causal structure function, and $U = (U_1, \ldots, U_d)$ are jointly independent noise variables with covariance matrix $\Omega = \text{cov}(U) = \text{diag}(\sigma_1^2, \ldots, \sigma_d^2)$.

The task of causal discovery is to infer a DAG $\mathcal{G}$ that describes the causal relationships among variables from $n$ independent and identically distributed (i.i.d.) observational data $\mathbf{X} \in \mathbb{R}^{n \times d}$, which are drawn from the joint probability distribution $P(X)$.

### 2.2 STRUCTURE IDENTIFIABILITY

To uniquely identify a DAG $\mathcal{G}$ from purely observational data $\mathbf{X}$ sampled from $P(X)$, we need to make assumptions about the SEM in (1). Considering a set of assumptions $\mathcal{A}$ on a structural causal model (SCM) $\mathcal{M}_{\mathcal{A}} = (P(X), \mathcal{G})$, the graph $\mathcal{G}$ is identifiable from $P(X)$ if there is no other SCM $\tilde{\mathcal{M}}_{\mathcal{A}} = (\tilde{P}(X), \tilde{\mathcal{G}})$ satisfying the same $\mathcal{A}$ such that $\tilde{\mathcal{G}} \neq \mathcal{G}$ and $\tilde{P}(X) = P(X)$. Existing identifiable causal models include: linear non-Gaussian acyclic models (Shimizu et al., 2006), linear Gaussian models with equal noise variances (Peters & Bühlmann, 2014), post-nonlinear models (Zhang & Hyvarinen, 2012) and nonlinear additive noise models (Hoyer et al., 2008; Peters et al., 2014).

### 2.3 DIFFERENTIABLE SCORE-BASED CAUSAL DISCOVERY

Traditional score-based causal discovery defines a combinatorial optimization problem:

$$\min_{\mathcal{G}} F(\mathcal{G}; \mathbf{X}) = \mathcal{L}_{\text{rec}}(\mathcal{G}; \mathbf{X}) + \lambda \mathcal{L}_{\text{sparse}}(\mathcal{G}) \quad \text{s.t.} \quad \mathcal{G} \in \text{DAG}, \tag{2}$$

where $F$ is a score function, $\mathcal{L}_{\text{rec}}(\mathcal{G}; \mathbf{X})$ represents the goodness-of-fit between the estimated DAG $\mathcal{G}$ and the true DAG, $\mathcal{L}_{\text{sparse}}(\mathcal{G})$ denotes the sparsity regularization term and $\lambda$ is a hyperparameter that controls the strength of regularization.

As the number of nodes rises, the total count of possible DAGs expands super-exponentially (Robinson, 1973). Consequently, most conventional score-based approaches utilize local heuristic search techniques, including greedy search (Chickering, 2002; Hauser & Bühlmann, 2012) and hill-climbing (Gámez et al., 2011; Tsamardinos et al., 2006).

In addition to search strategies, the design of score functions is also crucial. Commonly, score functions are classified into two categories: Bayesian-based scores and information-theoretic scores. Bayesian-based scores emphasize goodness-of-fit and enable the integration of prior knowledge, such as the Bayesian Dirichlet equivalent (Heckerman et al., 1995) and the K2 score (Kayaalp & Cooper, 2012). Information-theoretic scores, on the other hand, account for both model goodness-of-fit and complexity, including the Bayesian information criterion (Neath & Cavanaugh, 2012) and the Akaike information criterion (Akaike, 1998).

To overcome the challenges of combinatorial optimization, NOTEARS (Zheng et al., 2018) formulates the DAG structure learning task as:

$$\min_{\mathcal{G}} F(\mathcal{G}; \mathbf{X}) \quad \text{s.t.} \quad h(W(\mathcal{G})) = 0, \tag{3}$$

where $W(\mathcal{G}) \in \mathbb{R}^{d \times d}$ is a weighted adjacency matrix, $d$ is the number of nodes, $h(W(\mathcal{G})) = 0$ is a differentiable equality DAG constraint.

$h(W(\mathcal{G})) = 0$ if and only if $W(\mathcal{G})$ is a DAG. Commonly used DAG constraints include $h(W(\mathcal{G})) = \text{Tr}(e^{W \circ W}) - d$ (Zheng et al., 2018), $h(W(\mathcal{G})) = \text{Tr}[(I + \alpha W \circ W)^d] - d$ $(\alpha > 0)$ (Yu et al., 2019) and $h^s(W(\mathcal{G})) = -\log \det(sI - W \circ W) + d \log s$ $(s > 0)$ (Bello et al., 2022). Furthermore, we can transform the equality-constrained optimization (3) into unconstrained optimization (4) using the ALM (Nemirovsky, 1999):

$$\min_{\mathcal{G}} F(\mathcal{G}; \mathbf{X}) + \alpha_t h(W(\mathcal{G})) + \frac{\mu_t}{2} |h(W(\mathcal{G}))|^2, \tag{4}$$

where $\alpha_t$ and $u_t$ are the Lagrange multiplier and penalty parameter at the $t$-th iteration, respectively.

## 3 EXPERIMENTAL DESIGN

In this section, we introduce the generation of synthetic datasets with violated model assumptions, the tested causal discovery algorithms, the algorithm hyperparameters, and the evaluation metrics.

### 3.1 SYNTHETIC DATASETS

Many prevalent causal discovery approaches hinge upon unverifiable assumptions. Our study primarily scrutinizes the efficacy of these methodologies in circumstances where their underlying assumptions are breached. To achieve this, we commence by elucidating the baseline model under both linear and nonlinear frameworks, subsequently delving into scenarios where these fundamental assumptions fail to hold.

Table 1: Summary of the algorithm assumptions and their corresponding output graph types. The content within the cells indicates whether an algorithm supports (✓) or does not support (✗) the specific condition in the corresponding row. The table style is adjusted from Montagna et al. (2023).

| | PC | GES | DirectLiNGAM | CAM | Var-SortnRegress | $R^2$-SortnRegress | NOTEARS | GOLEM | NOTEARS-MLP | GraN-DAG | NoCurl | DAGMA |
|---|---|---|---|---|---|---|---|---|---|---|---|---|
| Gaussian noise | ✓ | ✓ | ✗ | ✓ | ✓ | ✓ | ✓ | ✓ | ✓ | ✓ | ✓ | ✓ |
| Non-Gaussian noise | ✓ | ✗ | ✓ | ✗ | ✓ | ✓ | ✓ | ✗ | ✓ | ✗ | ✓ | ✓ |
| Linear mechanisms | ✓ | ✓ | ✓ | ✗ | ✓ | ✓ | ✓ | ✓ | ✗ | ✗ | ✓ | ✓ |
| Nonlinear mechanisms | ✓ | ✓ | ✗ | ✓ | ✗ | ✗ | ✗ | ✗ | ✓ | ✓ | ✓ | ✓ |
| Unfaithful distribution | ✗ | ✗ | ✓ | ✓ | ✓ | ✓ | ✓ | ✓ | ✓ | ✓ | ✓ | ✓ |
| Confounding effects | ✗ | ✗ | ✗ | ✗ | ✗ | ✗ | ✗ | ✗ | ✗ | ✗ | ✗ | ✗ |
| Measurement errors | ✗ | ✗ | ✗ | ✗ | ✗ | ✗ | ✗ | ✗ | ✗ | ✗ | ✗ | ✗ |
| Autoregressive effects | ✗ | ✗ | ✗ | ✗ | ✗ | ✗ | ✗ | ✗ | ✗ | ✗ | ✗ | ✗ |
| Heterogeneous effects | ✗ | ✗ | ✗ | ✗ | ✗ | ✗ | ✗ | ✗ | ✗ | ✗ | ✗ | ✗ |
| Scale-variant effects | ✗ | ✗ | ✓ | ✗ | ✗ | ✓ | ✗ | ✗ | ✗ | ✗ | ✗ | ✗ |
| Missing mechanisms | ✗ | ✗ | ✗ | ✗ | ✗ | ✗ | ✗ | ✗ | ✗ | ✗ | ✗ | ✗ |
| Output | CPDAG | CPDAG | DAG | DAG | DAG | DAG | DAG | DAG | DAG | DAG | DAG | DAG |

**Linear vanilla model.** In linear SCM, following the settings of Zheng et al. (2018), coefficients are sampled from $U(-2, -0.5) \cup U(0.5, 2)$ with additive standard Gaussian noise. We refer to this model as the linear vanilla model, which satisfies both identifiability and the assumptions of most linear benchmark methods (see Table 1).

**Nonlinear vanilla model.** In nonlinear settings, following the settings of Zheng et al. (2020), the SEM in equation (1) is generated under the Gaussian process with radial basis function kernel of bandwidth one, where $f_i$ is additive noise models with $U_i$ as a standard Gaussian noise. We refer to this model as the nonlinear vanilla model, which satisfies both identifiability and the assumptions of nonlinear benchmark methods (see Table 1).

To eliminate the impact of Gaussian noise in the vanilla model on experimental results, we also consider cases where the vanilla model uses non-Gaussian noise (see Appendix G).

### 3.1.1 MODEL ASSUMPTION VIOLATION SCENARIOS

Four scenarios of model assumption violations are defined below. The other four cases, i.e., confounded, measurement error, unfaithful and autoregressive model, follow the same settings as Montagna et al. (2023). These eight misspecified scenarios can be applied to both the linear vanilla and nonlinear vanilla model to generate datasets.

**Heterogeneous model.** Existing causal discovery algorithms typically rely on the assumption of i.i.d. data. However, real data often exhibit distribution shifts (Huang et al., 2020). The heterogeneous multi-domain data considered in this paper primarily refers to scenarios where the underlying causal generation process remains unchanged, but the distribution of noise terms varies (Huang et al., 2020; Zhang et al., 2023; Wang et al., 2022). Specifically, we consider data from two domains $e_1$ and $e_2$. The proportion of data from $e_1$ is $P_1 \in \{0.1, 0.3, 0.5, 0.7, 0.9\}$ , and the proportion from $e_2$ is $1 - P_1$. The noise variance in $e_1$ is set the same as the vanilla model, while variance in $e_2$ is set to $\gamma \in \{0.01, 0.05, 0.1, 0.5\}$.

**Scale-variant model.** Reisach et al. (2021) observed a significant performance decline in linear gradient-based methods, such as NOTEARS (Zheng et al., 2018) and GOLEM (Ng et al., 2020), when applied to data with scale variation. However, there has been a notable absence of research investigating the performance of nonlinear methods in the context of scale variation. Therefore, we also consider scale variation as a misspecified scenario. The structural equations considered are:

$$\overline{X_i} = \frac{X_i - u_i}{\sqrt{\text{Var}(X_i)}}, \forall i = 1, \dots, d, \tag{5}$$

where $u_i$ and $\text{Var}(X_i)$ are the mean and variance of $X_i$, respectively. The input data are standardized, while the ground truth graph remains consistent with the causal graph that generates the original data.

**Missing model.** Missing data is a prevalent challenge in real-world datasets, necessitating that causal discovery algorithms effectively address this issue (Tu et al., 2019b). In our study, we adopt the Missing Completely At Random (MCAR) mechanism (Tu et al., 2019a), where the occurrence of missing values follows a Bernoulli distribution with a missingness probability of $\beta \in \{0.005, 0.01, 0.05, 0.1\}$. Given that the algorithms under consideration are incapable of directly processing datasets with missing values, we eliminate any records containing such gaps. To mitigate the influence of data quantity on the experimental outcomes, we ensure the volume of data remains consistent before and after the removal of incomplete records.

**Mechanism violation.** Most current causal discovery algorithms presuppose either linear or nonlinear mechanism, especially methods based on functional causal models (Shimizu et al., 2006; Peters & Bühlmann, 2014; Zhang & Hyvarinen, 2012; Hoyer et al., 2008; Peters et al., 2014). These methods necessitate specific assumptions about the SEM mechanism to guarantee identifiability. Given that the SEM in real-world data is typically unknown, the robustness of algorithms in the face of mechanism violation becomes critically important. In mechanism violation, the input data for algorithms designed for linear SEM will adhere to the nonlinear vanilla model, whereas the input data for algorithms tailored to nonlinear SEM will conform to the linear vanilla model.

### 3.1.2 DATA GENERATION

Following the data generation of Zheng et al. (2018; 2020) and Liu et al. (2023), different datasets are generated for both linear and nonlinear vanilla model. We simulate ER and SF graphs based on the number of nodes $d \in \{10, 20, 50\}$, average degree of nodes $k \in \{2, 4\}$. In addition, we consider scenarios with Gaussian Random Partitions (GRP) (Brandes et al., 2003) graph and an average node degree of 6. For each experimental configuration and scenario, 10 datasets of 2000 samples are generated. The mean and standard deviations of the evaluation metrics (Section 3.4) is reported to ensure a fair comparison.

## 3.2 METHODS

We select 12 mainstream causal discovery algorithms, including constraint-based, score-based, functional causal model-based, gradient-based and other representative methods. For a more detailed introduction to the various methods, see the Appendix B.

### 3.3 HYPERPARAMETER SETTINGS

PC (Spirtes & Glymour, 1991), CAM (Bühlmann et al., 2014), and GraN-DAG require adjustment of the significance level $\alpha$ in the statistical independence tests. NOTEARS, GOLEM, NOTEARS-MLP, and DAGMA need adjustment of the sparsity coefficient $\lambda_1$ for the $l_1$-norm regularization term. Typically, the ground truth of real data is unknown, making it difficult to effectively select hyperparameters for various algorithms. Thus, to ensure a fair comparison of various methods, we tune $\lambda_1$ in $\{0.005, 0.01, 0.05, 0.5, 2, 5\}$ and tune $\alpha$ in $\{0.001, 0.01, 0.05, 0.1\}$.

### 3.4 EVALUATION METRICS

We employ Structural Hamming Distance (SHD) and Structural Intervention Distance (SID) (Peters & Bühlmann, 2015) to evaluate performance. SHD counts the number of edge insertions, deletions, and reversals necessary to transform the estimated graph into the true graph. SID is used to assess the distinctions in intervention distribution between the estimated and the true graph. Intuitively, SHD focuses on differences in graph structure, while SID focuses on differences in causal ordering. Lower SHD and SID values indicate better estimation of the target causal graph by the algorithm. For cases where the output is a MEC, we follow the same approach as Zheng et al. (2018), evaluating them favorably by assuming the undirected edges in the MEC are in the correct direction.

## 4 CRITICAL EXPERIMENTAL RESULTS AND INSIGHTS

In this section, we first present the experimental results of the misspecified datasets generated according to Section 3.1.1, comparing them with the findings from the vanilla scenario to draw conclusions. Finally, we provide a more in-depth discussion on the performance of CAM (Section 4.1.1) and offer theoretical insights into the performance of differentiable causal discovery (Section 4.1.2). Due to space limitations, the main text focuses on the experimental results for ER-2 graphs of 10 nodes (Table 2.1, 2.2, 3.1, 3.2, 4.1, 4.2), whereas similar conclusions apply to different nodes, graph types and graph densities (see Appendix E and J). To visually and concisely present the results, the outcomes of the 10 nodes graph under linear, nonlinear, and MLP settings (Section 4.1.1) are summarized in Figure 1. We also consider the real-world Sachs (Sachs et al., 2005) dataset (see Appendix I), combined misspecified scenarios (see Appendix F), vanilla model with non-Gaussian noise (see Appendix G), semi-synthetic data (see Appendix L) and runtime of the benchmark methods (see Appendix D). For each scenario, we generate datasets with 10 different random seeds, each time drawing 2000 samples. We report the mean and standard deviations of the metrics over 10 trials. To guarantee a fair comparison of various methods, the hyperparameters for each method are determined as the optimal values relative to the specific dataset.

### 4.1 CURRENT METHODS' PERFORMANCE IN MISSPECIFIED SCENARIOS

Our experiments show that differentiable causal discovery algorithms almost always achieve optimal or competitive performance in commonly used misspecified scenarios other than scale variation. In this paper, robustness refers to the ability of the model to perform well in misspecified scenarios, consistent with the understanding of Montagna et al. (2023).

**Confounded, measurement error, autoregressive and heterogeneous model.** Table 2.1, 2.2, 3.1 and 3.2 indicate that under the commonly used confounded, measurement error, autoregressive and heterogeneous ($P_1 = 0.5$, $\gamma = 0.1$) scenarios, differentiable causal discovery achieves optimal or competitive performance compared to other methods. For the nonlinear Gaussian process mechanism, although CAM (Bühlmann et al., 2014) demonstrates better performance, the discussion in Section 4.1.1 reveals that CAM still has limitations compared to differentiable causal discovery.

**Missing model.** We generate missing data that are MCAR with the missingness probability $\alpha = 0.01$. Table 2.1, 2.2, 3.1 and 3.2 indicate that under MCAR, the result of various algorithms is close to that in the vanilla model. Our experiments show that the performance of differentiable causal discovery under missing data is consistent with traditional methods considered by Tu et al. (2019b;a), including PC and GES.

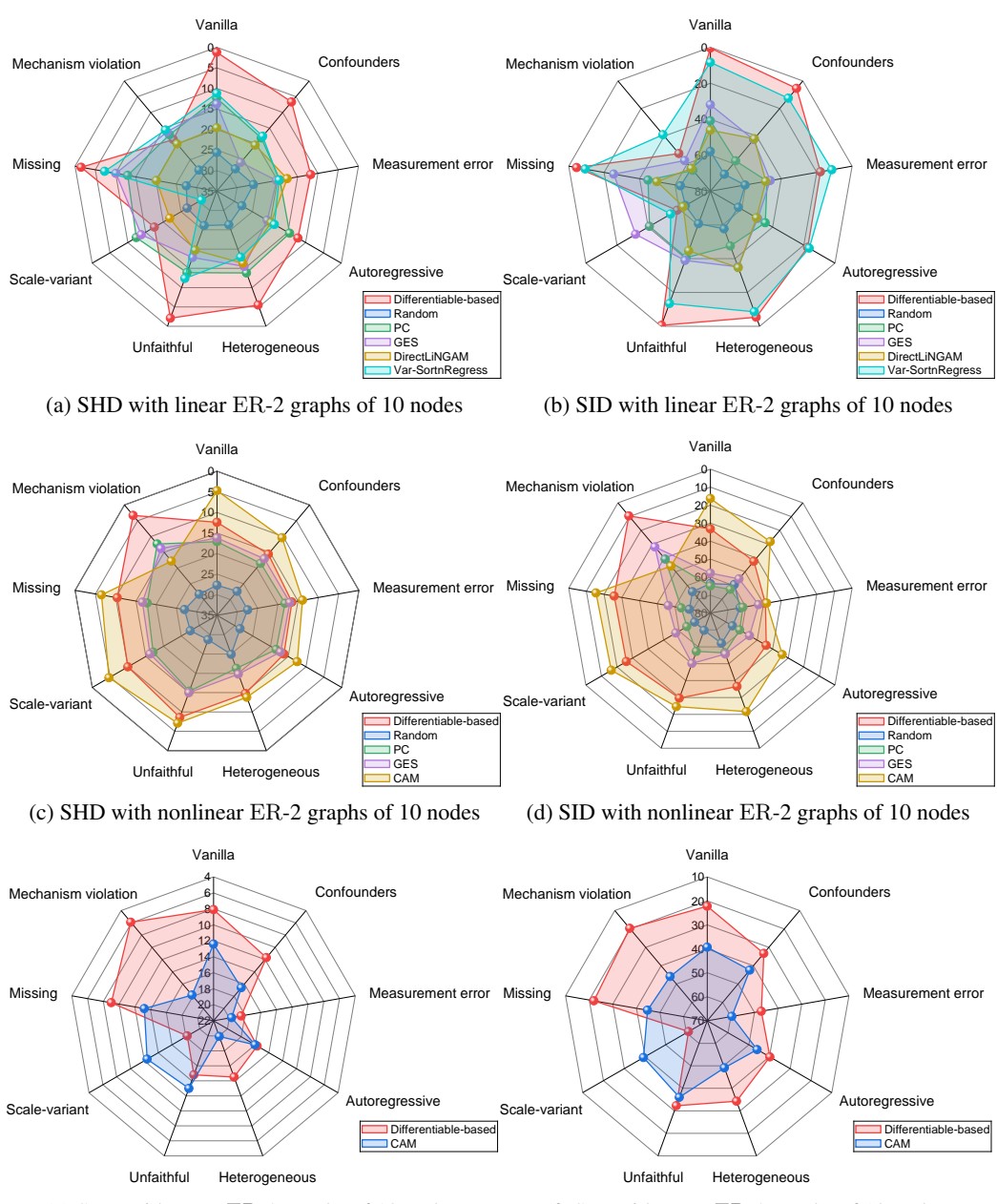

Figure 1: Experimental results under the linear, nonlinear, and mlp settings for both the vanilla scenario and the eight misspecified scenarios. SHD (the lower the better) and SID (the lower the better) are evaluated over 10 trials on the 10 nodes ER-2 graphs. For the differentiable causal discovery method, we present only the optimal results. As the nonlinear settings in Figure 1c and Figure 1d are more favorable to CAM, we conduct a more reasonable evaluation of CAM and differentiable causal discovery under the MLP setting (Section 4.1.1).

**Mechanism violation.** For mechanism violation in the linear setting of Table 2.1 and 2.2, despite PC and GES's ability to handle nonlinear mechanisms, we are surprised to observe that linear differentiable causal discovery algorithms achieve competitive performance. In the nonlinear setting of Table 3.1 and 3.2, we discover that nonlinear differentiable causal discovery algorithms, such as NOTEARS-MLP (Zheng et al., 2020), GraN-DAG (Lachapelle et al., 2019), and DAGMA (Bello et al., 2022), outperform other types of algorithms. From Table 3.1 and 3.2, we also see that CAM does not perform well under mechanism violation, although it excels in other scenarios. We speculate that this is because the Gaussian process mechanism used in the nonlinear vanilla model aligns well

with CAM's assumptions about SEM. For a more detailed discussion on CAM performance, see Section 4.1.1.

Table 2.1: Linear Setting, for ER-2 graphs of 10 nodes (Part I).

| 10 nodes | Vanilla model | | Latent confounders | | Measurement error | | Autoregressive | |
|---|---|---|---|---|---|---|---|---|
| | SHD↓ | SID↓ | SHD↓ | SID↓ | SHD↓ | SID↓ | SHD↓ | SID↓ |
| Random | 25.6±3.1 | 57.9±9.5 | 27.9±2.3 | 67.8±7.8 | 25.9±3.5 | 60.4±11.3 | 27.9±3.2 | 62.0±8.1 |
| PC | 12.4±3.1 | 40.9±13.4 | 18.1±4.7 | 58.1±15.6 | 19.4±4.1 | 48.0±13.1 | 14.5±2.0 | 44.8±9.5 |
| GES | 13.8±7.8 | 32.0±13.6 | 25.9±7.7 | 42.6±14.0 | 20.2±4.8 | 46.2±16.7 | 20.8±5.5 | 49.7±11.5 |
| DirectLiNGAM | 19.6±3.3 | 46.1±10.6 | 20.4±5.0 | 42.0±6.0 | 17.6±2.4 | 48.8±12.4 | 19.7±4.2 | 50.4±8.4 |
| Var-SortnRegress | 11.2±3.5 | 8.4±8.5 | 17.6±5.8 | 12.6±9.9 | 19.6±2.8 | **11.4±8.7** | 18.8±2.4 | **16.5±10.6** |
| $R^2$-SortnRegress | 20.2±4.8 | 32.4±14.0 | 25.7±4.1 | 37.6±13.0 | 25.6±6.0 | 39.2±16.0 | 25.6±4.9 | 38.8±19.0 |
| NOTEARS | 1.5±1.6 | 1.8±4.2 | 8.5±3.9 | 9.5±8.1 | 12.5±2.0 | 19.6±8.6 | **12.2±3.6** | 27.5±14.2 |
| GOLEM | 1.4±1.4 | **0.4±1.2** | **6.7±2.8** | 14.2±9.8 | 17.8±2.5 | 43.1±13.3 | 16.6±4.0 | 34.9±16.9 |
| NoCurl | 2.0±1.8 | 5.1±5.8 | 9.1±4.2 | **5.4±3.9** | 11.8±1.8 | 17.9±8.4 | 14.8±2.5 | **17.5±10.8** |
| DAGMA | **1.2±1.2** | 3.3±5.3 | 8.4±3.9 | 8.8±7.7 | 12.6±2.5 | 18.5±8.6 | **12.2±3.6** | 28.4±15.3 |

Table 2.2: Linear Setting, for ER-2 graphs of 10 nodes (Part II).

| 10 nodes | Heterogeneous | | Unfaithful | | Scale-variant | | Missing | | Mechanism violation | |
|---|---|---|---|---|---|---|---|---|---|---|
| | SHD↓ | SID↓ | SHD↓ | SID↓ | SHD↓ | SID↓ | SHD↓ | SID↓ | SHD↓ | SID↓ |
| Random | 26.3±3.5 | 57.7±7.6 | 26.1±3.7 | 60.7±12.9 | 26.7±3.0 | 64.0±8.7 | 27.5±3.4 | 63.1±6.0 | 28.3±3.0 | 63.6±7.8 |
| PC | 13.8±2.6 | 47.5±10.3 | 13.9±3.2 | 40.6±10.4 | **12.4±3.1** | 40.9±13.4 | 13.0±4.7 | 44.8±16.0 | 17.1±2.5 | 64.9±10.0 |
| GES | 15.5±6.1 | 35.2±12.2 | 17.8±6.4 | 39.0±15.7 | 13.8±7.8 | **32.0±13.6** | 10.1±5.2 | 25.4±12.6 | 16.2±2.2 | 57.8±10.7 |
| DirectLiNGAM | 16.3±3.9 | 34.7±9.9 | 19.7±4.3 | 44.7±13.9 | 21.8±4.3 | 62.6±9.3 | 20.1±4.3 | 49.8±11.1 | 20.0±0.0 | 63.5±7.7 |
| Var-SortnRegress | 17.9±3.3 | 8.6±9.3 | 12.4±3.1 | 13.5±8.0 | 30.7±5.1 | 54.5±10.3 | 7.3±3.5 | 9.3±8.4 | **15.6±3.3** | **39.0±6.7** |
| $R^2$-SortnRegress | 26.0±5.4 | 37.0±14.4 | 29.8±4.8 | 51.0±11.3 | 20.2±4.8 | 32.4±14.0 | 20.5±6.7 | 32.0±8.8 | 20.3±3.7 | 66.1±9.7 |
| NOTEARS | **5.5±2.7** | **5.4±5.1** | 2.7±3.1 | 3.1±5.1 | 18.0±1.2 | 60.5±7.3 | 2.3±1.7 | 6.4±8.6 | 19.0±0.9 | 58.3±8.0 |
| GOLEM | 6.5±4.5 | 9.8±8.1 | **2.1±2.2** | **0.6±1.8** | 17.5±1.2 | 64.4±6.8 | 1.7±1.7 | 6.2±10.8 | 18.6±1.6 | 52.7±4.3 |
| NoCurl | 6.6±2.9 | 5.5±5.7 | 2.2±2.3 | 2.0±4.4 | 27.2±5.1 | 69.9±7.9 | 3.1±3.2 | 4.7±5.8 | 19.1±1.0 | 58.9±9.5 |
| DAGMA | **5.5±2.3** | 12.0±8.2 | **2.1±2.2** | **0.6±1.8** | 17.9±1.4 | **58.7±6.8** | **1.5±1.4** | **4.5±7.1** | 19.0±0.9 | 58.3±8.0 |

**Scale-variant model.** In Table 2.1 and 2.2, we observe that the results of linear differentiable causal discovery algorithms, such as NOTEARS (Zheng et al., 2018), GOLEM (Ng et al., 2020), NoCurl (Yu et al., 2021), and DAGMA, significantly decline under scale-variant data, performing worse than PC and GES, which is consistent with the observations of Reisach et al. (2021). For nonlinear differentiable methods, performance under scale-variant data has not been explored in previous research. Table 3.1, 3.2, 4.1 and 4.2 indicate that nonlinear differentiable methods also show performance degradation under scale variation scenarios, and their results almost always lower than CAM. However, unlike the linear scenarios, the result of nonlinear differentiable algorithms is almost always superior to PC and GES.

**Unfaithful model.** In the linear setting of Table 2.1 and 2.2, we see a significant performance drop for Var-SortnRegress (Reisach et al., 2021) and $R^2$-SortnRegress (Reisach et al., 2023) under unfaithful distributions. The explanation for this is that for each triplet $X_i \rightarrow X_j \rightarrow X_k \leftarrow X_i$ in the graph, after the causal direct effect of $X_i \rightarrow X_k$ cancels out, the variance of node $X_k$ changes significantly. This reduces the Var-Sortability, further leading to a performance decline in the two SortnRegress algorithms and linear differentiable methods. In the nonlinear setting of Table 3.1 and 3.2, the SHD of various algorithms generally decline to some extent under unfaithful path cancellations. This is consistent with the experimental results of Montagna et al. (2023), which indicate that the cancellation of causal effects in unfaithful nonlinear scenarios makes structural inference of sparse graphs easier.

### 4.1.1 DISCUSSION ON CAM PERFORMANCE

**Motivations.** The nonlinear vanilla model adopts a Gaussian process that is consistent with the assumptions of the CAM. To provide a fair benchmark for CAM, we consider the nonlinear vanilla model following different functional mechanism. We compare CAM with the representative differentiable causal discovery method: NOTEARS-MLP.

**Simulations.** We simulate ER-2 graphs based on the number of nodes $d \in \{10, 20, 50\}$. Following the data generation mechanisms of Zheng et al. (2020), we consider $f_i$ in nonlinear vanilla model (Section 3.1) is modified to be parameterized by a neural network with one hidden layer of size 100.

**Results.** From Table 4.1 and 4.2, we observe that NOTEARS-MLP achieves better performance under almost all model assumption violations. Considering that the functional mechanisms of data in real-world scenarios are usually unknown, we believe that differentiable causal discovery has

a significant advantage over CAM in all types of assumption violation scenarios except for scale variation.

Table 3.1: Nonlinear Setting, for ER-2 graphs of 10 nodes (Part I).

| 10 nodes | Vanilla model | | Latent confounders | | Measurement error | | Autoregressive | |
|---|---|---|---|---|---|---|---|---|
| | SHD↓ | SID↓ | SHD↓ | SID↓ | SHD↓ | SID↓ | SHD↓ | SID↓ |
| Random | 27.7±3.2 | 63.6±11.2 | 27.4±2.5 | 59.3±9.4 | 27.4±3.6 | 63.2±7.7 | 28.5±2.3 | 65.8±8.6 |
| PC | 17.1±2.5 | 64.9±10.0 | 18.5±1.9 | 62.8±9.1 | 18.2±1.1 | 61.6±12.2 | 18.4±1.3 | 61.2±9.7 |
| GES | 16.2±2.2 | 57.8±10.7 | 17.1±2.1 | 55.2±15.5 | 17.0±1.0 | 52.8±9.1 | 17.2±2.0 | 54.9±7.8 |
| CAM | **4.7±1.9** | **16.3±9.5** | **10.4±2.8** | **28.2±5.3** | **13.9±1.9** | 48.2±7.4 | **12.5±3.0** | **33.9±16.3** |
| NOTEARS-MLP | 12.4±2.2 | 36.3±7.1 | 17.0±1.7 | 49.2±8.6 | 16.5±0.8 | 48.9±4.8 | 17.0±3.7 | 47.7±11.0 |
| GraN-DAG | 12.7±2.4 | **33.2±10.6** | 15.6±2.1 | 42.4±8.8 | 20.0±1.1 | 63.8±11.3 | **16.2±2.3** | **44.2±10.0** |
| DAGMA | 13.5±2.0 | 40.7±8.1 | 18.6±2.0 | 62.0±13.3 | 17.3±1.3 | 54.9±8.3 | 19.0±2.0 | 56.6±10.5 |

Table 3.2: Nonlinear Setting, for ER-2 graphs of 10 nodes (Part II).

| 10 nodes | Heterogeneous | | Unfaithful | | Scale-variant | | Missing | | Mechanism violation | |
|---|---|---|---|---|---|---|---|---|---|---|
| | SHD↓ | SID↓ | SHD↓ | SID↓ | SHD↓ | SID↓ | SHD↓ | SID↓ | SHD↓ | SID↓ |
| Random | 24.9±3.2 | 62.1±10.1 | 28.7±2.1 | 69.8±7.9 | 27.5±2.5 | 69.9±5.9 | 27.0±4.2 | 68.1±8.0 | 28.3±3.4 | 64.5±10.7 |
| PC | 21.3±3.2 | 56.5±10.4 | 15.4±1.4 | 57.3±9.5 | 17.1±2.5 | 64.9±10.0 | 17.8±3.1 | 63.5±10.5 | 12.4±3.1 | 40.9±13.4 |
| GES | 19.8±2.5 | 55.7±8.4 | 15.0±4.5 | 50.2±13.1 | 16.2±2.2 | 57.8±10.7 | 16.6±2.6 | 56.2±9.9 | 13.8±7.8 | 32.0±13.6 |
| CAM | **13.8±2.9** | **21.7±12.8** | **7.1±3.2** | **24.7±13.0** | **4.7±1.9** | **16.3±9.5** | **6.5±2.1** | **15.4±7.3** | 17.8±4.4 | 45.9±17.0 |
| NOTEARS-MLP | 16.4±3.7 | 42.6±10.6 | 11.4±2.1 | 43.8±9.2 | 16.1±2.5 | 48.3±9.7 | 12.3±2.1 | 33.6±7.0 | 5.9±2.5 | 19.7±8.7 |
| GraN-DAG | **14.8±2.1** | 40.5±10.8 | 11.2±3.1 | 37.7±10.0 | 10.0±3.7 | 26.1±12.0 | 10.4±3.4 | 25.7±8.1 | 16.3±2.1 | 54.7±5.1 |
| DAGMA | 16.4±4.3 | **36.6±15.7** | 8.6±3.1 | 29.6±14.8 | 15.7±2.7 | 53.3±12.9 | 13.7±2.7 | 41.4±7.8 | **3.3±3.1** | **9.5±11.6** |

### 4.1.2 THEORY ON DIFFERENTIABLE CAUSAL DISCOVERY IN MISSPECIFIED SCENARIOS

We analyze the performance of linear differentiable causal discovery under measurement error, unfaithful and missing scenarios by introducing the theories (Theorem 7 and Theorem 9) from Loh & Bühlmann (2014). Theorem 7 in Loh & Bühlmann (2014) states that for a linear model with equal noise variance, minimizing the least squares score will return the true DAG in the large sample limit. For a linear model with non-equal noise variances, we define the noise ratio

$$r = \frac{\max(\sigma_1^2, \ldots, \sigma_d^2)}{\min(\sigma_1^2, \ldots, \sigma_d^2)}. \tag{6}$$

Theorem 9 in Loh & Bühlmann (2014) states that if $r < 1 + \frac{\xi}{d}$, where $\xi > 0$ is the gap between the score of the true DAG and the next best DAG, minimizing the least squares score will return the true DAG in the large sample limit.

**Measurement error.** In measurement error model considered by Montagna et al. (2023), the observed variables are:

$$\tilde{X}_i = X_i + \epsilon_i, \forall i = 1, \ldots, d, \tag{7}$$

where $X_i = f_i(X_{pa(X_i)}) + U_i$, $f_i$ is a linear mechanism, $U_i \sim N(0, 1)$, $\epsilon_i \sim N(0, \delta * \text{Var}(X_i))$ with $\delta \in \{0.2, 0.4, 0.6, 0.8\}$. In the vanilla model, $r = 1$. However, in the measurement error model, the noise ratio becomes

$$\tilde{r} = \frac{\max(1 + \delta * \text{Var}(X_i), \ldots, 1 + \delta * \text{Var}(X_d))}{\min(1 + \delta * \text{Var}(X_i), \ldots, 1 + \delta * \text{Var}(X_d))}. \tag{8}$$

Due to the increasing trend of marginal variances of nodes along the causal direction (Reisach et al., 2021), we infer that $\tilde{r} > r = 1$. In this scenario, there is no guarantee that $\tilde{r} < 1 + \frac{\tilde{\xi}}{d}$ and linear differentiable causal discovery based on least squares cannot guarantee obtaining the true DAG, which can explain their performance decline in Table 2.1 and 2.2.

**Unfaithful model.** In unfaithful model considered by Montagna et al. (2023), for each triplet $X_i \to X_j \to X_k \leftarrow X_i$ in the graph, the causal mechanisms are adjusted such that the direct effect of $X_i$ on $X_k$ cancels out. To illustrate the change in noise ratio after path cancellation, we consider a DAG $\mathcal{G}$ with variable set $V(\mathcal{G}) = \{X_1, X_2, X_3\}$ and edge set $E(\mathcal{G}) = \{X_1 \to X_2, X_2 \to X_3, X_1 \to X_3\}$. The structural equations is defined as:

$$\begin{aligned} X_1 &= U_1, \\ X_2 &= f_1(X_1) + U_2, \\ X_3 &= f_1(X_1) - X_2 + U_3, \end{aligned} \tag{9}$$

where $f_1$ is a linear mechanism. After the direct causal effect of $X_1 \to X_3$ cancels out, the noise term of $X_3$ is $U_3 - U_2$ with the distribution of $N(0, 2)$. Similarly, in the unfaithful datasets with nodes $d \in \{10, 20, 50\}$ considered by our experiments, for each triplet $X_i \to X_j \to X_k \leftarrow X_i$ in the graph, once unfaithful path cancellation occurs, the noise term of $X_k$ is $U_k - U_j$ with the distribution of $N(0, 2)$. In this case, the noise ratio becomes $r' = 2 > r = 1$(vanilla model). Due to the increasing of the noise ratio, there is no guarantee that $r' < 1 + \frac{\xi'}{d}$ and linear differentiable causal discovery based on least squares cannot guarantee obtaining the true DAG, which can also explain their performance decline in Table 2.1 and 2.2.

**Missing model.** Under the MCAR case, we deleted the rows with missing values and regenerated the data under the i.i.d. assumption to ensure the unchanged sample size. The noise ratio $r = 1$ remains constant before and after data imputation. This explains the superior performance of differentiable causal discovery methods observed in Table 2.1 and 2.2.

Table 4.1: MLP Setting, for ER-2 graphs of 10 nodes (Part I).

| 10 nodes | Vanilla model | | Latent confounders | | Measurement error | | Autoregressive | |
|---|---|---|---|---|---|---|---|---|
| | SHD↓ | SID↓ | SHD↓ | SID↓ | SHD↓ | SID↓ | SHD↓ | SID↓ |
| CAM | 12.4±3.6 | 39.3±16.5 | 16.6±4.2 | 42.4±17.7 | 19.7±4.7 | 59.6±12.0 | 16.0±3.4 | 46.0±16.6 |
| NOTEARS-MLP | **8.1±2.7** | **22.2±10.6** | **11.7±5.5** | **33.3±17.0** | **18.5±3.7** | **47.1±11.9** | **15.7±4.3** | **39.8±9.6** |

Table 4.2: MLP Setting, for ER-2 graphs of 10 nodes (Part II).

| 10 nodes | Heterogeneous | | Unfaithful | | Scale-variant | | Missing | | Mechanism violation | |
|---|---|---|---|---|---|---|---|---|---|---|
| | SHD↓ | SID↓ | SHD↓ | SID↓ | SHD↓ | SID↓ | SHD↓ | SID↓ | SHD↓ | SID↓ |
| CAM | 19.9±3.6 | 49.1±7.2 | **13.0±3.2** | 36.0±14.0 | **12.4±3.6** | **39.3±16.5** | 13.2±3.6 | 44.7±16.7 | 17.8±4.4 | 45.9±17.0 |
| NOTEARS-MLP | **14.5±2.2** | **34.3±10.4** | 14.8±3.9 | **32.3±12.8** | 18.2±2.6 | 61.0±11.0 | **9.0±3.3** | **22.0±10.0** | **5.9±2.5** | **19.7±8.7** |

## 4.2 SUMMARY AND IMPLICATIONS FOR PRACTICE

In Appendix H, we summarize the results of the most competitive methods under misspecified scenarios. Differentiable causal discovery methods demonstrate optimal or competitive performance in commonly used scenarios other than scale variation. Notably, the recent work by Deng et al. (2024) shows that for linear differentiable methods, scale invariance can be achieved by appropriately choosing the loss function. This further reinforces our conclusion regarding the robustness of differentiable methods. In our benchmarks, the results in Table 11, Table 21 and Table 25 indicate that the performance of nonlinear differentiable methods under scale variation remains challenging and warrants further investigation. In practice, the misspecified scenarios are inevitably encountered, making the robustness of algorithms critically important. Based on the summarized results on eight misspecified synthetic datasets (see Appendix H), runtime results of benchmark methods (see Appendix D), real-world (see Appendix I) and semi-synthetic data (see Appendix L) results, we observe that differentiable causal discovery methods have the potential to achieve optimal or competitive performance on real-world data with an almost negligible time cost. The fast and robust characteristics of differentiable methods enable them to better address the challenges of applying causal discovery algorithms to real-world data, demonstrating their practical implementation potential.

## 5 CONCLUSION

This work assesses the efficacy of twelve preeminent causal discovery methods across eight scenarios involving violations of model assumptions. These methods encompass approaches grounded in independence constraints, scoring criteria, functional causal models, and differentiable causal discovery. Our experimental results show that differentiable causal discovery methods exhibit remarkable resilience in commonly used scenarios of model assumption violations, except for scale variation. It is not our intention to assert that differentiable causal discovery will achieve optimal performance across all circumstances, rather, we aim to underscore its substantial potential within the benchmarks we have evaluated, thereby emphasizing the necessity for further exploration in this direction. In future work, causal discovery methods for more semi-synthetic data and real-world scenarios will be explored. Finally, our study confines itself to non-temporal causal discovery algorithms. Equally crucial is the conduct of benchmark assessments for causal discovery in time series and event sequences under model assumption violations.

## 6 ACKNOWLEDGMENTS

This work was supported in part by the National Key R&D Program of China under Grant 2022ZD0120004, in part by the Zhishan Youth Scholar Program, in part by the National Natural Science Foundation of China under Grant 62233004, Grant 62273090, and Grant 62073076, and in part by the Jiangsu Provincial Scientific Research Center of Applied Mathematics under Grant BK20233002.

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

CONTENTS

# A    CAUSAL ASSUMPTIONS

In this section, we introduce the assumptions frequently used in the causal discovery literature.

## A.1    CAUSAL MARKOV PROPERTY

The joint probability distribution $P(X)$ satisfies the global Markov property (Peters et al., 2017) with respect to the DAG $\mathcal{G}$ if

$$X_A \perp\!\!\!\perp_\mathcal{G} X_B \mid X_C \Rightarrow X_A \perp\!\!\!\perp_{P(X)} X_B \mid X_C, \tag{10}$$

where $X_A$, $X_B$ and $X_C$ are the disjoint subsets of $X = (X_1, \ldots, X_d) \in \mathbb{R}^d$, $\perp\!\!\!\perp_\mathcal{G}$ denotes *d-separation* in the causal graph $\mathcal{G}$, and $\perp\!\!\!\perp_{P(X)}$ represents independence in the joint probability distribution $P(X)$.

In the causal graph $\mathcal{G}$ of SCM, each variable is independent of its non-descendant nodes when its parents are known, which is referred to as the local Markov property (Peters et al., 2017). Causal Markov property also implies that the joint probability distribution $P(X)$ can be factorized in the following form:

$$P(X) = \prod_i^d P\left(X_i \mid pa(X_i)\right). \tag{11}$$

## A.2    FAITHFULNESS

The joint probability distribution $P(X)$ is faithful (Peters et al., 2017) to the DAG $\mathcal{G}$ if

$$X_A \perp\!\!\!\perp_{P(X)} X_B \mid X_C \Rightarrow X_A \perp\!\!\!\perp_\mathcal{G} X_B \mid X_C. \tag{12}$$

Faithfulness assumption implies that the conditional independence in the $P(X)$ can be used to infer the graph structure. Constraint-based and traditional score-based causal discovery are typically founded on the faithfulness assumption. In a causal graph, the cancellation of effects along multiple causal paths can lead to a violation of faithfulness assumption.

## A.3    CAUSAL SUFFICIENCY

Causal sufficiency assumption is also referred to as no latent confounder. A set of variables $X$ is said to satisfy the causal sufficiency if there is no unobserved common cause variable $C$ that influences more than one variable in $X$ (Spirtes, 2010). This assumption is also frequently considered in causal discovery literature. However, since we cannot always observe all variables in the real world, causal sufficiency assumption is inevitably violated.

## A.4    INDEPENDENT AND IDENTICALLY DISTRIBUTED

Non-temporal causal discovery algorithms typically also require the i.i.d. assumption. In the main text, heterogeneous multi-domain data and autoregressive scenarios are two special cases where the i.i.d. assumption is violated. Heterogeneous multi-domain data is closely related to the non-stationary time series data considered in the literature on temporal causal discovery (Huang et al., 2020). Below, we introduce the connection between them. We consider the distribution of $X_i$ changing with domain or time index, where the mechanism for the $t$-th data point is as follows:

$$X_{i,t} = f_{i,t}\left(pa(X_{i,t}), \epsilon_{i,t}\right), \tag{13}$$

where $\epsilon_{i,t}$ is the noise term of $X_{i,t}$. In heterogeneous multi-domain data, $t$ represents the domain index, whereas in non-stationary time series data, $t$ denotes the time index.

## A.5    EQUAL NOISE VARIANCES

Ng et al. (2024) observe that the performance of linear differentiable causal discovery methods significantly declines in data with non-equal noise variances. They hypothesize that this may be due to the optimization problem becoming severely non-convex under non-equal noise variances, leading to local optimal solutions. Although differentiable methods do not explicitly assume equal noise variance, the performance decline suggests treating equal noise variance as a causal assumption.

# B  BENCHMARK METHODS

## B.1  PC

PC (Spirtes & Glymour, 1991) algorithm is a representative constraint-based causal discovery method. In the first step, under the faithfulness assumption, the global causal skeleton is determined based on conditional independence tests. In the second step, some edge directions in the skeleton are determined by identifying collider structures. Finally, the remaining edge directions are determined using orientation rules, resulting in the MEC. We use the implementation of the PC algorithm in causal-learn (Zheng et al., 2024) python package, available at `https://github.com/py-why/causal-learn`.

## B.2  GES

GES (Chickering, 2002) algorithm is a classical score-based causal discovery method. GES mainly includes two stages. In the first stage, starting from an empty graph, edges are added through greedy equivalence search, and the structures in the equivalence class of the new graph are scored. The graph with the highest score is selected, and the edge-adding process is repeated until the score reaches a local maximum. In the second stage, starting from the graph obtained in the first stage, edges are removed through greedy equivalence search, and the structures in the equivalence class of the new graph are scored. The graph with the highest score is selected, and the edge-removal process is repeated until the score reaches a local maximum. We use the implementation of the GES algorithm in causal-learn (Zheng et al., 2024) python package, available at `https://github.com/py-why/causal-learn`.

## B.3  DIRECTLINGAM

DirectLiNGAM (Shimizu et al., 2011) is a classical linear method based on the functional causal model. To address the issues of slow convergence and large errors in the ICA-based LiNGAM (Shimizu et al., 2006) algorithm, Shimizu et al. (2011) proposed DirectLiNGAM based on the principle of residual independence. Although the solving speed became slower, the accuracy and convergence improved. We use the implementation of DirectLiNGAM algorithm in gCastle (Zhang et al., 2021) python package, available at `https://github.com/huawei-noah/trustworthyAI/tree/master/gcastle`.

## B.4  CAM

CAM (Bühlmann et al., 2014) is a method used for high-dimensional additive structural equation models. CAM separates the search for the order of variables from the selection of edges. It performs the variable order search through nonregularized maximum likelihood estimation and uses sparse regression techniques for edge selection. We use the implementation of CAM algorithm in Causal Discovery Toolbox (Kalainathan & Goudet, 2019) python package, available at `https://github.com/FenTechSolutions/CausalDiscoveryToolbox`.

## B.5  SORTNREGRESS

SortnRegress algorithm includes Var-SortnRegress (Reisach et al., 2021) and $R^2$-SortnRegress (Reisach et al., 2023).

Reisach et al. (2021) emphasized that in the bivariate linear case, causal direction inferred by minimizing mean squared error (MSE) loss is from the variable with smaller variance to the variable with larger variance. They further hypothesized that, in the multivariate case, there is a consistency between the underlying causal direction of the data and the increasing order of the marginal variances of the variables. They provided a general definition of sortability:

$$\mathbf{v}_\tau(X, \mathcal{G}) = \frac{\sum_{i=1}^{d} \sum_{(s \to t) \in A(\mathcal{G})^i} \mathrm{incr}(\tau(X, s), \tau(X, t))}{\sum_{i=1}^{d} \sum_{(s \to t) \in A(\mathcal{G})^i} 1} \text{ where } \mathrm{incr}(a, b) = \begin{cases} 1 & a < b \\ 1/2 & a = b \\ 0 & a > b \end{cases}, \quad (14)$$

$\tau$ represents a function with $\tau(X) \in [0, 1]$, $A(\mathcal{G})$ is the adjacency matrix of $\mathcal{G}$, $A(\mathcal{G})^i$ is the $i$-th matrix power, $(s \rightarrow t) \in A(\mathcal{G})^i$ if and only if at least one directed path of length $i$ from $X_s$ to $X_t$.

In Var-Sortability, $\tau(X, s)$ denotes the variance of $X_s$. Var-Sortability measures the consistency between the causal structure order and the increasing order of marginal variances of nodes. Intuitively, the greater Var-Sortability of the data, the better performance of methods based on MSE loss. Subsequently, they proposed the Var-SortnRegress algorithm to discover causality using only variance. In the first step, the nodes are ranked according to the increasing order of marginal variances. In the second step, linear and Lasso regression are used for estimation. We use the implementation of Var-SortnRegress algorithm in CausalDisco python package provided by the authors, available at `https://github.com/CausalDisco/CausalDisco`.

Reisach et al. (2021) demonstrated that the performance of linear differentiable causal discovery algorithms is greatly affected by Var-Sortability. Building on previous work, Reisach et al. (2023) pointed out that the coefficient of determination $R^2$ remains unchanged after scaling the data, and proposed the $R^2$-SortnRegress algorithm, which achieves better performance on scale-variant data. The definition of $R^2$:

$$R^2 = 1 - \frac{\text{Var}\left(X_t - \mathbb{E}\left[X_t \mid X_{\{1,\ldots,d\}\setminus\{t\}}\right]\right)}{\text{Var}\left(X_t\right)}. \tag{15}$$

In $R^2$-Sortability, $\tau(X, s)$ denotes the coefficient of determination $R^2$ of $X_s$. $R^2$-Sortability measures the consistency between the causal structure order and the increasing order of $R^2$. If $\mathbf{v}_{R^2}(X, \mathcal{G}) = 1$, the causal order can be fully identified by the increasing order of $R^2$. If $\mathbf{v}_{R^2}(X, \mathcal{G}) = 0$, the causal order can be fully identified by the decreasing order of $R^2$. The only difference between $R^2$-SortnRegress and Var-SortnRegress lies in the definition of $\tau$. We use the implementation of $R^2$-SortnRegress algorithm in CausalDisco python package provided by the authors, available at `https://github.com/CausalDisco/CausalDisco`.

## B.6 NOTEARS

Based on (3), the NOTEARS score function is:

$$\min_{\mathcal{G}} F(\mathcal{G}; \mathbf{X}) = \frac{1}{2n}\|\mathbf{X} - \mathbf{X}W(\mathcal{G})\|_F^2 + \lambda\|W(\mathcal{G})\|_1 \quad \text{s.t.} \quad h(W(\mathcal{G})) = 0, \tag{16}$$

where $\|\cdot\|_F$ is the Frobenius norm, $\|\cdot\|_1$ is the sum of absolute values of all elements in the matrix.

The unconstrained objective function obtained through the ALM is:

$$\min_{\mathcal{G}} L_\mu(W(\mathcal{G}), \theta, \alpha) = \frac{1}{2n}\|\mathbf{X} - \mathbf{X}W(\mathcal{G})\|_F^2 + \lambda\|W(\mathcal{G})\|_1 + \alpha_t h(W(\mathcal{G})) + \frac{\mu_t}{2}|h(W(\mathcal{G}))|^2. \tag{17}$$

The update rule for the parameters is:

$$\begin{aligned} W(\mathcal{G})_k, \theta_k &= \arg\min_{W(\mathcal{G}),\theta} L_\mu(W(\mathcal{G}), \theta, \alpha) \\ \alpha_{k+1} &= \alpha_k + \mu_k h\left(W(\mathcal{G})_k\right) \\ \mu_{k+1} &= \begin{cases} \eta\mu_k, & \text{if } |h\left(W(\mathcal{G})_k\right)| > \gamma\left|h\left(W(\mathcal{G})_{k-1}\right)\right| \\ \mu_k, & \text{otherwise} \end{cases} \end{aligned}, \tag{18}$$

where $\theta$ represents the parameters of a neural network used to fit a nonlinear function, and $\theta$ can be ignored for a linear model. The hyperparameters are usually set as $\eta = 10$ and $\gamma = \frac{1}{4}$.

In practice, the optimization stopping criterion is $h\left(W(\mathcal{G})_k\right) < \epsilon \in \left\{1e^{-6}, 1e^{-8}, 1e^{-10}\right\}$, which does not guarantee the output to be a DAG. Finally, for values in $W(\mathcal{G})$ with absolute values smaller than a threshold $\tau$, we set them to 0 in order to obtain a DAG as closely as possible. We use the implementation of NOTEARS algorithm provided by the authors, available at `https://github.com/xunzheng/notears`.

## B.7 GOLEM

GOLEM (Ng et al., 2020) proposed an improved loss function to address the numerical and ill-conditioned issues that often arise during the multiple iterations of optimization in NOTEARS. The

unconstrained optimization problem formulated by GOLEM is:

$$\min_{W(\mathcal{G}) \in \mathbb{R}^{d \times d}} \mathcal{S}_i(W(\mathcal{G}); \mathbf{X}) = \mathcal{L}_i(W(\mathcal{G}); \mathbf{X}) + \lambda_1 \|W(\mathcal{G})\|_1 + \lambda_2 h(W(\mathcal{G})), \quad (19)$$

where $\lambda_1$ and $\lambda_2$ are hyperparameters, $i \in \{1, 2\}$.

When assuming linear Gaussian with non-equal variances, that is, GOLEM-NV:

$$\mathcal{L}_1(W(\mathcal{G}); \mathbf{X}) = \frac{1}{2} \sum_{i=1}^{d} \log \left( \sum_{k=1}^{n} \left( X_i^{(k)} - W(\mathcal{G})_i^{\top} X^{(k)} \right)^2 \right) - \log |\det(I - W(\mathcal{G}))|, \quad (20)$$

where $X_i^{(k)}$ denotes $k$-th data point of $X_i$.

When assuming linear Gaussian with equal variances, that is, GOLEM-EV:

$$\mathcal{L}_2(W(\mathcal{G}); \mathbf{X}) = \frac{d}{2} \log \left( \sum_{i=1}^{d} \sum_{k=1}^{n} \left( X_i^{(k)} - W(\mathcal{G})_i^{\top} X^{(k)} \right)^2 \right) - \log |\det(I - W(\mathcal{G}))|. \quad (21)$$

The authors proved that in the case of linear Gaussian with equal variances, when the hard DAG constraint is not satisfied, the least-squares optimal solution of NOTEARS returns a cyclic graph, whereas the optimal solution of GOLEM-EV corresponds to the ground-truth. GOLEM combines the maximum likelihood objective function with a soft DAG constraint, replacing the least-squares objective function and hard DAG constraint, making the optimization easier to solve and the results better. We use the implementation of GOLEM-EV algorithm provided by the authors, available at `https://github.com/ignavierng/golem`.

## B.8 NOTEARS-MLP

NOTEARS-MLP (Zheng et al., 2020) extends the differentiable causal discovery framework to the nonlinear case. Each variable $X_j$ is defined as follows:

$$X_j = f_j(X_{pa(X_j)}, U_j), \forall j = 1, \cdots, d. \quad (22)$$

The authors proved that $f_j$ is independent of $X_k$ if and only if $\|\partial_k f_j\|_{L^2} = 0$, where $\|\cdot\|_{L^2}$ is the $L^2$-norm. Next, they define nonlinear causal effects through partial derivatives:

$$[W(f)]_{kj} = \|\partial_k f_j\|_{L^2}, \quad (23)$$

where $[W(f)]_{kj}$ is the causal effects from $X_k$ to $X_j$, and $W(f) = W(f_1, \ldots, f_d) \in \mathbb{R}^{d \times d}$.

In practice, neural networks are used to fitting nonlinear functional relationships $f_j$:

$$\mathrm{MLP}_j \left( \mathbf{X}; A^{(1)}, \ldots, A^{(h)} \right) = \sigma \left( A^{(h)} \sigma \left( \cdots A^{(2)} \sigma \left( A^{(1)} \mathbf{X} \right) \right) \right), \quad (24)$$

where $A^{(\ell)} \in \mathbb{R}^{m_\ell \times m_{\ell-1}}$, $\sigma$ is the activation function.

For the convenience of derivative computation, the authors proved that $\mathrm{MLP}_j$ are independent of $X_k$ if and only if the $k$-th column of the first-layer weight matrix $A^{(1)}$ is entirely zero. The parameters of $\mathrm{MLP}_j$ are $\theta_j = \left( A_j^{(1)}, \ldots, A_j^{(h)} \right)$. The authors ultimately obtain a weighted adjacency matrix representation that is independent of the depth of the neural network:

$$[W(\theta)]_{kj} = \left\| kth - \mathrm{column} \left( A_j^{(1)} \right) \right\|_2. \quad (25)$$

The objective function of NOTEARS-MLP is:

$$\min_{\theta} \frac{1}{n} \sum_{j=1}^{d} L \left( X_j, \mathrm{MLP}_j \left( \mathbf{X}; \theta_j \right) \right) + \lambda \left\| A_j^{(1)} \right\|_1 \quad \text{s.t.} \quad h(W(\theta)) = 0. \quad (26)$$

NOTEARS-MLP trains $d$ neural networks and represents the acyclicity constraint only through the first-layer parameters of neural networks. We use the implementation of NOTEARS-MLP algorithm provided by the authors, available at `https://github.com/xunzheng/notears`.

### B.9 GRAN-DAG

GraN-DAG also extends NOTEARS to the nonlinear case and considers the ANM data generation mechanism:

$$X_j = f_j(X_{pa(X_j)}) + U_j, \forall j = 1, \cdots, d. \tag{27}$$

The authors state that the parameters of $j$-th neural network (NN) are:

$$\phi_{(j)} = \left\{ W_{(j)}^{(1)}, \ldots, W_{(j)}^{(L+1)} \right\}, \tag{28}$$

where $W_{(j)}^{(\ell)}$ is the $\ell$-th weight matrix of the $j$-th NN.

They define the $j$-th connection matrix:

$$C_{(j)} = \left| W_{(j)}^{(L+1)} \right| \cdots \left| W_{(j)}^{(2)} \right| \left| W_{(j)}^{(1)} \right|. \tag{29}$$

Based on the above definition, the authors construct a weighted adjacency matrix related to the depth of the NN:

$$(W_\phi)_{ij} = \begin{cases} \sum_{k=1}^m \left( C_{(j)} \right)_{ki}, & \text{if } j \neq i \\ 0, & \text{otherwise} \end{cases}, \tag{30}$$

where $m$ is the output dimension of the NN.

The objective function of GraN-DAG is:

$$\max_\phi \mathbb{E}_{X \sim P(X)} \sum_{j=1}^d \log p_j \left( X_j \mid X_{pa(X_j)}; \phi_{(j)} \right) \quad \text{s.t. } h(\phi) = \text{Tr}\, e^{W_\phi} - d = 0. \tag{31}$$

The unconstrained objective function of GraN-DAG is:

$$\max_\phi \mathcal{L} \left( \phi, \alpha_t, \mu_t \right) = \mathbb{E}_{X \sim P(X)} \sum_{j=1}^d \log p_j \left( X_j \mid X_{pa(X_j)}; \phi_{(j)} \right) - \alpha_t h(\phi) - \frac{\mu_t}{2} h(\phi)^2. \tag{32}$$

When the data generation mechanism follows the nonlinear Gaussian additive noise model, it can be proven that the optimal solution of GraN-DAG corresponds to the ground-truth. We use the implementation of GraN-DAG algorithm provided by the authors, available at `https://github.com/kurowasan/GraN-DAG`.

### B.10 NOCURL

Since a DAG is related to curl-free functions on its edge set, NoCurl proposed a new representation of a DAG:

$$A = \gamma(W, p), \tag{33}$$

where $W$ is a skew-symmetric matrix with $W = -W^T$, $p \in \mathbb{R}^d$ is the potential function on the vertices of the graph.

The authors further proved that:

$$\gamma(W, p) = W \circ \text{ReLU}(\text{grad}(p)), \tag{34}$$

where grad is the gradient operator.

The optimization problem established by NoCurl is:

$$(W^*, p^*) = \underset{W,p}{\operatorname{argmin}} \quad F(\gamma(W, p), \mathbf{X}), \tag{35}$$

with the optimal DAG $A^* = W^* \circ \text{ReLU}\left(\text{grad}\left(p^*\right)\right)$. NoCurl implicitly ensures the acyclicity constraint, overcoming the shortcomings of the ALM, avoiding multiple iterations, and improving computational efficiency. We use the implementation of NoCurl algorithm provided by the authors, available at `https://github.com/fishmoon1234/DAG-NoCurl`.

### B.11 DAGMA

DAGMA (Bello et al., 2022) proposed a log-determinant form of acyclicity representation $h_{\mathrm{ldet}}^s(W) = -\log \det(sI - W \circ W) + d \log s$ $(s > 0)$, which has three advantages compared to the exponential acyclicity constraints $h_{\mathrm{expm}}(W) = \mathrm{Tr}(e^{W \circ W}) - d$ (Zheng et al., 2018) and polynomial acyclicity constraints $h_{\mathrm{poly}}(W) = \mathrm{Tr}[(I + \alpha W \circ W)^d] - d$ $(\alpha > 0)$ (Yu et al., 2019). The authors proved that:

$$h_{\mathrm{expm}}(W) = \sum_{k=0}^{\infty} \frac{1}{k!} \mathrm{Tr}\left((W \circ W)^k\right) - d,$$

$$h_{\mathrm{poly}}(W) = \sum_{k=0}^{d} \frac{\binom{d}{k}}{d^k} \mathrm{Tr}\left((W \circ W)^k\right) - d,$$

(36)

where $\mathrm{Tr}\left((W \circ W)^k\right)$ represents the information of cycles of length $k$. The information of cycles of length $k$ in $h_{\mathrm{expm}}(W)$ and $h_{\mathrm{poly}}(W)$ is weakened by $\frac{1}{k!}$ and $\frac{\binom{d}{k}}{d^k}$, respectively. It can be theoretically proven that $h_{\mathrm{ldet}}^s(W)$ is an upper bound of $h_{\mathrm{expm}}(W)$ and $h_{\mathrm{poly}}(W)$, retaining more information about the cycles.

The authors also proved that:

$$\nabla h_{\mathrm{expm}}(W) = 2\left(e^{W \circ W}\right)^{\top} \circ W$$

$$\nabla h_{\mathrm{poly}}(W) = 2\left(\left(I + \frac{1}{d} W \circ W\right)^{d-1}\right)^{\top} \circ W.$$

$$\nabla h_{\mathrm{ldet}}^s(W) = 2\left((sI - W \circ W)^{-1}\right)^{\top} \circ W$$

(37)

$\nabla h_{\mathrm{expm}}(W)$ and $\nabla h_{\mathrm{poly}}(W)$ are prone to the vanishing gradient problem. It can be theoretically proven that $\nabla h_{\mathrm{ldet}}^s(W)$ is an upper bound of $\nabla h_{\mathrm{expm}}(W)$ and $\nabla h_{\mathrm{poly}}(W)$, retaining more information about the cycles.

The third advantage is that, in practice, $h_{\mathrm{ldet}}^s(W)$ and $\nabla h_{\mathrm{ldet}}^s(W)$ are faster to compute. Because the computation of $h_{\mathrm{ldet}}^s(W)$ and $\nabla h_{\mathrm{ldet}}^s(W)$ involves matrix log-determinant and matrix inverse, both of which have been extensively studied and solved. In contrast, other acyclicity constraints and their partial derivatives involve multiple matrix-to-matrix multiplications, which are slower. We use the implementation of DAGMA algorithm provided by the authors, available at `https://github.com/kevinsbello/dagma`.

## C  RELATED WORK

**Differentiable causal discovery methods.** Building on traditional score-based causal discovery algorithms, NOTEARS (Zheng et al., 2018) transformed discrete constrained optimization into smooth equality-constrained optimization. This formulation has been extended to various settings, including more efficient linear models (GOLEM (Ng et al., 2020), NoCurl (Yu et al., 2021), NOFEARS (Wei et al., 2020), LEAST (Zhu et al., 2021)), neural networks (NOTEARS-MLP (Zheng et al., 2020), GraN-DAG (Lachapelle et al., 2019), DAGMA (Bello et al., 2022), DARING (He et al., 2021), CASTLE (Kyono et al., 2020)), generative adversarial networks (SAM (Kalainathan et al., 2022)), variational autoencoders (D-VAE (Zhang et al., 2019)), graph neural network (GAE (Ng et al., 2019), DAG-GNN (Yu et al., 2019)) , federated learning (FedDAG (Gao et al., 2021)), reinforcement learning (RL-BIC (Zhu et al., 2019)), interventional data (DCDI (Brouillard et al., 2020)), time series data (DYNOTEARS (Pamfil et al., 2020)), multi-domain data (DICD (Wang et al., 2022), ReScore (Zhang et al., 2023), CASPER (Liu et al., 2023)), and domain adaptation (CAE (Yang et al., 2021)). Although differentiable causal discovery has made significant progress, it is also affected by Var-Sortability (Reisach et al., 2021; 2023) and highly non-convex optimization problems (Ng et al., 2024). Recent research by Deng et al. (2024) shows that differentiable causal discovery methods can achieve scale invariance and global optimization when the correct loss function is used.

# D    TABLE RESULTS FOR RUNTIME OF THE BENCHMARK METHODS

The results in Table 8, 9, 10 and 11 show that differentiable causal discovery, exemplified by DAGMA, NOTEARS-MLP, and NoCurl, achieve superior performance with almost negligible runtime cost.

Table 8: Results for runtime (in seconds) on degree $k = 2$ graphs of 10 and 20 nodes. The reported results are the mean and standard deviation of the runtime over 10 repetitions across different graph types, vanilla and model assumption violation scenarios.

| Method | $d$ | Runtime (seconds) |
|---|---|---|
| PC | 10 | $1.29\pm0.24$ |
|  | 20 | $1.91\pm0.35$ |
| GES | 10 | $1.64\pm0.53$ |
|  | 20 | $7.82\pm2.03$ |
| DirectLiNGAM | 10 | $1.25\pm0.28$ |
|  | 20 | $2.06\pm0.43$ |
| Var-SortnRegress | 10 | $1.11\pm0.25$ |
|  | 20 | $1.26\pm0.29$ |
| $R^2$-SortnRegress | 10 | $1.13\pm0.16$ |
|  | 20 | $1.24\pm0.35$ |
| NOTEARS | 10 | $9.35\pm2.63$ |
|  | 20 | $35.57\pm4.71$ |
| GOLEM | 10 | $130.23\pm2.54$ |
|  | 20 | $178.52\pm3.41$ |
| NoCurl | 10 | $5.68\pm0.49$ |
|  | 20 | $10.29\pm1.84$ |
| CAM | 10 | $45.73\pm2.67$ |
|  | 20 | $113.37\pm3.85$ |
| NOTEARS-MLP | 10 | $4.70\pm0.82$ |
|  | 20 | $5.84\pm0.97$ |
| GraN-DAG | 10 | $119.28\pm2.15$ |
|  | 20 | $211.59\pm4.35$ |
| DAGMA | 10 | $\mathbf{2.41\pm0.32}$ |
|  | 20 | $\mathbf{3.19\pm0.48}$ |

# E   TABLE RESULTS ACROSS NODES, GRAPH TYPES, AND GRAPH DENSITIES

More experimental results for linear, nonlinear and MLP settings are reported in the Appendix as Table 9, 10, 11, 12, 13, 14, 15, 16, 17, 18, 19, 20, 21 shows.

Table 9: Linear Setting, for ER-2 graphs of 10, 20, 50 nodes.

| 10 nodes | Vanilla model SHD↓ | SID↓ | Latent confounders SHD↓ | SID↓ | Measurement error SHD↓ | SID↓ | Autoregressive SHD↓ | SID↓ | Heterogeneous SHD↓ | SID↓ | Unfaithful SHD↓ | SID↓ | Scale-variant SHD↓ | SID↓ | Missing SHD↓ | SID↓ | Mechanism violation SHD↓ | SID↓ |
|---|---|---|---|---|---|---|---|---|---|---|---|---|---|---|---|---|---|---|
| Random | 25.6±3.1 | 57.9±9.5 | 27.9±2.3 | 67.8±7.8 | 25.9±3.5 | 60.4±11.3 | 27.9±3.2 | 62.0±8.1 | 26.3±3.5 | 57.7±7.6 | 26.1±3.7 | 60.7±12.9 | 26.7±3.0 | 64.0±8.7 | 27.5±3.4 | 63.1±6.0 | 28.3±3.0 | 63.6±7.8 |
| PC | 12.4±3.1 | 40.9±13.4 | 18.1±4.7 | 58.1±15.6 | 19.4±4.1 | 48.0±13.1 | 14.5±2.0 | 44.8±9.5 | 13.8±2.6 | 47.5±10.3 | 13.9±3.2 | 40.6±10.4 | 12.4±3.1 | 40.9±13.4 | 13.0±4.7 | 44.8±16.0 | 17.1±2.5 | 64.9±10.0 |
| GES | 13.8±7.8 | 32.0±13.6 | 25.9±7.7 | 42.6±14.0 | 20.2±4.8 | 46.2±16.7 | 20.8±5.5 | 49.7±11.5 | 15.7±4.6 | 35.2±12.2 | 17.8±6.4 | 39.0±15.7 | 13.8±7.8 | 32.0±13.6 | 10.1±5.2 | 25.4±12.6 | 16.2±2.7 | 57.8±10.7 |
| DirectLiNGAM | 19.6±3.3 | 46.1±10.6 | 20.4±5.0 | 42.0±6.0 | 17.6±2.4 | 48.8±12.4 | 19.7±4.2 | 50.4±8.4 | 16.3±3.9 | 34.7±9.9 | 19.7±4.3 | 44.7±13.9 | 21.8±4.3 | 62.6±9.3 | 20.1±4.3 | 49.8±11.1 | 20.0±0.0 | 63.5±7.7 |
| Var-SortnRegress | 11.2±3.5 | 8.4±8.5 | 17.6±5.8 | 12.6±9.9 | 19.6±2.8 | 11.4±8.7 | 18.8±2.4 | 16.5±10.6 | 17.9±3.3 | 8.6±9.3 | 12.4±3.1 | 13.5±8.0 | 30.7±5.1 | 54.5±10.3 | 7.3±3.5 | 9.3±8.4 | 15.6±3.3 | 39.0±6.7 |
| R²-SortnRegress | 20.2±4.8 | 32.4±14.0 | 25.7±4.1 | 37.6±13.0 | 25.6±6.0 | 39.2±16.0 | 25.6±4.9 | 38.8±19.0 | 26.0±5.4 | 37.0±14.4 | 29.8±4.8 | 51.0±11.3 | 20.2±4.8 | 32.4±14.0 | 20.5±6.7 | 32.0±8.8 | 20.3±3.7 | 66.1±9.7 |
| NOTEARS | 1.5±1.6 | 1.8±4.2 | 8.5±3.9 | 9.5±8.1 | 12.5±2.0 | 19.6±8.6 | 12.2±3.6 | 27.5±14.2 | 5.5±2.7 | 5.4±5.1 | 2.7±3.1 | 3.1±5.1 | 18.0±1.2 | 60.5±7.3 | 2.3±1.7 | 6.4±8.6 | 19.0±0.9 | 58.3±8.0 |
| GOLEM | 1.4±1.4 | 0.4±1.2 | 6.7±2.8 | 14.2±9.8 | 17.8±2.5 | 43.1±13.3 | 16.6±4.0 | 34.9±16.9 | 6.5±4.5 | 9.8±8.1 | 2.1±2.2 | 0.6±1.8 | 17.5±1.2 | 64.4±6.8 | 1.7±1.7 | 6.2±10.8 | 18.6±1.6 | 52.7±4.3 |
| NoCurl | 2.0±1.8 | 5.1±5.8 | 9.1±4.2 | 5.4±3.9 | 11.8±1.8 | 17.9±8.4 | 14.8±2.5 | 17.5±10.8 | 6.6±2.9 | 5.5±5.7 | 2.2±2.3 | 2.0±4.4 | 27.2±5.1 | 69.9±7.9 | 3.1±3.2 | 4.7±5.8 | 19.1±1.0 | 58.9±9.5 |
| DAGMA | 1.2±1.2 | 3.3±5.3 | 8.4±3.9 | 8.8±7.7 | 12.6±2.5 | 18.5±8.6 | 12.2±3.6 | 28.4±15.3 | 5.5±2.3 | 12.0±8.2 | 2.1±2.2 | 0.6±1.8 | 17.9±1.4 | 58.7±6.8 | 1.5±1.4 | 4.5±7.1 | 19.0±0.9 | 58.3±8.0 |

| 20 nodes | SHD↓ | SID↓ | SHD↓ | SID↓ | SHD↓ | SID↓ | SHD↓ | SID↓ | SHD↓ | SID↓ | SHD↓ | SID↓ | SHD↓ | SID↓ | SHD↓ | SID↓ | SHD↓ | SID↓ |
|---|---|---|---|---|---|---|---|---|---|---|---|---|---|---|---|---|---|---|
| Random | 107.9±7.0 | 253.2±26.3 | 105.2±8.0 | 239.8±22.1 | 106.0±6.3 | 244.8±33.5 | 109.1±8.0 | 254.9±18.8 | 107.8±5.7 | 247.6±26.1 | 106.7±5.5 | 246.6±40.8 | 103.9±4.2 | 239.6±24.0 | 107.3±9.9 | 237.7±34.2 | 104.2±7.1 | 222.9±26.9 |
| PC | 31.6±6.5 | 168.5±27.6 | 50.5±4.3 | 213.0±28.5 | 48.0±7.8 | 201.6±30.4 | 36.7±3.7 | 193.1±27.3 | 33.5±5.7 | 181.3±26.0 | 29.6±7.3 | 145.1±44.3 | 31.6±6.5 | 168.5±27.6 | 29.8±5.5 | 166.1±26.7 | 34.5±3.4 | 226.3±36.7 |
| GES | 34.3±24.6 | 104.5±51.1 | 133.5±12.5 | 153.7±32.0 | 57.6±8.2 | 203.7±40.0 | 53.1±15.2 | 116.9±42.5 | 49.4±18.3 | 114.6±29.4 | 37.8±17.0 | 101.8±36.6 | 34.3±24.6 | 104.5±51.1 | 48.4±16.0 | 136.2±45.0 | 32.6±3.3 | 190.1±42.6 |
| DirectLiNGAM | 55.7±9.1 | 166.4±31.0 | 72.5±9.7 | 152.6±26.2 | 40.0±6.7 | 132.3±22.5 | 55.0±9.0 | 166.7±17.4 | 50.5±11.2 | 158.6±18.4 | 51.4±10.5 | 165.6±33.6 | 59.4±6.1 | 260.5±29.1 | 53.4±11.0 | 176.8±27.5 | 39.9±0.3 | 223.5±26.0 |
| Var-SortnRegress | 39.0±16.3 | 26.8±15.9 | 117.3±6.0 | 34.0±12.2 | 74.8±11.7 | 36.6±16.6 | 85.1±11.6 | 37.9±21.6 | 74.4±13.0 | 27.9±14.5 | 45.3±9.4 | 33.5±14.3 | 128.3±24.7 | 190.7±27.7 | 39.5±12.0 | 36.1±18.7 | 33.6±3.9 | 174.0±32.8 |
| R²-SortnRegress | 88.7±20.8 | 117.3±38.6 | 136.6±10.3 | 96.7±39.1 | 89.4±15.1 | 156.6±42.3 | 106.9±15.4 | 140.8±45.6 | 110.7±13.9 | 123.0±37.0 | 107.7±15.5 | 148.8±36.8 | 88.7±20.8 | 117.3±38.6 | 95.6±28.5 | 132.3±38.9 | 43.2±7.1 | 247.2±36.0 |
| NOTEARS | 6.5±4.1 | 23.4±17.0 | 26.3±7.5 | 106.7±41.0 | 34.8±8.0 | 114.8±24.0 | 39.5±8.2 | 102.1±29.5 | 14.8±6.1 | 49.8±14.6 | 9.0±5.9 | 17.9±11.5 | 38.4±0.8 | 207.1±22.7 | 7.0±4.3 | 10.0±13.3 | 37.0±1.1 | 198.8±24.1 |
| GOLEM | 4.4±2.2 | 18.0±13.5 | 24.5±8.2 | 98.0±36.9 | 39.5±1.1 | 212.7±24.2 | 39.6±3.5 | 203.2±21.2 | 14.5±6.6 | 41.7±20.5 | 6.8±3.7 | 20.8±12.1 | 36.6±2.2 | 224.1±30.0 | 4.7±2.9 | 11.5±12.3 | 38.1±1.6 | 194.2±30.7 |
| NoCurl | 8.0±6.0 | 13.6±9.0 | 60.1±10.1 | 39.8±13.1 | 38.3±8.9 | 69.7±27.8 | 60.0±9.8 | 59.1±16.2 | 23.7±8.0 | 22.2±12.4 | 10.5±7.3 | 21.8±15.5 | 79.0±8.2 | 264.2±20.4 | 8.1±5.8 | 21.5±21.1 | 38.5±1.7 | 200.8±27.6 |
| DAGMA | 5.4±3.9 | 14.2±10.3 | 20.7±6.4 | 111.9±42.3 | 34.2±7.0 | 115.7±24.0 | 37.7±7.0 | 148.6±32.4 | 12.7±5.7 | 41.5±15.6 | 7.4±4.2 | 12.5±10.4 | 37.3±1.1 | 206.3±21.4 | 6.6±6.0 | 6.2±10.4 | 37.0±1.1 | 198.8±24.1 |

| 50 nodes | SHD↓ | SID↓ | SHD↓ | SID↓ | SHD↓ | SID↓ | SHD↓ | SID↓ | SHD↓ | SID↓ | SHD↓ | SID↓ | SHD↓ | SID↓ | SHD↓ | SID↓ | SHD↓ | SID↓ |
|---|---|---|---|---|---|---|---|---|---|---|---|---|---|---|---|---|---|---|
| Random | 637.2±24.3 | 1393.2±162.2 | 652.6±21.5 | 1416.5±109.4 | 642.9±16.1 | 1436.3±96.8 | 635.5±15.6 | 1431.1±147.4 | 641.7±17.0 | 1387.0±118.5 | 640.7±18.0 | 1406.1±147.9 | 634.8±10.4 | 1456.3±112.2 | 632.7±16.5 | 1372.7±178.7 | 635.5±20.2 | 1413.6±154.1 |
| DirectLiNGAM | 146.4±27.3 | 781.7±103.0 | 300.9±18.0 | 884.4±175.3 | 104.4±13.1 | 777.6±122.6 | 178.8±33.9 | 683.8±126.1 | 130.2±16.8 | 702.2±144.2 | 143.7±21.8 | 739.4±113.9 | 130.5±11.4 | 1248.0±112.0 | 158.3±19.3 | 810.3±120.6 | 99.8±0.7 | 1051.4±84.3 |
| Var-SortnRegress | 106.0±16.3 | 113.1±46.8 | 920.6±16.1 | 1457.7±19.3 | 281.6±27.1 | 208.3±51.2 | 427.3±44.8 | 207.3±83.8 | 309.5±48.3 | 121.8±43.7 | 138.5±27.7 | 127.5±62.4 | 665.6±51.0 | 1154.7±154.3 | 114.3±23.8 | 112.6±37.3 | 86.4±6.2 | 854.3±76.0 |
| R²-SortnRegress | 388.3±72.4 | 555.8±142.2 | 946.6±22.1 | 401.3±104.6 | 346.0±18.3 | 903.4±138.6 | 552.5±56.8 | 853.3±215.0 | 549.4±68.9 | 573.7±133.0 | 442.7±105.4 | 642.1±176.9 | 388.3±72.4 | 555.8±142.2 | 420.2±68.4 | 547.0±78.1 | 108.1±7.9 | 1313.4±89.6 |
| NOTEARS | 15.2±6.2 | 71.4±53.5 | 46.5±15.4 | 266.5±90.3 | 81.0±10.2 | 619.0±88.3 | 122.4±32.7 | 944.2±134.6 | 23.3±5.4 | 181.3±51.5 | 16.7±5.2 | 66.6±54.9 | 90.8±3.8 | 972.0±82.0 | 15.6±6.9 | 91.7±50.6 | 92.4±1.9 | 959.7±97.0 |
| GOLEM | 11.8±4.7 | 58.9±38.1 | 52.7±8.6 | 290.4±98.3 | 97.2±1.0 | 1028.7±84.7 | 99.2±5.0 | 1026.4±95.8 | 25.1±7.0 | 142.7±60.3 | 13.2±4.1 | 64.3±46.7 | 82.4±7.7 | 952.1±94.5 | 12.2±2.8 | 79.8±56.5 | 94.9±2.7 | 976.5±99.4 |
| NoCurl | 22.8±12.1 | 60.9±41.0 | 332.1±30.5 | 199.6±105.7 | 109.1±15.7 | 402.7±76.7 | 294.3±54.6 | 287.8±166.4 | 74.4±15.1 | 99.5±49.5 | 25.4±12.1 | 59.3±59.3 | 260.3±42.9 | 1584.5±97.9 | 30.5±10.5 | 61.5±53.6 | 93.1±2.9 | 955.1±98.1 |
| DAGMA | 12.4±4.3 | 47.5±39.7 | 42.7±10.1 | 284.2±107.8 | 81.2±9.4 | 635.0±72.0 | 103.0±12.3 | 932.1±91.9 | 24.4±7.6 | 170.7±62.0 | 14.6±4.4 | 55.0±33.3 | 90.6±4.0 | 969.3±88.7 | 13.3±4.3 | 79.0±54.9 | 92.4±1.9 | 959.7±97.0 |

Table 10: Nonlinear Setting, for ER-2 graphs of 10, 20, 50 nodes.

| 10 nodes | Vanilla model SHD↓ | SID↓ | Latent confounders SHD↓ | SID↓ | Measurement error SHD↓ | SID↓ | Autoregressive SHD↓ | SID↓ | Heterogeneous SHD↓ | SID↓ | Unfaithful SHD↓ | SID↓ | Scale-variant SHD↓ | SID↓ | Missing SHD↓ | SID↓ | Mechanism violation SHD↓ | SID↓ |
|---|---|---|---|---|---|---|---|---|---|---|---|---|---|---|---|---|---|---|
| Random | 27.7±3.2 | 63.6±11.2 | 27.4±2.5 | 59.3±9.4 | 27.4±3.6 | 63.2±7.7 | 28.5±2.3 | 65.8±8.6 | 24.9±3.2 | 62.1±10.1 | 28.7±2.1 | 69.8±7.9 | 27.5±2.5 | 69.9±5.9 | 27.0±4.2 | 68.3±5.9 | 24.5±6.0 | 54.5±10.7 |
| PC | 17.1±2.5 | 64.9±10.0 | 18.5±1.9 | 62.8±9.1 | 18.2±1.1 | 61.6±12.2 | 18.4±1.3 | 61.2±9.7 | 21.3±3.2 | 56.5±10.4 | 15.4±1.4 | 57.3±9.5 | 17.1±2.5 | 64.9±10.0 | 17.8±3.1 | 63.5±10.5 | 12.4±3.1 | 40.9±13.4 |
| GES | 16.2±2.2 | 57.8±10.7 | 17.1±2.1 | 55.2±15.5 | 17.0±1.0 | 52.8±9.1 | 17.2±2.0 | 54.9±7.8 | 19.8±2.5 | 55.7±8.4 | 17.3±2.0 | 54.9±7.3 | 16.2±2.2 | 57.8±10.7 | 16.6±2.6 | 56.2±9.9 | 13.8±7.8 | 32.0±13.6 |
| CAM | 4.7±1.9 | 16.3±9.6 | 10.4±2.8 | 28.2±5.3 | 13.9±1.9 | 48.2±7.4 | 12.5±3.0 | 33.9±16.3 | 13.8±2.9 | 21.7±12.8 | 7.1±3.2 | 24.7±13.0 | 4.7±1.9 | 16.3±9.5 | 6.5±2.1 | 15.4±7.3 | 17.8±4.4 | 45.9±17.0 |
| NOTEARS-MLP | 12.4±2.2 | 36.3±7.1 | 17.0±1.7 | 49.2±8.6 | 16.5±4.8 | 48.9±4.8 | 17.0±3.7 | 47.7±11.0 | 16.4±3.7 | 42.6±10.6 | 11.4±2.1 | 43.8±9.2 | 16.1±2.5 | 48.3±9.7 | 12.3±2.1 | 33.6±7.0 | 5.9±2.5 | 19.7±8.7 |
| GraN-DAG | 12.7±2.4 | 33.2±10.6 | 15.6±2.1 | 42.4±8.8 | 20.0±1.1 | 63.8±11.3 | 16.2±2.3 | 44.2±10.0 | 14.8±2.1 | 40.5±10.8 | 11.2±3.1 | 37.7±10.0 | 10.0±3.7 | 26.1±12.0 | 10.4±3.4 | 25.7±8.1 | 3.3±3.1 | 9.5±11.6 |
| DAGMA | 13.5±2.0 | 40.7±8.1 | 18.6±2.0 | 62.0±13.3 | 17.3±1.3 | 54.9±8.3 | 19.0±2.0 | 56.6±10.5 | 16.4±4.3 | 36.6±15.7 | 8.6±3.1 | 29.6±14.8 | 15.7±2.7 | 53.3±12.9 | 13.7±2.7 | 41.4±7.8 | 3.3±3.1 | 9.5±11.6 |

| 20 nodes | SHD↓ | SID↓ | SHD↓ | SID↓ | SHD↓ | SID↓ | SHD↓ | SID↓ | SHD↓ | SID↓ | SHD↓ | SID↓ | SHD↓ | SID↓ | SHD↓ | SID↓ | SHD↓ | SID↓ |
|---|---|---|---|---|---|---|---|---|---|---|---|---|---|---|---|---|---|---|
| Random | 103.7±7.8 | 251.0±26.8 | 102.5±8.9 | 237.5±33.7 | 103.5±5.2 | 230.9±25.5 | 105.6±7.2 | 249.0±31.2 | 108.1±1.1 | 256.4±27.0 | 103.5±6.1 | 232.6±21.6 | 106.7±4.8 | 247.7±36.9 | 102.7±7.8 | 233.2±29.6 | 101.1±5.7 | 234.6±42.6 |
| PC | 34.5±3.4 | 226.3±36.7 | 37.8±1.8 | 202.0±22.8 | 36.2±2.0 | 213.2±33.9 | 38.8±2.8 | 226.8±35.4 | 49.9±6.3 | 213.9±24.8 | 32.7±3.1 | 221.6±29.5 | 34.5±3.4 | 226.3±36.7 | 34.2±2.6 | 217.9±31.6 | 31.6±6.5 | 168.5±27.6 |
| GES | 32.6±3.3 | 190.1±42.6 | 37.7±2.5 | 199.2±20.2 | 34.9±2.4 | 191.0±41.3 | 36.9±2.3 | 199.0±28.1 | 51.0±5.4 | 211.7±23.6 | 32.7±7.0 | 187.9±35.8 | 32.6±3.3 | 190.1±42.6 | 31.6±1.9 | 189.9±33.4 | 34.3±24.6 | 104.5±51.1 |
| CAM | 15.0±2.4 | 70.8±21.7 | 34.5±2.8 | 166.4±24.5 | 31.0±2.1 | 178.9±32.0 | 27.9±6.6 | 125.4±25.5 | 41.3±5.2 | 90.3±25.8 | 16.5±5.2 | 119.4±40.6 | 15.0±2.4 | 70.8±21.7 | 13.1±2.6 | 81.5±21.5 | 41.2±8.1 | 213.4±47.0 |
| NOTEARS-MLP | 27.8±4.0 | 133.3±28.4 | 37.8±1.5 | 204.4±24.8 | 33.0±2.4 | 174.1±33.6 | 39.9±0.3 | 215.2±26.4 | 40.0±0.0 | 216.0±24.9 | 24.8±2.1 | 163.0±26.5 | 32.0±3.8 | 181.0±36.1 | 27.4±3.4 | 139.8±25.1 | 19.2±4.9 | 87.2±17.7 |
| GraN-DAG | 30.8±4.2 | 148.8±26.6 | 48.1±2.1 | 218.2±25.3 | 41.2±7.0 | 208.0±24.4 | 36.8±2.0 | 194.3±22.2 | 32.3±4.1 | 161.7±30.9 | 28.3±4.0 | 162.2±30.6 | 31.0±4.3 | 150.7±26.2 | 31.0±3.2 | 149.1±27.3 | 16.3±2.1 | 198.8±26.5 |
| DAGMA | 27.3±4.4 | 141.9±36.4 | 36.2±1.3 | 175.0±21.9 | 35.1±2.1 | 196.0±31.5 | 39.9±1.1 | 213.5±27.5 | 40.0±0.0 | 216.0±24.9 | 19.8±6.0 | 125.1±33.0 | 32.8±2.6 | 193.4±32.6 | 27.3±3.3 | 141.2±39.0 | 14.9±5.8 | 69.7±35.0 |

| 50 nodes | SHD↓ | SID↓ | SHD↓ | SID↓ | SHD↓ | SID↓ | SHD↓ | SID↓ | SHD↓ | SID↓ | SHD↓ | SID↓ | SHD↓ | SID↓ | SHD↓ | SID↓ | SHD↓ | SID↓ |
|---|---|---|---|---|---|---|---|---|---|---|---|---|---|---|---|---|---|---|
| Random | 646.2±13.6 | 1455.0±112.9 | 633.9±11.3 | 1427.1±145.2 | 651.7±18.8 | 1428.8±149.6 | 628.4±23.2 | 1412.3±115.2 | 641.6±18.3 | 1442.8±163.8 | 643.6±15.9 | 1427.0±172.9 | 630.9±14.8 | 1427.1±156.7 | 636.5±17.5 | 1438.1±123.6 | 643.8±21.2 | 1365.4±120.0 |
| PC | 89.1±3.2 | 1211.6±139.1 | 132.5±7.1 | 1238.3±87.9 | 91.4±2.0 | 1072.9±145.2 | 105.3±6.7 | 1176.0±102.5 | 151.5±10.1 | 1211.6±134.5 | 63.0±4.7 | 1150.8±141.6 | 89.1±3.2 | 1136.8±139.1 | 86.4±4.3 | 1119.2±109.6 | 53.6±6.2 | 713.6±112.6 |
| GES | 88.6±9.2 | 1002.6±98.8 | 143.7±8.8 | 1341.3±89.3 | 84.4±2.0 | 881.6±105.2 | 120.4±9.0 | 1142.2±82.3 | 225.8±10.3 | 1224.0±100.4 | 80.0±5.3 | 1057.4±138.7 | 79.7±3.4 | 903.2±97.2 | 80.6±5.1 | 959.9±113.7 | 51.8±23.6 | 315.4±136.3 |
| CAM | 41.4±4.2 | 460.0±80.1 | 75.6±4.2 | 843.1±84.5 | 80.6±3.5 | 864.8±89.8 | 89.2±11.6 | 729.9±127.3 | 127.4±11.1 | 469.5±74.8 | 40.6±5.6 | 591.0±85.8 | 41.4±4.2 | 460.0±80.1 | 41.9±8.4 | 448.3±56.2 | 77.8±11.8 | 877.2±165.6 |
| NOTEARS-MLP | 71.1±5.9 | 733.4±99.1 | 89.2±9.3 | 936.4±97.2 | 87.2±3.0 | 911.1±108.9 | 101.7±4.8 | 1039.6±88.3 | 89.1±9.8 | 807.4±112.3 | 69.5±4.1 | 829.0±74.5 | 78.3±3.7 | 876.2±99.7 | 69.0±4.7 | 689.2±66.6 | 44.7±5.9 | 543.8±72.1 |
| GraN-DAG | 86.0±3.8 | 832.5±115.8 | 92.4±7.3 | 903.8±94.5 | 101.0±0.0 | 1041.8±86.5 | 94.8±2.0 | 964.7±80.7 | 96.0±2.3 | 995.7±85.8 | 83.7±5.4 | 916.7±83.2 | 88.8±6.1 | 881.7±102.7 | 91.5±2.9 | 940.4±78.5 | 90.6±6.2 | 953.6±92.1 |
| DAGMA | 69.9±5.0 | 737.2±99.2 | 78.1±6.3 | 873.4±82.3 | 86.6±3.1 | 924.5±111.9 | 100.0±0.0 | 1039.4±88.3 | 100.0±0.0 | 1039.4±88.3 | 55.6±7.6 | 689.6±60.7 | 84.3±4.5 | 1018.1±91.2 | 69.2±8.2 | 708.2±95.0 | 39.5±11.0 | 480.8±86.2 |

Table 11: MLP Setting, for ER-2 graphs of 10, 20, 50 nodes.

| 10 nodes | Vanilla model SHD↓ | SID↓ | Latent confounders SHD↓ | SID↓ | Measurement error SHD↓ | SID↓ | Autoregressive SHD↓ | SID↓ | Heterogeneous SHD↓ | SID↓ | Unfaithful SHD↓ | SID↓ | Scale-variant SHD↓ | SID↓ | Missing SHD↓ | SID↓ | Mechanism violation SHD↓ | SID↓ |
|---|---|---|---|---|---|---|---|---|---|---|---|---|---|---|---|---|---|---|
| CAM | 12.4±3.6 | 39.3±16.5 | 16.6±4.2 | 42.4±17.7 | 19.7±4.7 | 59.6±12.0 | 16.0±3.4 | 46.0±16.6 | 19.9±3.6 | 49.1±7.2 | 13.0±3.2 | 36.0±14.0 | 12.4±3.6 | 39.3±16.5 | 13.2±3.6 | 44.7±16.7 | 17.8±4.4 | 45.9±17.0 |
| NOTEARS-MLP | 8.1±2.7 | 22.2±10.6 | 11.7±5.5 | 33.3±17.0 | 18.5±3.7 | 47.1±11.9 | 15.7±4.3 | 39.8±9.6 | 14.5±2.2 | 34.3±10.4 | 14.8±3.9 | 32.3±12.8 | 18.2±2.6 | 61.0±11.0 | 9.0±3.3 | 22.0±10.0 | 5.9±2.5 | 19.7±8.7 |

| 20 nodes | SHD↓ | SID↓ | SHD↓ | SID↓ | SHD↓ | SID↓ | SHD↓ | SID↓ | SHD↓ | SID↓ | SHD↓ | SID↓ | SHD↓ | SID↓ | SHD↓ | SID↓ | SHD↓ | SID↓ |
|---|---|---|---|---|---|---|---|---|---|---|---|---|---|---|---|---|---|---|
| CAM | 25.2±4.3 | 151.1±62.3 | 32.5±7.6 | 189.0±34.6 | 50.3±7.3 | 230.8±30.1 | 36.4±9.3 | 164.0±55.5 | 48.3±8.0 | 205.5±38.1 | 28.5±5.9 | 149.3±47.7 | 25.2±8.3 | 151.1±62.3 | 26.3±8.6 | 158.1±63.8 | 41.2±8.1 | 213.4±47.0 |
| NOTEARS-MLP | 15.6±3.6 | 74.3±17.2 | 25.8±3.8 | 160.6±39.6 | 37.3±2.1 | 197.4±25.0 | 36.9±5.0 | 187.3±23.3 | 29.4±6.0 | 98.6±26.8 | 28.4±4.2 | 105.8±27.7 | 34.2±3.5 | 209.1±21.2 | 15.9±3.7 | 74.9±20.2 | 19.2±4.9 | 87.2±17.7 |

| 50 nodes | SHD↓ | SID↓ | SHD↓ | SID↓ | SHD↓ | SID↓ | SHD↓ | SID↓ | SHD↓ | SID↓ | SHD↓ | SID↓ | SHD↓ | SID↓ | SHD↓ | SID↓ | SHD↓ | SID↓ |
|---|---|---|---|---|---|---|---|---|---|---|---|---|---|---|---|---|---|---|
| CAM | 59.7±10.9 | 837.5±159.7 | 83.4±12.1 | 942.5±164.2 | 128.8±10.3 | 1220.2±124.7 | 101.3±17.7 | 855.8±172.9 | 126.7±18.0 | 1008.1±135.9 | 65.2±9.1 | 717.1±103.0 | 59.7±10.9 | 837.5±159.7 | 61.4±11.4 | 809.9±158.3 | 77.8±11.8 | 877.2±165.6 |
| NOTEARS-MLP | 41.5±11.3 | 378.3±104.5 | 67.4±9.7 | 788.6±145.5 | 93.7±2.6 | 958.2±115.7 | 113.7±21.6 | 1003.5±131.0 | 74.1±4.0 | 826.4±108.6 | 67.2±4.5 | 724.8±67.8 | 87.4±4.6 | 1052.0±107.0 | 41.0±9.9 | 392.6±90.5 | 44.7±5.9 | 543.8±72.1 |

Table 12: Linear Setting, for ER-4 graphs of 10, 20, 50 nodes.

| 10 nodes | Vanilla model SHD↓ | SID↓ | Latent confounders SHD↓ | SID↓ | Measurement error SHD↓ | SID↓ | Autoregressive SHD↓ | SID↓ | Heterogeneous SHD↓ | SID↓ | Unfaithful SHD↓ | SID↓ | Scale-variant SHD↓ | SID↓ | Missing SHD↓ | SID↓ | Mechanism violation SHD↓ | SID↓ |
|---|---|---|---|---|---|---|---|---|---|---|---|---|---|---|---|---|---|---|
| Random | 31.5±4.0 | 75.9±6.8 | 31.9±2.0 | 73.4±7.3 | 33.8±3.3 | 77.9±6.7 | 31.8±4.4 | 75.5±6.9 | 33.2±3.2 | 75.2±7.0 | 31.8±4.0 | 72.5±7.9 | 30.7±3.7 | 73.5±7.3 | 33.1±3.4 | 76.7±5.7 | **34.0±1.1** | 75.1±6.3 |
| PC | 28.6±3.2 | 65.0±7.2 | 30.4±2.1 | 72.0±7.1 | 27.9±5.3 | 64.7±14.6 | 31.2±3.1 | 70.1±9.9 | 29.0±2.1 | 67.3±6.8 | 28.8±2.8 | 71.4±8.4 | 28.6±3.2 | 65.0±7.2 | 28.4±2.7 | 71.1±8.4 | 36.7±1.7 | 81.6±5.2 |
| GES | 27.9±2.7 | 65.0±10.6 | 26.3±3.9 | 66.8±9.6 | 23.7±4.8 | 61.7±10.5 | 20.4±3.9 | 49.4±14.0 | 28.5±3.5 | 68.9±8.7 | 25.1±5.4 | 61.2±9.7 | 27.9±2.7 | 65.0±10.6 | 25.0±4.1 | 63.4±10.6 | 37.3±1.0 | 77.3±11.8 |
| DirectLiNGAM | 27.2±4.9 | 66.7±8.4 | 24.9±5.7 | 61.0±11.9 | 29.7±2.0 | 63.4±7.3 | 23.5±4.6 | 55.8±15.2 | 25.9±3.6 | 58.1±7.3 | 26.6±4.6 | 63.3±10.6 | 36.9±3.6 | 81.8±5.8 | 23.2±5.3 | 65.6±9.5 | 40.0±0.0 | 81.4±6.7 |
| Var-SortnRegress | 5.4±3.4 | 14.5±7.2 | 9.1±3.2 | 24.2±13.3 | **10.6±3.4** | **20.9±7.6** | **8.2±2.0** | **26.7±12.3** | **7.2±2.6** | **15.4±8.4** | 8.7±3.7 | 25.9±12.0 | 23.6±3.6 | 61.9±6.5 | 6.1±2.8 | 19.9±9.5 | 34.9±1.6 | **65.9±6.6** |
| R²-SortnRegress | 8.8±3.8 | 23.1±13.2 | 11.4±2.9 | 33.1±8.3 | 20.0±4.9 | 54.1±12.1 | 18.3±5.9 | 53.5±13.3 | 12.2±5.0 | 33.7±16.0 | 17.9±4.9 | 51.9±14.2 | **8.8±3.8** | **23.1±13.2** | 11.2±2.9 | 36.8±8.4 | 39.1±1.2 | 85.0±3.5 |
| NOTEARS | 4.2±2.6 | 15.6±9.3 | 9.6±4.4 | 22.7±19.2 | **14.1±5.5** | **27.5±10.7** | **9.6±4.0** | 27.0±15.4 | 8.5±3.1 | 22.4±6.9 | 4.3±3.4 | 18.5±11.7 | 35.9±3.4 | 83.3±3.9 | 5.4±3.3 | 23.1±10.1 | 38.6±0.7 | 73.3±5.7 |
| GOLEM | 2.9±3.2 | 11.8±10.6 | 10.2±3.8 | 28.1±14.6 | 28.2±2.4 | 61.0±9.3 | 17.4±7.1 | 49.3±18.2 | 8.6±2.9 | 21.1±11.4 | **1.9±1.6** | **9.0±8.8** | 37.1±3.0 | 84.2±3.0 | **2.2±2.2** | 12.7±13.6 | **38.4±0.8** | **71.2±4.7** |
| NoCurl | 4.5±2.7 | 19.8±9.2 | 10.1±4.4 | 26.9±11.0 | 14.2±5.3 | 27.9±9.8 | **9.6±3.6** | **26.8±10.3** | **7.3±1.7** | 23.6±7.0 | 5.6±3.6 | 22.6±16.9 | 37.9±4.1 | **81.8±6.4** | 6.9±4.1 | 25.8±12.3 | 38.7±0.8 | 72.6±5.7 |
| DAGMA | **2.4±2.5** | **9.3±9.4** | **7.4±4.1** | **19.6±11.6** | 15.2±5.1 | 30.5±9.2 | 10.6±6.5 | 28.5±16.5 | 7.9±3.7 | **16.4±10.6** | 5.1±2.4 | 18.0±8.0 | 34.7±3.4 | 82.5±3.4 | 3.1±2.6 | **9.1±6.5** | 38.6±0.7 | 73.3±5.7 |

| 20 nodes | SHD↓ | SID↓ | SHD↓ | SID↓ | SHD↓ | SID↓ | SHD↓ | SID↓ | SHD↓ | SID↓ | SHD↓ | SID↓ | SHD↓ | SID↓ | SHD↓ | SID↓ | SHD↓ | SID↓ |
|---|---|---|---|---|---|---|---|---|---|---|---|---|---|---|---|---|---|---|
| Random | 117.9±7.1 | 320.8±13.1 | 115.3±6.6 | 317.1±11.7 | 116.5±5.2 | 318.0±9.8 | 115.3±3.9 | 304.7±23.3 | 113.3±6.4 | 307.3±18.8 | 115.2±7.0 | 299.8±23.2 | 116.4±5.0 | 306.5±15.5 | 109.7±4.5 | 313.6±13.5 | 118.2±3.7 | 313.1±16.8 |
| PC | 76.6±2.2 | 304.8±19.8 | 86.6±4.7 | 319.9±23.4 | 87.7±8.9 | 304.1±21.0 | 81.1±5.6 | 290.5±22.0 | 77.8±5.1 | 306.6±25.6 | 76.4±6.3 | 301.8±22.3 | 76.6±2.2 | 304.8±19.8 | 76.7±4.9 | 289.5±27.6 | 76.9±1.8 | 333.6±18.7 |
| GES | 116.0±15.0 | 224.4±41.7 | 128.9±11.2 | 232.2±22.8 | 113.8±5.8 | 308.3±20.2 | 109.9±10.3 | 212.1±32.9 | 129.2±13.7 | 242.5±39.6 | 119.0±14.4 | 225.4±31.6 | 116.0±15.0 | 224.4±41.7 | 106.4±9.6 | 240.1±60.1 | **74.8±2.2** | 316.9±18.9 |
| DirectLiNGAM | 88.9±10.6 | 264.7±34.4 | 88.0±5.2 | 265.2±39.4 | 81.1±7.9 | 258.4±23.0 | 87.4±7.0 | 219.9±32.4 | 91.2±9.2 | 247.2±20.8 | 89.3±8.2 | 266.4±35.2 | 104.2±10.5 | 341.8±13.1 | 89.6±10.2 | 248.5±33.6 | 79.7±0.5 | 329.3±20.3 |
| Var-SortnRegress | 62.6±14.8 | 72.7±38.0 | 90.6±9.9 | **55.9±22.4** | 95.0±9.3 | **105.5±37.0** | 99.0±4.5 | **80.9±30.9** | **46.5±20.2** | 148.5±42.7 | 76.6±15.4 | 82.4±46.6 | 139.9±10.2 | 263.7±17.6 | 58.2±7.7 | 63.2±31.7 | 75.5±4.1 | **282.5±27.7** |
| R²-SortnRegress | 95.6±14.4 | 138.3±31.7 | 105.1±6.9 | 116.5±15.4 | 114.4±7.9 | 217.8±27.6 | 114.5±9.2 | 183.9±37.2 | 116.5±9.8 | 157.1±30.1 | 120.0±16.3 | 199.3±33.8 | 95.6±14.4 | **138.3±31.7** | 91.4±18.3 | 141.5±38.5 | 83.7±5.7 | 342.2±16.2 |
| NOTEARS | 18.6±10.1 | 49.8±22.3 | 39.2±10.9 | 138.0±16.0 | **70.8±8.3** | 148.8±26.8 | 79.4±7.9 | 220.8±24.5 | 48.8±7.7 | 144.1±30.5 | 21.4±6.9 | 59.6±25.3 | 79.1±1.4 | **319.0±17.7** | 19.5±9.7 | 61.8±44.1 | **77.7±1.3** | 301.2±21.8 |
| GOLEM | 15.2±6.8 | 35.7±19.8 | 37.3±8.1 | 136.4±15.9 | 79.6±4.4 | 299.9±17.9 | 78.4±2.3 | 310.9±17.2 | 59.3±8.6 | 229.7±35.8 | 19.2±9.5 | 49.3±35.5 | 78.6±2.5 | 353.4±13.0 | **11.3±5.2** | **29.9±24.4** | 78.7±1.0 | 300.6±10.6 |
| NoCurl | 19.7±15.3 | 55.0±28.7 | 61.9±11.7 | **91.3±36.5** | 71.7±8.5 | **148.0±30.9** | 90.5±5.8 | **110.2±56.5** | 60.5±10.5 | **79.2±45.4** | 27.1±12.6 | 76.7±43.8 | 130.2±7.9 | 343.4±12.5 | 21.4±13.9 | 51.0±30.0 | 79.0±1.3 | 304.0±14.8 |
| DAGMA | **13.2±5.9** | **34.4±23.4** | **35.4±7.2** | 132.7±24.8 | 71.0±5.7 | 183.6±31.2 | **75.5±6.6** | 248.6±37.7 | 47.5±6.9 | 134.6±40.8 | **18.4±5.7** | **42.1±22.5** | **78.2±1.7** | 320.1±13.7 | 13.6±7.3 | 36.4±25.3 | **77.7±1.2** | **300.0±15.8** |

| 50 nodes | SHD↓ | SID↓ | SHD↓ | SID↓ | SHD↓ | SID↓ | SHD↓ | SID↓ | SHD↓ | SID↓ | SHD↓ | SID↓ | SHD↓ | SID↓ | SHD↓ | SID↓ | SHD↓ | SID↓ |
|---|---|---|---|---|---|---|---|---|---|---|---|---|---|---|---|---|---|---|
| Random | 661.5±18.5 | 1968.7±40.1 | 656.2±14.8 | 1965.0±54.8 | 645.3±21.6 | 1939.8±67.2 | 657.2±16.7 | 1963.2±30.2 | 656.9±14.3 | 1965.8±40.5 | 664.5±11.2 | 1967.2±69.6 | 657.0±18.2 | 1950.1±51.8 | 657.5±21.8 | 1974.7±53.5 | 657.7±14.1 | 1967.6±55.0 |
| DirectLiNGAM | 356.5±26.4 | 1630.5±89.3 | 434.0±27.6 | 1600.7±136.2 | 256.4±10.7 | 1745.2±123.7 | 423.4±35.3 | 1367.6±119.5 | 368.8±23.9 | 1505.5±148.5 | 361.5±33.3 | 1600.5±152.7 | 346.9±38.6 | 2131.9±78.5 | 351.7±26.8 | 1585.9±139.2 | 199.8±0.4 | 1986.5±96.8 |
| Var-SortnRegress | 384.1±9.5 | 430.9±156.4 | 864.7±11.9 | **353.8±88.1** | 552.1±25.1 | **828.4±136.4** | 785.6±37.6 | **537.0±149.1** | 773.9±44.8 | **428.7±177.1** | 427.2±39.1 | 519.2±141.0 | 984.0±30.7 | 1574.6±91.4 | 384.0±60.5 | 382.8±115.0 | **190.1±6.3** | **1773.6±84.9** |
| R²-SortnRegress | 700.4±10.0 | 978.1±251.5 | 898.5±19.0 | 794.7±156.3 | 628.3±32.5 | 1570.2±94.8 | 859.3±31.1 | 1080.3±205.7 | 914.0±45.1 | 1071.2±191.7 | 773.3±95.5 | 1130.1±187.6 | 700.6±10.3 | **978.2±251.6** | 727.1±18.5 | 996.6±170.6 | 207.0±6.3 | 2127.9±115.9 |
| NOTEARS | 70.1±30.8 | 528.0±163.3 | 89.6±37.0 | 1112.2±262.5 | 220.4±19.4 | 1843.9±101.8 | 281.1±30.2 | 1739.4±136.0 | 157.4±26.2 | 907.3±151.8 | 75.0±27.7 | 579.9±254.7 | 195.2±3.7 | 1936.4±81.0 | 74.2±31.0 | 532.6±150.4 | **194.7±1.6** | 1886.2±108.3 |
| GOLEM | 56.8±14.9 | **216.6±101.6** | 84.2±32.5 | 1237.6±224.7 | **199.0±1.3** | 1965.6±101.9 | **199.6±1.6** | 1966.5±103.9 | 170.0±25.2 | 1494.4±131.8 | 60.4±11.4 | 748.9±149.4 | **187.4±5.8** | 2041.9±135.1 | 53.4±19.9 | **205.1±170.9** | 197.1±2.2 | 1889.8±89.1 |
| NoCurl | 156.2±81.5 | 614.9±400.6 | 462.2±33.7 | **873.4±192.2** | 334.2±43.2 | **1199.7±104.8** | 477.9±26.3 | **1153.5±396.1** | 396.7±54.8 | 876.4±246.6 | 169.6±92.2 | 653.5±346.3 | 614.2±43.2 | 2158.5±54.1 | 166.6±67.0 | 586.2±206.4 | 196.6±3.1 | **1886.1±83.9** |
| DAGMA | **49.4±16.6** | 411.1±168.5 | **77.5±19.4** | 1107.3±162.0 | 211.8±12.0 | 1900.0±92.9 | 224.5±25.2 | 1821.2±88.3 | **155.0±30.2** | **870.3±124.2** | **58.3±15.6** | **504.8±161.9** | 194.2±3.9 | **1917.3±77.3** | **52.6±19.1** | 401.5±173.0 | **194.7±1.7** | 1886.8±104.9 |

Table 13: Nonlinear Setting, for ER-4 graphs of 10, 20, 50 nodes.

| 10 nodes | Vanilla model SHD↓ | SID↓ | Latent confounders SHD↓ | SID↓ | Measurement error SHD↓ | SID↓ | Autoregressive SHD↓ | SID↓ | Heterogeneous SHD↓ | SID↓ | Unfaithful SHD↓ | SID↓ | Scale-variant SHD↓ | SID↓ | Missing SHD↓ | SID↓ | Mechanism violation SHD↓ | SID↓ |
|---|---|---|---|---|---|---|---|---|---|---|---|---|---|---|---|---|---|---|
| Random | 31.1±3.4 | 74.2±6.7 | 33.2±2.4 | 76.2±3.8 | 33.2±2.9 | 76.6±3.7 | 31.0±3.5 | 75.7±5.4 | 33.7±3.3 | 77.7±4.1 | 31.9±2.0 | 76.0±5.8 | 30.5±2.7 | 74.2±4.1 | 32.0±3.2 | 73.9±5.4 | 32.0±3.2 | 74.8±6.2 |
| PC | 36.7±1.7 | 81.6±5.2 | 36.4±2.2 | 80.5±4.7 | 36.1±1.4 | 77.7±10.5 | 37.6±2.5 | 78.7±6.5 | 32.7±1.8 | 74.0±6.0 | 28.7±2.9 | 67.9±7.8 | 36.7±1.7 | 81.6±5.2 | 35.5±2.4 | 78.6±7.3 | 28.6±3.2 | 65.0±7.2 |
| GES | 37.3±1.0 | 77.3±11.8 | 36.9±1.3 | 74.3±9.2 | 38.1±1.4 | 70.4±8.0 | 38.6±1.1 | 77.4±7.0 | 32.8±1.7 | 71.1±6.6 | 24.2±2.5 | 60.8±7.2 | 37.3±1.0 | 77.3±11.8 | 36.6±1.8 | 72.2±9.0 | 27.9±2.7 | 65.0±10.6 |
| CAM | 23.1±3.4 | 40.5±9.3 | **28.8±2.7** | **55.4±12.1** | 31.3±2.3 | 62.4±8.5 | 29.8±2.1 | 57.4±7.6 | **15.3±3.8** | **33.3±12.7** | **13.9±4.3** | 37.4±10.2 | 23.1±3.4 | 40.5±9.3 | 23.8±2.8 | **36.1±7.2** | 30.0±2.1 | 75.3±5.7 |
| NOTEARS-MLP | 34.8±1.9 | 63.7±8.1 | 37.6±1.6 | 70.1±9.4 | 35.9±1.4 | **65.5±5.5** | 36.3±2.7 | 66.7±7.9 | 28.9±4.0 | 58.4±9.1 | 24.5±3.6 | 59.7±8.5 | 37.5±0.8 | 75.1±9.1 | 34.8±1.9 | 64.9±10.7 | 27.0±3.2 | 66.9±13.0 |
| GraN-DAG | **21.6±8.3** | **38.0±16.2** | 29.3±5.3 | 66.2±11.3 | 29.5±4.1 | 68.4±6.7 | **28.5±5.6** | **63.9±12.2** | **20.2±7.5** | 46.4±15.3 | 20.6±3.0 | 52.9±11.0 | **24.2±3.5** | **51.2±11.7** | **24.9±4.5** | 54.3±13.8 | 29.9±4.6 | 70.7±4.8 |
| DAGMA | 36.5±1.8 | 69.8±10.0 | 39.0±1.3 | 79.8±9.9 | 38.5±0.9 | 75.7±6.8 | 38.1±1.6 | 73.8±9.0 | 27.5±3.1 | 58.5±7.0 | **14.7±3.5** | **37.4±10.6** | 37.4±1.3 | 76.3±7.6 | 35.7±2.2 | 67.4±11.1 | **23.8±8.2** | **56.0±13.0** |

| 20 nodes | SHD↓ | SID↓ | SHD↓ | SID↓ | SHD↓ | SID↓ | SHD↓ | SID↓ | SHD↓ | SID↓ | SHD↓ | SID↓ | SHD↓ | SID↓ | SHD↓ | SID↓ | SHD↓ | SID↓ |
|---|---|---|---|---|---|---|---|---|---|---|---|---|---|---|---|---|---|---|
| Random | 118.1±6.5 | 312.4±25.3 | 112.2±7.2 | 300.7±16.4 | 116.4±7.1 | 322.0±13.7 | 116.4±5.7 | 323.9±12.7 | 115.6±5.3 | 308.7±13.4 | 115.1±7.0 | 322.9±8.9 | 117.6±6.1 | 321.6±12.4 | 114.8±5.6 | 325.6±32.9 | 118.0±4.5 | 317.1±14.5 |
| PC | 76.9±1.8 | 333.6±18.7 | 78.5±1.6 | 311.2±17.2 | 78.1±1.4 | 327.1±19.6 | 79.5±2.0 | 321.2±18.7 | 85.0±5.1 | 311.9±15.0 | 65.9±4.7 | 309.0±18.7 | 76.9±1.8 | 333.6±32.9 | 75.8±3.0 | 323.6±32.9 | 97.8±12.2 | **243.7±41.9** |
| GES | 74.8±2.2 | 316.9±18.9 | 78.0±1.9 | 306.5±16.5 | 76.4±1.6 | 304.1±28.4 | 77.5±2.0 | 320.0±16.4 | 83.5±8.2 | 303.3±22.7 | 65.2±5.9 | 276.6±26.7 | 74.8±2.2 | 316.9±18.9 | 75.8±3.0 | 310.4±22.5 | 85.3±7.1 | 304.4±22.9 |
| CAM | **57.6±3.9** | **199.6±28.8** | **74.2±3.3** | **275.6±26.1** | 73.7±3.3 | 285.1±28.9 | **72.1±4.8** | **272.6±39.5** | **63.2±3.9** | **206.7±24.5** | 36.4±7.2 | **190.5±35.0** | **57.6±3.9** | **199.6±28.8** | **57.1±4.3** | **193.4±31.1** | 85.3±7.1 | 304.4±22.9 |
| NOTEARS-MLP | **71.1±2.3** | **267.1±22.9** | 78.0±1.5 | 302.5±14.7 | **74.4±2.5** | **281.2±23.6** | 79.9±0.3 | 319.5±15.9 | 76.7±4.0 | **277.3±20.6** | 52.5±4.2 | 273.3±16.9 | **74.5±2.5** | 287.8±29.4 | **72.0±2.3** | 269.5±18.5 | **57.3±9.3** | **252.8±52.4** |
| GraN-DAG | 74.7±2.7 | 272.5±27.1 | 84.1±7.5 | 320.5±10.4 | 80.3±3.8 | 302.3±14.8 | **78.0±2.4** | **308.1±27.2** | **76.4±1.9** | 282.9±25.1 | 69.0±5.8 | 291.0±20.9 | 74.9±2.8 | **276.7±28.2** | 73.1±4.1 | **269.1±23.4** | 73.6±5.2 | 298.3±23.3 |
| DAGMA | 74.0±3.4 | 292.4±26.9 | **75.3±2.2** | **288.6±15.3** | 77.3±2.3 | 302.9±19.8 | 79.8±0.9 | 317.5±16.6 | 78.2±4.3 | 288.1±21.6 | **42.1±10.7** | **205.4±49.3** | **74.1±2.4** | 302.8±21.1 | 73.2±2.2 | 287.1±28.8 | 63.9±13.8 | 271.9±51.5 |

| 50 nodes | SHD↓ | SID↓ | SHD↓ | SID↓ | SHD↓ | SID↓ | SHD↓ | SID↓ | SHD↓ | SID↓ | SHD↓ | SID↓ | SHD↓ | SID↓ | SHD↓ | SID↓ | SHD↓ | SID↓ |
|---|---|---|---|---|---|---|---|---|---|---|---|---|---|---|---|---|---|---|
| Random | 672.0±20.1 | 1957.0±44.7 | 657.7±14.7 | 1983.2±47.1 | 665.7±18.8 | 1989.5±48.5 | 664.9±13.8 | 1993.2±78.8 | 664.9±18.8 | 1965.4±63.8 | 665.1±16.0 | 1945.5±57.7 | 671.8±18.0 | 1965.4±48.3 | 656.7±11.7 | 1978.2±46.1 | 654.0±24.0 | 1978.4±42.0 |
| CAM | **148.4±6.0** | **1345.8±170.1** | **167.9±3.8** | **1635.7±137.4** | **185.1±4.1** | **1766.6±129.1** | **193.6±17.8** | **1663.8±149.3** | 195.7±9.9 | **1347.6±131.4** | **113.7±6.0** | **1498.7±98.1** | **148.4±6.0** | **1345.8±170.1** | **150.6±5.7** | **1401.5±156.1** | 222.5±15.2 | 1971.5±155.6 |
| NOTEARS-MLP | **180.2±4.2** | **1653.8±135.1** | 195.7±5.3 | 1845.3±143.5 | 193.3±2.2 | **1848.3±83.8** | 203.2±7.1 | 1966.5±96.4 | **194.2±7.1** | 1776.0±58.0 | 149.5±8.1 | 1819.5±123.7 | **183.0±3.8** | 1770.2±119.1 | **182.0±4.6** | **1698.7±124.7** | **150.9±18.0** | 1734.5±143.1 |
| GraN-DAG | 193.4±2.8 | 1831.1±112.1 | 197.8±5.6 | 1935.7±14.9 | 200.8±0.6 | 1968.7±108.1 | **196.7±1.3** | **1876.4±107.3** | 196.0±2.0 | 1881.1±85.0 | 180.8±7.5 | 1844.7±107.9 | 189.8±5.1 | **1714.2±185.7** | 195.4±2.2 | 1860.0±106.8 | 190.3±7.4 | 1892.5±103.8 |
| DAGMA | 180.8±3.8 | 1711.2±107.1 | **186.7±4.9** | **1826.1±103.7** | 192.5±2.6 | 1867.5±97.7 | 200.0±0.0 | 1969.8±103.7 | 199.6±2.1 | 1925.7±91.9 | **134.2±5.6** | **1678.6±113.7** | 186.6±4.4 | 1879.6±150.0 | 183.3±3.2 | 1776.5±126.9 | 159.5±32.7 | **1674.3±285.6** |

Table 14: Linear Setting, for SF-2 graphs of 10, 20, 50 nodes.

| 10 nodes | Vanilla model SHD↓ | SID↓ | Latent confounders SHD↓ | SID↓ | Measurement error SHD↓ | SID↓ | Autoregressive SHD↓ | SID↓ | Heterogeneous SHD↓ | SID↓ | Unfaithful SHD↓ | SID↓ | Scale-variant SHD↓ | SID↓ | Missing SHD↓ | SID↓ | Mechanism violation SHD↓ | SID↓ |
|---|---|---|---|---|---|---|---|---|---|---|---|---|---|---|---|---|---|---|
| Random | 24.8±2.3 | 65.6±9.2 | 26.5±2.4 | 66.1±8.9 | 28.8±2.7 | 70.6±3.7 | 26.9±2.6 | 67.8±8.3 | 27.5±1.3 | 66.6±7.0 | 26.0±4.9 | 63.7±17.6 | 27.8±3.1 | 72.4±5.4 | 26.6±4.2 | 66.3±9.0 | 29.1±3.9 | 71.2±5.3 |
| PC | 15.0±4.2 | 71.9±6.0 | 19.3±2.3 | 69.9±8.9 | 23.6±2.9 | 59.7±14.8 | 19.5±3.4 | 69.3±12.1 | 18.7±5.6 | 74.2±9.6 | 16.0±4.5 | 61.2±23.3 | 21.0±2.4 | 71.9±6.0 | 15.3±4.8 | 71.1±8.4 | 20.1±3.0 | 80.8±8.5 |
| GES | 7.7±6.3 | 32.3±18.4 | 25.1±6.7 | 49.4±9.5 | 24.2±3.4 | 52.6±18.8 | 18.0±4.7 | 44.8±18.7 | 18.8±7.2 | 50.7±13.2 | 8.7±5.0 | 35.0±14.2 | **7.7±6.3** | **32.3±18.4** | 12.5±8.1 | 45.9±19.9 | 16.5±4.0 | 73.3±6.6 |
| DirectLiNGAM | 14.7±4.1 | 61.2±13.2 | 21.7±4.6 | 56.9±15.2 | 16.4±3.0 | 62.8±11.7 | 16.4±3.0 | 51.2±16.9 | 17.0±3.6 | 60.3±9.8 | 14.5±6.9 | 54.2±17.6 | 15.8±3.8 | 72.8±5.5 | 16.0±4.3 | 60.7±12.2 | 16.9±0.3 | 81.4±3.7 |
| Var-SortnRegress | 9.5±3.8 | 15.5±12.1 | 19.2±5.7 | 21.8±11.1 | 25.9±2.8 | **16.4±11.9** | 23.5±3.4 | **16.7±11.8** | 20.8±5.6 | **38.7±11.6** | 12.1±4.6 | 22.3±13.5 | 21.9±2.3 | 65.5±8.0 | 7.0±2.6 | 10.2±8.4 | 20.8±5.1 | **41.2±12.0** |
| R²-SortnRegress | 28.4±8.7 | 58.1±8.8 | 33.9±3.9 | 59.0±13.8 | 35.4±2.6 | 60.3±6.9 | 36.4±2.6 | 59.6±8.6 | 33.6±5.2 | 58.8±8.6 | 31.6±5.7 | 62.4±6.8 | 28.4±8.7 | 58.1±8.8 | 28.4±6.6 | 58.0±5.5 | 27.4±3.7 | 77.8±4.5 |
| NOTEARS | 1.3±1.2 | 9.3±16.5 | **9.4±3.7** | **12.5±9.2** | **11.1±3.6** | **27.9±14.6** | **14.5±4.1** | 17.6±13.2 | 3.2±1.6 | 8.0±8.4 | 1.6±1.6 | 6.4±6.9 | 13.4±3.8 | 68.2±12.8 | 2.1±1.7 | 14.4±9.9 | 14.4±1.5 | 73.3±4.7 |
| GOLEM | **0.2±0.4** | **0.0±0.0** | 10.7±2.7 | 39.1±10.7 | 16.0±4.1 | 54.8±11.7 | 17.0±0.0 | 61.0±0.0 | 2.6±1.7 | 6.0±5.2 | 1.6±1.0 | 10.0±9.2 | 12.5±3.9 | 64.5±17.0 | 1.6±1.8 | 10.0±9.7 | 13.1±2.2 | **67.9±8.0** |
| NoCurl | 1.1±1.4 | 7.0±9.0 | 12.2±3.5 | 14.8±12.7 | 11.8±3.4 | 28.3±14.4 | 15.0±5.4 | 22.1±15.6 | 3.4±2.2 | 8.2±8.5 | 1.8±1.6 | 10.0±9.2 | 12.5±3.9 | 64.5±17.0 | 1.6±1.8 | 10.2±9.7 | 13.8±1.2 | 72.3±4.5 |
| DAGMA | 0.8±1.0 | 4.9±6.4 | 10.1±2.5 | 15.8±8.8 | 11.3±3.4 | 29.7±15.7 | 14.8±4.6 | **17.2±12.2** | **2.5±2.0** | 5.6±6.5 | **1.3±1.6** | **4.1±5.3** | **10.5±4.1** | **56.5±15.4** | 0.9±1.5 | 3.2±4.4 | 14.4±1.5 | 73.4±4.7 |

| 20 nodes | SHD↓ | SID↓ | SHD↓ | SID↓ | SHD↓ | SID↓ | SHD↓ | SID↓ | SHD↓ | SID↓ | SHD↓ | SID↓ | SHD↓ | SID↓ | SHD↓ | SID↓ | SHD↓ | SID↓ |
|---|---|---|---|---|---|---|---|---|---|---|---|---|---|---|---|---|---|---|
| Random | 106.8±6.4 | 278.5±34.4 | 101.2±4.9 | 274.8±28.2 | 103.4±4.9 | 278.1±22.4 | 106.0±8.7 | 281.5±15.9 | 105.3±5.3 | 286.4±11.7 | 107.9±7.1 | 253.0±35.8 | 104.5±4.1 | 276.5±20.0 | 105.6±6.7 | 263.2±28.4 | 104.4±9.0 | 270.1±16.1 |
| PC | 45.0±4.9 | 314.7±31.1 | 48.7±5.1 | 318.7±27.8 | 68.8±4.5 | 305.4±18.7 | 53.4±6.1 | 327.0±21.9 | 45.8±4.4 | 322.7±21.4 | 40.4±5.7 | 303.3±24.9 | 45.0±4.9 | 314.7±31.1 | 44.1±9.0 | 316.7±27.7 | 43.3±5.6 | 346.5±16.2 |
| GES | 23.5±13.8 | 132.0±53.7 | 131.9±9.3 | 180.3±27.2 | 97.4±8.3 | 235.9±39.5 | 92.9±10.7 | 185.8±45.4 | 38.0±4.1 | 154.0±31.5 | 23.5±13.8 | 132.0±53.7 | 24.8±22.6 | 132.0±53.7 | 24.8±22.6 | 132.0±53.7 | 37.8±6.2 | 292.9±33.8 |
| DirectLiNGAM | 41.2±7.4 | 254.8±51.5 | 95.3±6.6 | 218.2±53.6 | 37.1±4.3 | 319.0±27.3 | 51.5±8.6 | 273.6±38.1 | 41.7±6.2 | 268.7±40.5 | 35.8±6.9 | 249.3±37.2 | 42.7±7.6 | 298.4±46.6 | 39.5±6.3 | 273.1±36.4 | 39.4±6.9 | 364.6±11.0 |
| Var-SortnRegress | 44.5±9.7 | 53.0±24.8 | 124.0±11.7 | **57.3±34.0** | 122.7±10.1 | **74.6±29.7** | 115.0±10.4 | **72.8±32.3** | 94.4±15.5 | 55.2±22.6 | 40.8±9.5 | 94.2±17.3 | 103.3±17.0 | 212.5±33.1 | 35.2±5.7 | 40.3±27.1 | 62.3±15.4 | **248.0±23.6** |
| R²-SortnRegress | 129.3±9.7 | 245.0±14.4 | 153.4±5.6 | 222.0±27.9 | 151.3±12.2 | 230.9±23.2 | 153.0±9.6 | 229.8±32.5 | 147.3±10.2 | 245.0±11.4 | 135.6±24.1 | 241.9±18.7 | 129.3±9.7 | 245.0±14.4 | 129.1±28.3 | 243.4±20.0 | 81.9±14.7 | 339.2±13.2 |
| NOTEARS | 5.6±2.8 | 17.5±10.3 | 26.7±7.3 | 174.1±49.7 | **28.7±3.5** | 227.7±41.9 | 48.7±8.1 | 332.9±13.4 | 11.6±6.8 | 53.3±28.7 | **3.2±1.6** | **16.4±16.7** | 32.7±3.0 | 292.9±28.3 | 5.3±2.4 | 43.5±38.7 | 32.0±1.3 | 338.1±12.7 |
| GOLEM | 4.7±5.0 | 37.4±36.1 | 29.3±8.4 | 213.5±51.4 | 36.6±6.0 | 358.6±5.6 | **35.2±1.7** | 361.0±0.0 | 11.5±6.0 | 75.4±45.0 | 5.2±5.4 | 45.1±48.5 | **28.5±2.1** | 335.8±33.8 | 2.3±2.0 | 18.1±18.8 | **30.3±1.5** | **331.7±18.9** |
| NoCurl | 5.3±3.3 | 34.8±26.8 | 55.9±18.1 | **70.0±34.6** | 33.4±5.7 | 202.7±60.7 | 66.5±13.0 | **86.4±43.6** | 16.0±5.9 | 44.5±25.9 | 5.0±2.6 | 23.1±24.3 | 32.9±9.4 | **248.4±34.8** | 4.0±3.0 | 30.5±21.3 | 31.5±1.7 | 336.7±14.6 |
| DAGMA | **2.9±1.6** | **18.2±18.9** | 27.5±7.3 | 181.7±52.9 | 29.1±3.3 | 235.9±48.2 | 42.5±3.8 | 339.2±10.5 | **9.5±5.6** | **35.3±23.5** | 3.4±2.0 | 17.7±16.0 | 29.8±7.0 | 259.5±40.5 | **2.0±1.7** | **9.9±22.0** | 32.0±1.1 | 338.4±12.5 |

| 50 nodes | SHD↓ | SID↓ | SHD↓ | SID↓ | SHD↓ | SID↓ | SHD↓ | SID↓ | SHD↓ | SID↓ | SHD↓ | SID↓ | SHD↓ | SID↓ | SHD↓ | SID↓ | SHD↓ | SID↓ |
|---|---|---|---|---|---|---|---|---|---|---|---|---|---|---|---|---|---|---|
| Random | 637.3±14.4 | 1682.5±98.4 | 640.1±8.9 | 1712.6±83.9 | 641.4±21.2 | 1693.8±145.4 | 636.9±14.1 | 1681.7±143.3 | 638.6±15.6 | 1734.1±71.6 | 637.0±15.9 | 1691.0±111.0 | 643.1±23.0 | 1705.8±60.6 | 637.0±14.5 | 1721.4±64.4 | 634.1±16.8 | **1657.4±157.2** |
| DirectLiNGAM | 108.8±15.2 | 1924.0±160.3 | 254.9±23.3 | 1710.4±29.3 | 113.7±8.9 | 2283.7±46.5 | 176.5±23.6 | 1720.2±282.3 | 121.3±14.9 | 1835.9±220.6 | 103.4±17.3 | 1852.6±109.3 | 110.9±11.8 | 2146.6±79.7 | 108.7±14.6 | 1911.6±177.7 | 95.7±1.3 | 2399.6±22.0 |
| Var-SortnRegress | 201.2±26.9 | 351.6±121.6 | 938.9±19.6 | 468.6±128.9 | 675.7±79.3 | **484.0±93.0** | 689.3±49.9 | **480.0±95.4** | 593.7±48.1 | 344.6±113.1 | 239.7±73.3 | 588.0±245.9 | 471.9±77.6 | **1263.1±259.6** | 206.5±39.7 | 447.9±118.6 | 209.4±45.8 | 1742.4±217.7 |
| R²-SortnRegress | 689.3±59.4 | 1668.3±35.3 | 1009.7±11.9 | 1428.9±61.0 | 777.9±44.2 | 1600.4±133.8 | 858.2±58.5 | 1462.0±103.5 | 851.3±48.8 | 1595.1±39.3 | 733.8±36.6 | 1670.0±114.2 | 689.3±59.4 | **1668.3±35.3** | 689.1±53.8 | 1611.9±35.9 | 267.6±63.6 | 2284.8±58.9 |
| NOTEARS | 25.0±4.2 | 519.2±134.6 | 63.1±8.9 | 1632.1±197.7 | **94.0±7.5** | 2235.5±60.5 | 151.3±12.2 | 230.9±23.2 | 49.1±9.1 | 800.4±224.6 | 24.1±10.3 | 451.1±241.5 | 74.4±3.6 | 1987.5±127.3 | 28.4±4.4 | 636.9±150.9 | 85.3±3.7 | 2327.3±55.6 |
| GOLEM | 17.6±6.3 | 461.5±202.3 | 74.2±9.6 | 1973.6±186.2 | 96.6±1.2 | 2395.5±14.0 | **97.4±1.1** | 2399.8±0.9 | 48.3±10.8 | 1072.2±200.0 | 11.5±4.0 | 229.1±107.8 | **65.5±7.8** | 2079.6±205.1 | 16.7±5.7 | 466.8±201.5 | **83.7±2.6** | **2319.6±63.9** |
| NoCurl | 15.0±7.1 | **130.6±78.1** | 318.3±25.3 | **395.4±167.4** | 123.2±25.6 | **1735.2±136.5** | 301.6±41.5 | **597.7±185.6** | 73.9±17.8 | **271.5±70.4** | 15.3±9.9 | **99.2±62.0** | 126.7±20.5 | **1660.5±191.6** | 18.9±8.6 | **115.2±63.5** | 80.0±3.4 | 2339.7±46.4 |
| DAGMA | **12.3±4.3** | 162.4±201.2 | **62.0±8.1** | 1827.6±174.6 | 95.9±4.8 | 2375.6±23.3 | 126.1±12.7 | 2293.9±74.5 | **42.1±10.3** | 561.0±177.6 | **9.1±4.4** | 102.7±42.5 | 73.4±8.0 | 1962.4±164.4 | **12.9±5.9** | 145.1±174.8 | 85.0±3.1 | 2326.5±57.5 |

Table 15: Nonlinear Setting, for SF-2 graphs of 10, 20, 50 nodes.

| 10 nodes | Vanilla model SHD↓ | SID↓ | Latent confounders SHD↓ | SID↓ | Measurement error SHD↓ | SID↓ | Autoregressive SHD↓ | SID↓ | Heterogeneous SHD↓ | SID↓ | Unfaithful SHD↓ | SID↓ | Scale-variant SHD↓ | SID↓ | Missing SHD↓ | SID↓ | Mechanism violation SHD↓ | SID↓ |
|---|---|---|---|---|---|---|---|---|---|---|---|---|---|---|---|---|---|---|
| Random | 26.3±3.9 | 66.6±8.7 | 25.3±2.5 | 66.6±7.6 | 27.8±3.0 | 70.2±4.4 | 27.7±2.1 | 66.7±6.3 | 26.1±2.8 | 65.8±7.3 | 27.4±1.8 | 66.4±10.8 | 27.3±2.6 | 68.1±10.2 | 27.0±3.0 | 71.0±4.4 | 27.7±2.3 | 61.5±10.6 |
| PC | 20.1±3.0 | 80.8±8.5 | 16.4±1.4 | 85.7±4.1 | 16.5±2.4 | 79.0±8.4 | 17.7±1.9 | 78.3±9.3 | 22.9±2.3 | 71.1±8.0 | 17.1±3.1 | 74.1±6.1 | 20.1±3.0 | 80.8±8.5 | 19.2±2.3 | 77.3±7.8 | 15.0±4.2 | 71.9±6.0 |
| GES | 16.5±4.0 | 73.3±6.6 | 13.3±1.3 | 72.4±11.0 | 15.1±2.9 | 78.6±8.7 | 15.5±3.2 | 74.4±8.3 | 22.4±4.1 | 60.9±8.5 | 14.9±1.6 | 50.8±16.6 | 16.5±4.0 | 73.3±6.6 | 16.2±3.7 | 60.2±10.0 | 7.7±6.3 | 32.3±18.4 |
| CAM | **2.1±1.6** | **0.7±2.1** | **7.6±6.4** | **9.5±2.5** | **4.3±3.16.9** | **3.9±2.1** | **13.9±10.6** | 15.3±2.3 | **6.6±6.2** | **3.7±1.2** | **22.3±7.1** | 2.1±1.6 | **0.7±2.1** | 2.7±2.4 | **1.0±3.0** | 14.7±3.8 | 59.2±10.5 |
| NOTEARS-MLP | 8.9±4.2 | 41.7±15.5 | 12.3±2.4 | 62.1±10.5 | **11.4±2.7** | **52.7±15.4** | 12.2±2.4 | 52.7±11.8 | 16.4±3.1 | 49.9±7.2 | 9.0±1.8 | 58.5±11.0 | 10.8±3.9 | 49.5±12.2 | 8.9±1.9 | 40.0±13.0 | 5.3±2.3 | 27.0±7.9 |
| GraN-DAG | **3.0±3.9** | **11.2±13.4** | **9.7±2.0** | **48.8±11.3** | 13.5±2.0 | 64.5±17.9 | **9.5±4.3** | **45.8±23.3** | **5.1±2.0** | **31.9±15.1** | **7.4±2.2** | **47.7±12.3** | **1.9±1.9** | **8.9±12.3** | **1.4±1.7** | **7.3±10.2** | 9.3±2.7 | 53.1±14.4 |
| DAGMA | 10.2±4.5 | 45.2±18.2 | 14.2±1.7 | 76.2±10.4 | 13.2±1.8 | 68.0±9.9 | 14.1±2.2 | 59.0±10.0 | 16.6±5.0 | 42.5±8.5 | 8.2±2.7 | **46.2±15.7** | 13.5±2.5 | 66.8±8.3 | 8.9±1.6 | 32.7±13.4 | **2.5±3.1** | **14.6±21.1** |

| 20 nodes | SHD↓ | SID↓ | SHD↓ | SID↓ | SHD↓ | SID↓ | SHD↓ | SID↓ | SHD↓ | SID↓ | SHD↓ | SID↓ | SHD↓ | SID↓ | SHD↓ | SID↓ | SHD↓ | SID↓ |
|---|---|---|---|---|---|---|---|---|---|---|---|---|---|---|---|---|---|---|
| Random | 106.6±6.6 | 280.1±10.9 | 102.6±6.6 | 273.8±32.1 | 104.6±7.1 | **266.8±28.7** | 104.3±6.3 | 268.2±16.0 | 100.1±5.2 | 270.2±33.8 | 101.6±6.2 | 278.9±20.9 | 101.6±7.9 | 267.1±36.6 | 104.2±8.0 | 259.1±30.5 | 104.5±5.6 | 265.1±31.6 |
| PC | 43.3±5.6 | 346.5±16.2 | 34.7±1.7 | 360.6±16.0 | 38.8±4.2 | 360.1±14.9 | 40.3±4.1 | 361.9±10.0 | 63.2±4.8 | 327.3±12.3 | 38.2±5.9 | 336.2±25.0 | 43.3±5.6 | 346.5±16.2 | 44.6±5.6 | 336.8±19.1 | 45.0±4.9 | 314.7±31.1 |
| GES | 37.8±6.2 | 292.9±33.8 | 33.0±2.7 | 354.0±16.3 | 33.4±3.5 | 331.1±30.0 | 33.0±3.3 | 315.3±35.4 | 67.5±6.4 | 286.4±26.1 | 35.2±7.4 | 252.3±43.7 | 37.8±6.2 | 292.9±33.8 | 39.6±7.6 | 314.3±10.8 | 23.5±13.8 | 219.4±43.0 |
| CAM | **6.7±2.3** | **18.5±12.1** | 22.1±6.3 | **245.8±40.2** | 26.5±4.9 | **279.0±47.0** | **12.3±7.1** | **71.2±40.5** | 15.7±2.4 | 60.9±8.5 | 14.9±5.1 | 90.1±32.1 | **6.7±2.3** | **18.5±12.1** | **4.8±1.7** | **9.7±10.8** | 33.3±8.9 | 219.4±43.0 |
| NOTEARS-MLP | 22.6±3.3 | 214.8±43.7 | 31.9±2.9 | 336.4±21.4 | 29.2±2.5 | 291.0±28.2 | 37.7±2.1 | 369.9±0.3 | 37.0±0.0 | 361.0±0.0 | 20.5±2.8 | 256.7±26.9 | 26.0±1.8 | 264.5±43.4 | 18.8±3.1 | **184.5±40.6** | 13.3±2.8 | 123.8±23.6 |
| GraN-DAG | **19.1±13.9** | **91.7±78.7** | 38.3±6.3 | 345.0±14.4 | 31.4±9.8 | 346.3±29.8 | **30.3±3.7** | **300.7±55.2** | **26.5±4.2** | **291.3±37.0** | 22.8±2.5 | 289.9±31.9 | **16.4±16.0** | **16.7±66.4** | 21.0±7.7 | 210.7±67.0 | 26.1±7.3 | 286.9±17.5 |
| DAGMA | 23.0±3.8 | 198.2±42.9 | 34.1±4.2 | 347.2±18.2 | 29.6±1.7 | 328.8±26.2 | 37.0±0.8 | 360.1±1.2 | 37.0±0.0 | 361.0±0.0 | **20.0±5.5** | **217.4±36.6** | 26.1±3.0 | 240.6±47.0 | 20.8±5.2 | 189.5±41.1 | **8.2±3.2** | **88.1±47.2** |

| 50 nodes | SHD↓ | SID↓ | SHD↓ | SID↓ | SHD↓ | SID↓ | SHD↓ | SID↓ | SHD↓ | SID↓ | SHD↓ | SID↓ | SHD↓ | SID↓ | SHD↓ | SID↓ | SHD↓ | SID↓ |
|---|---|---|---|---|---|---|---|---|---|---|---|---|---|---|---|---|---|---|
| Random | 640.0±12.1 | 1727.7±57.4 | 632.7±16.0 | **1707.8±62.3** | 632.6±9.9 | **1674.7±104.8** | 629.2±13.1 | 1734.0±62.4 | 634.9±16.0 | 1716.2±71.8 | 637.7±21.5 | 1724.3±97.4 | 649.8±12.8 | 1690.1±136.5 | 638.5±18.3 | 1700.5±78.9 | 639.9±17.1 | 1693.9±148.1 |
| CAM | **15.2±4.2** | **189.7±60.8** | 152.6±12.5 | **1523.6±125.9** | **68.8±7.0** | **1807.7±217.3** | **54.2±12.2** | **579.4±107.5** | 151.2±19.3 | 1544.4±42.0 | 14.1±4.6 | **190.5±43.9** | **15.2±4.2** | **189.7±60.8** | 16.2±3.7 | 602.3±109.5 | 10.7±1.9 | 1390.4±194.5 |
| NOTEARS-MLP | **53.4±3.7** | 1399.8±201.1 | 93.5±7.3 | 1957.3±135.2 | 82.1±3.7 | **2227.7±96.4** | 97.6±0.8 | 2400.4±0.8 | 88.4±6.5 | 2019.2±120.2 | **54.0±7.7** | **1410.1±199.9** | 62.1±3.9 | 1820.7±151.9 | 58.6±5.5 | **1298.3±216.1** | 39.1±7.9 | 982.8±79.4 |
| GraN-DAG | 68.9±39.2 | 1572.7±310.6 | 89.3±4.2 | 1753.5±127.3 | 98.0±0.0 | 2399.8±0.0 | **89.7±19.7** | **2167.6±194.3** | 90.3±3.0 | 2356.4±38.5 | 77.1±5.6 | 2267.5±71.3 | **57.2±9.3** | **1635.1±316.4** | 62.1±21.4 | 1409.3±330.5 | 78.4±8.6 | 2167.7±185.2 |
| DAGMA | 53.7±6.0 | **1344.0±232.1** | 83.2±2.7 | 1649.3±121.7 | **77.6±4.0** | 2247.0±110.9 | 96.9±0.3 | 2400.9±0.0 | 95.2±1.7 | 2379.7±34.7 | 58.0±5.4 | 1562.1±204.7 | 64.6±6.3 | 1756.0±156.4 | 55.6±10.0 | 1407.4±173.7 | **29.2±5.2** | **865.0±132.9** |

Table 16: Linear Setting, for SF-4 graphs of 10, 20, 50 nodes.

| 10 nodes | Vanilla model | | Latent confounders | | Measurement error | | Autoregressive | | Heterogeneous | | Unfaithful | | Scale-variant | | Missing | | Mechanism violation | |
|---|---|---|---|---|---|---|---|---|---|---|---|---|---|---|---|---|---|---|
| | SHD↓ | SID↓ | SHD↓ | SID↓ | SHD↓ | SID↓ | SHD↓ | SID↓ | SHD↓ | SID↓ | SHD↓ | SID↓ | SHD↓ | SID↓ | SHD↓ | SID↓ | SHD↓ | SID↓ |
| Random | 28.7±2.8 | 77.2±4.4 | 30.0±3.8 | 71.4±9.8 | 30.6±3.8 | 74.1±7.6 | 29.3±3.8 | 73.2±6.1 | 28.9±3.1 | 76.4±3.6 | 31.1±3.7 | 78.6±3.8 | 30.8±3.0 | 73.1±6.1 | 30.0±4.6 | 75.2±5.5 | 27.0±3.8 | 76.0±8.7 |
| PC | 27.3±2.8 | 71.4±6.1 | 26.1±2.5 | 76.6±2.6 | 25.8±4.1 | 70.8±8.0 | 27.1±3.1 | 73.2±5.6 | 28.8±2.4 | 75.7±5.2 | 26.5±2.3 | 74.2±5.4 | 27.3±2.8 | 71.4±6.1 | 26.6±3.7 | 71.1±6.2 | 26.3±2.7 | 77.7±6.9 |
| GES | 23.0±5.1 | 62.7±9.8 | 27.2±6.6 | 64.3±11.3 | 24.5±3.7 | 60.0±14.5 | 22.8±5.5 | 53.5±14.0 | 27.4±6.6 | 64.7±13.8 | 24.6±4.5 | 67.5±6.5 | 23.0±5.1 | 62.7±9.8 | 26.1±4.6 | 67.0±6.7 | 26.1±2.2 | 82.6±5.0 |
| DirectLiNGAM | 21.6±5.0 | 63.8±11.6 | 24.9±2.8 | 65.1±6.5 | 26.1±2.2 | 67.8±4.3 | 21.4±4.1 | 61.6±7.9 | 23.0±4.2 | 58.6±8.0 | 19.8±4.7 | 61.7±4.5 | 24.9±5.5 | 74.1±9.4 | 22.3±4.9 | 65.0±8.5 | 30.0±0.0 | 81.9±2.7 |
| Var-SortnRegress | 10.7±2.7 | 22.5±6.9 | 13.0±4.7 | 21.4±10.5 | 17.8±1.9 | 27.6±6.0 | 15.7±3.6 | 18.6±13.6 | 15.5±3.3 | 20.1±7.1 | 13.2±5.2 | 24.3±10.8 | 26.2±4.4 | 61.8±6.4 | 9.9±4.0 | 17.2±12.9 | 20.6±4.5 | 57.0±9.7 |
| R²-SortnRegress | 25.3±5.0 | 60.2±8.5 | 24.5±4.6 | 51.1±11.7 | 27.9±3.3 | 60.2±7.2 | 23.5±4.5 | 49.2±11.5 | 25.8±5.4 | 57.4±11.6 | 26.5±5.9 | 57.3±8.4 | 25.3±5.0 | 60.2±8.5 | 21.5±4.7 | 47.7±15.1 | 29.9±1.5 | 83.9±5.6 |
| NOTEARS | 3.2±3.2 | 16.5±12.4 | 8.0±2.5 | 18.5±11.3 | 15.3±3.5 | 35.3±10.2 | 15.4±3.5 | 21.6±12.3 | 10.9±4.4 | 27.1±11.9 | 2.4±1.6 | 14.7±11.8 | 24.7±3.9 | 79.2±6.3 | 4.0±2.7 | 18.8±10.9 | 29.1±0.7 | 77.7±4.2 |
| GOLEM | 2.2±2.1 | 9.0±8.8 | 9.3±2.7 | 21.3±14.5 | 24.6±2.6 | 69.4±6.2 | 18.4±1.7 | 44.1±10.5 | 9.1±3.0 | 21.5±8.6 | 1.6±1.9 | 7.0±7.8 | 25.7±4.5 | 79.9±5.1 | 1.4±1.4 | 6.9±7.9 | 28.2±1.2 | 73.1±7.7 |
| NoCurl | 4.0±3.2 | 17.4±13.1 | 10.4±2.8 | 24.9±13.4 | 15.5±2.7 | 37.6±8.0 | 14.1±3.7 | 24.2±12.8 | 10.2±3.6 | 21.3±15.3 | 2.5±1.4 | 17.0±10.6 | 26.8±2.8 | 75.9±9.2 | 2.6±2.4 | 11.7±10.6 | 29.0±1.1 | 77.7±5.6 |
| DAGMA | 2.0±2.2 | 10.9±11.9 | 8.5±2.5 | 20.6±8.5 | 15.2±3.5 | 37.9±7.6 | 15.5±3.8 | 25.7±12.0 | 9.6±3.5 | 20.5±10.9 | 1.3±1.5 | 7.3±10.6 | 25.7±2.9 | 79.3±4.1 | 1.1±1.1 | 6.4±6.6 | 29.1±0.7 | 77.7±4.2 |
| **20 nodes** | SHD↓ | SID↓ | SHD↓ | SID↓ | SHD↓ | SID↓ | SHD↓ | SID↓ | SHD↓ | SID↓ | SHD↓ | SID↓ | SHD↓ | SID↓ | SHD↓ | SID↓ | SHD↓ | SID↓ |
| Random | 113.4±6.5 | 314.7±10.9 | 109.9±5.4 | 303.6±17.9 | 110.9±7.0 | 308.7±17.0 | 109.9±8.4 | 303.2±13.1 | 114.5±4.7 | 310.0±14.8 | 113.3±6.9 | 309.1±10.0 | 113.1±4.7 | 301.9±8.9 | 111.5±7.2 | 303.0±15.9 | 115.3±8.5 | 311.8±8.6 |
| PC | 77.5±2.8 | 345.1±10.6 | 78.9±6.4 | 349.5±7.9 | 92.6±5.1 | 329.8±15.4 | 82.8±3.5 | 341.9±7.3 | 78.9±4.4 | 347.3±6.1 | 76.2±4.7 | 347.6±10.9 | 77.5±2.8 | 345.1±10.6 | 78.1±4.2 | 350.5±10.8 | 67.6±2.7 | 371.1±7.9 |
| GES | 96.6±14.4 | 257.5±23.5 | 131.0±11.1 | 235.8±30.8 | 116.0±6.8 | 293.5±12.5 | 115.2±11.9 | 241.3±27.0 | 127.5±13.4 | 259.1±22.8 | 88.9±10.5 | 250.7±18.7 | 96.6±14.4 | 257.5±23.5 | 81.9±16.2 | 267.3±27.9 | 62.3±2.0 | 368.9±2.6 |
| DirectLiNGAM | 58.8±13.0 | 283.4±17.1 | 76.9±6.6 | 280.1±11.8 | 67.1±8.0 | 318.0±12.8 | 81.3±5.4 | 284.8±27.5 | 72.9±16.8 | 292.2±38.2 | 65.6±5.8 | 296.3±26.4 | 68.3±10.9 | 344.1±11.1 | 60.2±12.1 | 287.6±22.0 | 69.9±0.3 | 368.5±9.2 |
| Var-SortnRegress | 54.7±10.4 | 104.0±42.3 | 103.5±10.3 | 88.9±21.5 | 111.2±7.1 | 119.0±41.7 | 108.9±7.3 | 107.1±35.8 | 106.3±6.9 | 100.0±36.3 | 66.8±19.7 | 118.2±35.5 | 108.5±20.7 | 235.3±21.8 | 57.9±9.9 | 105.5±22.3 | 61.3±4.8 | 310.9±15.1 |
| R²-SortnRegress | 128.5±10.1 | 251.4±16.7 | 132.7±9.8 | 229.1±25.9 | 135.7±7.4 | 243.4±26.1 | 133.3±5.9 | 225.8±18.4 | 143.7±5.3 | 246.9±12.8 | 137.5±18.1 | 261.9±13.0 | 128.5±10.1 | 251.4±16.7 | 127.5±19.9 | 254.7±15.7 | 77.7±4.2 | 371.7±5.1 |
| NOTEARS | 12.5±8.8 | 76.2±62.8 | 41.1±5.9 | 109.1±26.4 | 55.7±8.8 | 247.7±18.3 | 84.5±16.2 | 185.3±40.5 | 36.0±9.7 | 160.3±45.1 | 10.3±6.6 | 68.6±36.2 | 68.2±5.5 | 355.1±9.5 | 10.5±6.8 | 87.8±41.0 | 68.8±0.6 | 349.0±9.2 |
| GOLEM | 8.5±5.4 | 83.9±50.1 | 52.3±7.4 | 136.2±27.6 | 69.8±0.4 | 360.5±0.4 | 70.5±0.8 | 360.5±0.8 | 40.1±8.1 | 190.1±51.8 | 10.1±5.7 | 82.3±49.2 | 65.1±6.2 | 351.8±10.3 | 7.9±4.0 | 72.7±28.9 | 67.9±0.9 | 346.3±8.8 |
| NoCurl | 10.7±6.6 | 64.9±36.7 | 55.0±8.5 | 125.3±38.3 | 55.6±9.5 | 236.5±20.8 | 89.3±12.6 | 160.3±38.6 | 43.3±10.8 | 120.6±37.5 | 14.5±11.9 | 76.5±60.2 | 72.0±10.7 | 336.5±20.6 | 9.7±6.1 | 63.1±40.9 | 68.8±0.7 | 350.8±9.6 |
| DAGMA | 8.3±6.2 | 58.1±30.5 | 41.8±6.5 | 111.2±24.3 | 55.6±7.0 | 270.6±15.4 | 77.3±11.9 | 207.9±70.5 | 35.9±9.7 | 137.0±43.1 | 7.2±3.7 | 38.9±20.6 | 65.6±4.3 | 345.2±20.0 | 7.5±5.3 | 52.5±40.6 | 68.8±0.6 | 349.0±9.2 |
| **50 nodes** | SHD↓ | SID↓ | SHD↓ | SID↓ | SHD↓ | SID↓ | SHD↓ | SID↓ | SHD↓ | SID↓ | SHD↓ | SID↓ | SHD↓ | SID↓ | SHD↓ | SID↓ | SHD↓ | SID↓ |
| Random | 659.9±20.4 | 1901.2±49.6 | 670.5±23.8 | 1899.4±20.8 | 658.5±18.7 | 1904.2±40.1 | 667.6±19.5 | 1908.0±47.7 | 668.3±14.9 | 1914.0±28.9 | 659.2±28.8 | 1918.7±22.9 | 664.4±18.1 | 1914.4±31.5 | 669.1±13.1 | 1911.2±28.4 | 648.4±21.4 | 1911.2±27.7 |
| DirectLiNGAM | 207.1±25.8 | 2126.2±120.4 | 338.7±22.7 | 1979.3±103.8 | 261.8±11.4 | 2323.8±19.1 | 317.5±37.7 | 1807.5±122.5 | 238.4±20.2 | 2095.8±84.5 | 210.6±23.5 | 2156.2±93.1 | 226.4±18.5 | 2347.1±17.9 | 206.5±23.0 | 2101.7±139.7 | 190.0±0.0 | 2405.9±14.7 |
| Var-SortnRegress | 347.1±52.9 | 622.1±80.6 | 880.7±16.3 | 694.0±106.9 | 695.6±39.7 | 1112.8±90.2 | 770.7±42.6 | 747.4±88.4 | 744.3±36.8 | 611.2±102.7 | 441.3±70.4 | 800.0±149.5 | 660.6±113.9 | 1515.4±95.1 | 352.9±37.6 | 617.3±108.6 | 170.8±7.3 | 2284.0±39.9 |
| R²-SortnRegress | 777.6±55.5 | 1635.7±56.4 | 968.6±11.7 | 1489.3±58.1 | 788.3±39.4 | 1721.3±53.2 | 914.1±37.2 | 1419.7±81.7 | 937.9±24.6 | 1530.2±23.4 | 907.2±59.8 | 1639.2±49.5 | 777.6±55.5 | 1635.7±56.4 | 791.4±76.5 | 1658.6±41.2 | 206.2±4.5 | 2437.0±3.8 |
| NOTEARS | 62.6±12.9 | 984.5±202.4 | 122.7±19.3 | 1782.4±177.0 | 205.6±26.7 | 2143.8±53.3 | 324.0±47.1 | 2214.9±53.0 | 135.1±19.3 | 1585.1±250.7 | 60.2±18.5 | 952.6±261.0 | 181.3±7.1 | 2353.6±37.4 | 61.5±16.6 | 1113.8±186.4 | 188.7±0.9 | 2375.7±24.7 |
| GOLEM | 46.3±19.1 | 1062.9±316.5 | 119.3±17.2 | 1579.3±160.3 | 186.3±23.4 | 2231.6±67.1 | 191.3±0.9 | 2399.1±1.1 | 134.4±23.4 | 1814.1±234.2 | 55.0±15.0 | 1171.7±389.1 | 174.0±6.1 | 2392.9±22.3 | 46.9±13.0 | 1134.8±267.6 | 187.4±0.9 | 2374.4±24.0 |
| NoCurl | 24.6±10.5 | 252.4±70.2 | 344.3±34.6 | 811.9±150.9 | 213.8±31.1 | 2045.1±45.5 | 454.7±63.5 | 1289.7±473.1 | 175.0±38.2 | 872.5±231.3 | 36.1±13.8 | 374.9±126.5 | 244.1±41.2 | 2177.2±95.6 | 31.4±14.2 | 291.9±115.3 | 188.8±0.9 | 2380.6±24.0 |
| DAGMA | 27.9±14.6 | 484.9±278.0 | 113.4±9.9 | 1957.3±116.1 | 197.2±4.7 | 2387.8±6.9 | 237.5±34.4 | 2326.0±47.7 | 123.7±21.5 | 1326.7±206.6 | 33.3±12.3 | 596.9±237.5 | 181.6±7.9 | 2317.6±70.9 | 24.1±9.4 | 421.5±123.1 | 188.7±0.9 | 2375.7±24.7 |

Table 17: Nonlinear Setting, for SF-4 graphs of 10, 20, 50 nodes.

| 10 nodes | Vanilla model | | Latent confounders | | Measurement error | | Autoregressive | | Heterogeneous | | Unfaithful | | Scale-variant | | Missing | | Mechanism violation | |
|---|---|---|---|---|---|---|---|---|---|---|---|---|---|---|---|---|---|---|
| | SHD↓ | SID↓ | SHD↓ | SID↓ | SHD↓ | SID↓ | SHD↓ | SID↓ | SHD↓ | SID↓ | SHD↓ | SID↓ | SHD↓ | SID↓ | SHD↓ | SID↓ | SHD↓ | SID↓ |
| Random | 29.5±2.9 | 75.8±5.1 | 29.9±4.0 | 75.6±4.1 | 28.2±2.7 | 73.9±4.5 | 31.6±3.2 | 78.0±4.1 | 32.1±4.0 | 76.4±7.9 | 29.2±2.7 | 73.9±4.4 | 30.3±3.4 | 72.7±5.7 | 28.5±3.2 | 71.9±5.2 | 30.0±3.0 | 74.9±6.5 |
| PC | 26.3±2.7 | 77.7±6.9 | 27.5±1.4 | 85.5±3.4 | 27.5±2.2 | 85.3±4.1 | 27.9±2.0 | 78.7±7.2 | 29.1±3.0 | 72.6±6.0 | 26.1±1.9 | 69.8±8.8 | 26.3±2.7 | 77.7±6.9 | 25.6±2.2 | 77.5±4.8 | 27.3±2.8 | 71.4±6.1 |
| GES | 26.1±2.2 | 82.6±5.0 | 26.3±1.5 | 78.4±6.4 | 28.3±1.3 | 82.9±7.5 | 28.2±2.2 | 78.1±8.7 | 26.8±3.2 | 71.1±8.9 | 21.3±3.1 | 60.5±11.7 | 26.1±2.2 | 82.6±5.0 | 25.0±2.4 | 79.8±5.4 | 23.0±5.1 | 62.7±9.8 |
| CAM | 5.2±2.4 | 16.1±8.4 | 13.8±4.4 | 43.7±16.8 | 20.8±3.8 | 64.9±14.5 | 15.8±4.3 | 46.0±11.1 | 11.9±3.5 | 31.9±15.3 | 9.9±4.1 | 44.9±11.2 | 5.2±2.4 | 16.1±8.4 | 4.5±1.6 | 18.4±7.5 | 25.7±4.2 | 73.2±5.0 |
| NOTEARS-MLP | 10.7±4.4 | 12.4±15.7 | 20.3±4.9 | 51.7±12.6 | 23.7±2.8 | 65.6±3.1 | 26.5±3.2 | 66.9±9.8 | 17.4±3.7 | 44.0±13.1 | 15.9±4.3 | 48.9±14.9 | 11.5±5.6 | 69.8±12.1 | 12.2±6.8 | 40.1±17.4 | 41.6±11.7 | |
| GraN-DAG | 22.3±2.6 | 68.5±10.9 | 24.5±3.6 | 73.2±7.5 | 28.0±0.9 | 77.5±5.4 | 28.3±1.3 | 74.7±4.2 | 22.2±3.5 | 56.4±11.0 | 11.5±3.4 | 40.4±14.7 | 25.8±2.3 | 81.6±7.0 | 22.1±2.8 | 65.4±9.5 | 11.9±5.3 | 48.3±19.9 |
| DAGMA | | | | | | | | | | | | | | | | | | |
| **20 nodes** | SHD↓ | SID↓ | SHD↓ | SID↓ | SHD↓ | SID↓ | SHD↓ | SID↓ | SHD↓ | SID↓ | SHD↓ | SID↓ | SHD↓ | SID↓ | SHD↓ | SID↓ | SHD↓ | SID↓ |
| Random | 111.2±4.5 | 307.7±14.1 | 114.1±4.8 | 302.9±10.2 | 111.6±5.7 | 303.8±17.9 | 116.0±6.9 | 311.1±13.9 | 114.4±5.7 | 306.5±10.9 | 110.6±5.0 | 303.1±9.6 | 114.2±8.1 | 307.5±16.5 | 116.0±7.5 | 305.3±8.5 | 109.1±6.2 | 305.7±6.9 |
| PC | 67.6±2.7 | 371.1±7.9 | 68.2±1.1 | 355.7±6.9 | 68.5±1.9 | 373.2±8.2 | 70.0±1.1 | 369.4±13.3 | 85.2±3.5 | 349.9±9.3 | 68.4±5.0 | 348.1±2.6 | 67.6±2.7 | 371.1±7.9 | 66.3±2.0 | 370.3±6.5 | 77.5±2.8 | 345.1±10.6 |
| GES | 62.3±2.0 | 368.9±2.6 | 67.8±1.2 | 358.0±7.7 | 65.9±1.6 | 362.5±15.3 | 68.1±2.0 | 362.9±14.5 | 81.4±5.4 | 337.4±10.9 | 70.9±8.5 | 294.3±16.8 | 62.3±2.0 | 368.9±2.6 | 60.0±3.3 | 360.4±9.4 | 79.3±13.9 | 268.5±23.6 |
| CAM | 18.8±3.1 | 159.3±31.6 | 63.1±4.6 | 331.0±14.1 | 63.6±3.4 | 354.8±15.0 | 53.5±9.4 | 303.7±38.3 | 46.1±6.1 | 220.3±44.4 | 27.4±5.6 | 209.0±36.9 | 18.8±3.1 | 159.3±31.6 | 21.0±3.8 | 189.4±38.9 | 74.6±9.7 | 321.1±10.2 |
| NOTEARS-MLP | 56.1±4.1 | 324.4±16.4 | 68.7±0.9 | 379.5±5.4 | 64.4±2.2 | 342.7±11.7 | 70.0±0.0 | 361.0±0.0 | 66.7±6.5 | 320.3±13.8 | 37.3±4.9 | 259.8±39.1 | 61.0±2.5 | 340.7±22.8 | 54.9±4.7 | 322.2±15.0 | 34.2±4.4 | 239.6±32.6 |
| GraN-DAG | 59.4±11.3 | 264.2±67.6 | 67.3±3.2 | 354.6±8.6 | 69.0±1.5 | 354.9±16.0 | 69.4±2.9 | 330.9±9.1 | 64.9±3.9 | 330.9±9.1 | 35.8±3.3 | 258.7±42.5 | 60.0±2.7 | 352.9±18.3 | 57.4±4.1 | 341.5±19.4 | 37.3±4.8 | 291.3±30.4 |
| DAGMA | 58.5±3.3 | 349.3±18.3 | 65.2±2.7 | 358.3±19.6 | 67.3±1.9 | 351.6±19.8 | 69.9±4.5 | 360.7±0.5 | 67.6±7.5 | 324.7±22.9 | 35.8±3.3 | 258.7±42.5 | 60.0±2.7 | 352.9±18.3 | 57.4±4.1 | 341.5±19.4 | 37.3±4.8 | 291.3±30.4 |
| **50 nodes** | SHD↓ | SID↓ | SHD↓ | SID↓ | SHD↓ | SID↓ | SHD↓ | SID↓ | SHD↓ | SID↓ | SHD↓ | SID↓ | SHD↓ | SID↓ | SHD↓ | SID↓ | SHD↓ | SID↓ |
| Random | 660.0±17.3 | 1892.2±35.2 | 657.4±16.4 | 1933.4±30.8 | 656.6±27.0 | 1909.8±33.6 | 666.0±15.4 | 1894.2±34.5 | 667.8±11.4 | 1901.0±29.7 | 665.7±18.5 | 1909.3±40.7 | 655.5±12.3 | 1904.0±29.7 | 657.1±14.3 | 1899.0±46.3 | 661.7±23.1 | 1902.6±49.4 |
| CAM | 75.7±11.2 | 1619.3±168.4 | 160.7±2.1 | 1834.1±125.6 | 181.8±2.9 | 2390.8±33.6 | 175.5±17.3 | 2249.3±56.3 | 183.8±10.6 | 1941.0±66.1 | 73.9±12.4 | 1412.9±136.6 | 75.7±11.2 | 1619.3±168.4 | 74.4±5.1 | 1614.5±132.3 | 272.4±13.1 | 2261.2±27.3 |
| NOTEARS-MLP | 171.6±8.9 | 2307.1±47.9 | 187.3±9.2 | 2543.2±37.2 | 187.7±1.4 | 2374.6±32.7 | 189.3±18.6 | 2381.0±76.3 | 189.1±7.9 | 2331.8±30.2 | 115.3±7.6 | 2243.4±65.3 | 173.4±8.8 | 2374.8±44.0 | 165.7±7.1 | 2310.0±29.7 | 104.4±13.0 | 1865.2±149.9 |
| GraN-DAG | 187.8±1.4 | 2360.8±36.6 | 205.4±8.7 | 2431.6±52.3 | 190.7±15.6 | 2394.1±55.2 | 178.3±5.6 | 2369.7±46.8 | 187.0±3.7 | 2410.9±41.0 | 165.7±11.8 | 2375.9±29.7 | 187.4±1.7 | 2360.1±19.3 | 188.4±2.6 | 2383.8±38.3 | 170.6±6.6 | 2372.5±37.6 |
| DAGMA | 170.5±4.3 | 2332.4±43.1 | 179.5±6.2 | 2378.4±46.2 | 187.6±1.7 | 2374.6±39.1 | 178.3±5.6 | 2369.7±46.8 | 189.5±1.7 | 2394.6±16.5 | 106.9±9.6 | 2108.3±83.2 | 174.5±3.6 | 2390.4±32.2 | 170.0±5.5 | 2346.8±57.7 | 129.9±30.4 | 2219.4±134.3 |

Table 18: Linear Setting, for ER-6 graphs of 20 nodes.

| 20 nodes | Vanilla model | | Latent confounders | | Measurement error | | Autoregressive | | Heterogeneous | | Unfaithful | | Scale-variant | | Missing | | Mechanism violation | |
|---|---|---|---|---|---|---|---|---|---|---|---|---|---|---|---|---|---|---|
| | SHD↓ | SID↓ | SHD↓ | SID↓ | SHD↓ | SID↓ | SHD↓ | SID↓ | SHD↓ | SID↓ | SHD↓ | SID↓ | SHD↓ | SID↓ | SHD↓ | SID↓ | SHD↓ | SID↓ |
| Random | 107.8±5.0 | 260.2±18.9 | 105.2±6.2 | 236.6±20.0 | 102.5±5.6 | 267.4±24.6 | 101.7±7.2 | 253.9±29.2 | 103.7±8.4 | 254.7±29.0 | 101.1±5.3 | 247.7±22.9 | 106.2±5.6 | 275.5±8.3 | 103.9±9.9 | 265.1±18.9 | 105.0±5.6 | 255.5±33.6 |
| PC | 113.3±3.9 | 327.6±26.5 | 116.3±3.0 | 335.0±11.9 | 117.7±5.6 | 333.8±24.8 | 115.2±5.0 | 323.6±30.8 | 124.3±10.9 | 345.9±13.8 | 115.7±9.6 | 334.1±25.8 | 113.3±3.9 | 327.6±26.5 | 115.5±3.3 | 337.7±21.6 | 115.1±3.6 | 332.9±19.0 |
| GES | 121.8±13.4 | 267.5±23.5 | 122.1±7.8 | 267.0±15.0 | 125.3±10.2 | 321.2±18.2 | 105.8±9.5 | 290.3±34.5 | 103.6±9.6 | 299.5±10.5 | 115.9±10.4 | 296.1±28.4 | 114.7±9.7 | 297.3±32.4 | 121.8±13.4 | 267.5±23.5 | 62.7±1.9 | 368.1±27.9 |
| DirectLiNGAM | 99.0±5.0 | 277.2±31.3 | 104.8±4.6 | 278.3±25.9 | 100.8±7.2 | 277.0±13.9 | 99.6±7.8 | 269.0±23.3 | 102.7±9.9 | 268.8±24.9 | 103.3±11.8 | 278.9±35.0 | 127.9±6.9 | 346.2±18.8 | 104.3±9.4 | 278.2±27.2 | 107.9±7.5 | 287.2±19.4 |
| Var-SortnRegress | 45.7±10.1 | 118.0±39.7 | 70.7±7.5 | 90.6±37.3 | 80.0±5.6 | 161.7±32.6 | 74.1±6.3 | 120.5±44.1 | 71.5±4.6 | 117.8±36.9 | 64.6±11.3 | 148.5±45.5 | 125.7±8.4 | 289.6±16.3 | 41.5±13.9 | 105.6±33.1 | 38.1±13.4 | 91.0±43.8 |
| R²-SortnRegress | 73.3±8.6 | 161.4±21.9 | 80.3±5.7 | 139.5±26.8 | 106.1±6.2 | 265.8±23.2 | 90.1±13.4 | 192.0±36.5 | 90.9±6.6 | 182.0±31.4 | 90.1±14.6 | 204.1±41.3 | 66.6±17.8 | 164.5±34.5 | 59.6±13.8 | 123.7±41.3 | | |
| NOTEARS | 29.8±9.8 | 120.6±43.0 | 56.6±10.4 | 142.9±39.3 | 80.7±4.6 | 187.9±19.6 | 81.4±13.7 | 206.5±47.4 | 59.8±6.6 | 124.0±31.3 | 41.1±11.8 | 128.4±34.2 | 118.5±2.9 | 357.3±12.0 | 32.4±12.4 | 117.0±35.4 | 24.6±9.6 | 100.3±33.2 |
| GOLEM | 29.0±6.5 | 116.0±35.0 | 55.8±9.9 | 137.4±39.2 | 80.6±3.8 | 179.8±19.2 | 80.2±13.3 | 204.8±46.2 | 55.8±6.8 | 122.5±6.9 | 37.8±12.3 | 122.0±30.2 | 131.5±4.2 | 349.7±10.8 | 28.6±10.0 | 87.1±33.5 | 18.8±9.1 | 79.5±22.7 |
| NoCurl | 38.9±16.2 | 114.9±34.2 | 61.1±9.6 | 156.0±34.2 | 81.2±5.4 | 191.2±18.9 | 80.4±9.8 | 183.2±34.6 | 69.3±13.3 | 149.2±48.0 | 43.1±15.3 | 127.1±24.9 | 151.2±8.7 | 355.9±10.9 | 40.5±16.7 | 129.4±51.7 | 42.3±20.3 | 136.9±65.3 |
| DAGMA | 20.2±8.6 | 61.7±24.8 | 51.6±9.9 | 122.5±27.1 | 102.6±12.2 | 289.5±49.8 | 93.0±11.1 | 235.2±60.6 | 59.3±4.6 | 111.5±17.8 | 29.6±13.2 | 92.4±36.6 | 118.7±3.5 | 358.5±12.1 | 22.8±8.2 | 69.2±34.9 | 11.5±7.0 | 40.9±15.6 |

Table 19: MLP Setting, for ER-6 graphs of 20 nodes.

| 20 nodes | Vanilla model | | Latent confounders | | Measurement error | | Autoregressive | | Heterogeneous | | Unfaithful | | Scale-variant | | Missing | | Mechanism violation | |
|---|---|---|---|---|---|---|---|---|---|---|---|---|---|---|---|---|---|---|
| | SHD↓ | SID↓ | SHD↓ | SID↓ | SHD↓ | SID↓ | SHD↓ | SID↓ | SHD↓ | SID↓ | SHD↓ | SID↓ | SHD↓ | SID↓ | SHD↓ | SID↓ | SHD↓ | SID↓ |
| Random | 98.5±7.3 | 231.4±15.4 | 117.4±8.3 | 273.1±25.4 | 112.3±9.1 | 264.3±23.7 | 118.3±7.1 | 278.4±26.1 | 118.7±6.1 | 254.9±25.6 | 125.2±8.3 | 286.3±28.4 | 104.7±9.2 | 271.1±19.3 | 113.2±6.9 | 256.1±20.7 | 116.4±5.9 | 283.4±23.1 |
| PC | 110.1±7.3 | 330.5±18.4 | 109.8±7.0 | 349.2±13.0 | 115.6±7.4 | 337.6±16.4 | 108.0±3.3 | 335.8±13.8 | 118.7±6.1 | 339.7±19.6 | 114.4±6.0 | 337.2±18.0 | 110.1±7.3 | 330.5±18.4 | 109.5±7.4 | 329.3±18.1 | 113.3±3.9 | 327.6±26.5 |
| GES | 115.9±10.4 | 296.1±28.4 | 106.7±12.1 | 290.7±23.8 | 105.7±8.7 | 299.8±16.5 | 103.6±9.6 | 299.8±24.5 | 120.7±9.6 | 285.3±15.7 | 122.4±12.0 | 285.6±17.6 | 115.9±10.4 | 296.1±28.4 | 114.7±9.7 | 297.3±32.4 | 121.8±13.4 | 267.5±23.5 |
| CAM | 80.6±11.4 | 248.9±43.5 | 85.1±8.0 | 307.0±18.4 | 109.1±9.5 | 323.6±17.8 | 84.7±9.2 | 288.4±36.8 | 95.0±9.0 | 296.3±22.5 | 91.2±4.7 | 271.3±30.4 | 80.6±11.4 | 248.9±43.5 | 76.8±10.7 | 248.2±37.9 | 110.0±6.0 | 322.8±14.4 |
| NOTEARS-MLP | 35.1±10.3 | 155.8±53.4 | 59.7±7.0 | 215.9±36.4 | 101.2±10.2 | 216.6±28.0 | 85.9±16.2 | 211.3±49.3 | 55.9±8.3 | 192.0±40.2 | 90.6±11.3 | 226.3±27.7 | 102.2±6.1 | 340.9±15.8 | 42.8±7.4 | 174.1±50.5 | 103.9±14.3 | 303.3±27.4 |
| GraN-DAG | 102.6±8.0 | 289.8±20.6 | 95.1±8.1 | 311.4±34.5 | 114.5±3.2 | 312.8±20.9 | 111.4±4.8 | 318.6±16.8 | 111.4±5.3 | 323.8±12.0 | 109.9±6.0 | 311.3±17.0 | 108.8±5.6 | 332.0±17.6 | 104.7±9.7 | 298.9±31.7 | 106.4±10.6 | 307.1±35.3 |
| DAGMA | 49.6±5.1 | 162.5±39.4 | 77.5±9.1 | 299.7±20.5 | 114.5±2.6 | 341.2±10.6 | 77.3±8.2 | 194.3±42.1 | 65.6±7.0 | 155.4±36.1 | 107.2±13.0 | 310.0±35.7 | 89.0±6.5 | 280.1±20.8 | 40.3±5.6 | 158.3±37.2 | 94.2±11.7 | 240.2±27.3 |

Table 20: Linear Setting, for GRP-2 graphs of 20 nodes.

| 20 nodes | Vanilla model | | Latent confounders | | Measurement error | | Autoregressive | | Heterogeneous | | Unfaithful | | Scale-variant | | Missing | | Mechanism violation | |
|---|---|---|---|---|---|---|---|---|---|---|---|---|---|---|---|---|---|---|
| | SHD↓ | SID↓ | SHD↓ | SID↓ | SHD↓ | SID↓ | SHD↓ | SID↓ | SHD↓ | SID↓ | SHD↓ | SID↓ | SHD↓ | SID↓ | SHD↓ | SID↓ | SHD↓ | SID↓ |
| Random | 107.6±9.2 | 277.5±8.8 | 101.6±7.4 | 239.9±22.2 | 100.5±8.7 | 255.9±22.9 | 103.3±8.2 | 241.4±24.7 | 107.3±8.6 | 257.2±18.4 | 106.2±7.7 | 272.8±15.6 | 103.4±8.6 | 255.7±20.8 | 104.7±9.5 | 257.4±21.3 | 105.2±8.8 | 262.8±30.4 |
| PC | 2.7±3.5 | 7.3±9.9 | 33.8±3.8 | 39.3±23.2 | 13.0±5.7 | 15.1±20.7 | 11.4±4.2 | 24.9±21.3 | 4.3±2.8 | 10.5±10.6 | 5.4±4.2 | 12.4±12.6 | 2.7±3.5 | 7.3±9.9 | 3.5±3.0 | 10.5±14.3 | 12.2±4.5 | 42.7±27.2 |
| GES | 4.4±3.3 | 8.2±9.1 | 120.7±17.6 | 17.9±9.3 | 18.9±8.5 | 17.8±12.1 | 19.7±7.0 | 19.9±14.3 | 14.3±3.5 | 12.8±5.8 | 8.5±3.2 | 18.4±12.1 | 4.4±3.3 | 8.2±9.1 | 5.5±4.3 | 11.2±11.9 | 11.3±5.2 | 37.2±28.0 |
| DirectLiNGAM | 15.6±3.3 | 32.1±8.3 | 45.3±12.5 | 32.4±13.8 | 14.4±3.8 | 25.3±11.7 | 14.8±5.4 | 30.5±14.1 | 14.2±5.7 | 34.9±18.9 | 7.8±4.6 | 16.2±11.8 | 3.7±2.2 | 8.6±7.4 | 3.3±2.5 | 10.4±10.5 | 10.8±4.6 | 39.9±23.5 |
| Var-SortnRegress | 1.9±2.2 | 2.7±4.1 | 130.0±12.0 | 7.0±5.1 | 15.8±7.6 | 3.8±4.2 | 19.7±5.6 | 10.1±8.5 | 8.4±5.4 | 2.8±4.1 | 6.2±3.3 | 9.1±8.2 | 20.6±8.2 | 43.6±19.2 | 3.0±2.4 | 4.4±3.8 | 9.7±4.7 | 24.9±16.0 |
| R²-SortnRegress | 15.2±7.9 | 26.9±12.4 | 138.3±13.4 | 29.9±11.6 | 26.0±12.9 | 26.1±6.3 | 26.1±6.3 | 31.6±15.2 | 18.7±5.7 | 34.2±16.2 | 15.2±7.9 | 26.9±12.4 | 14.3±7.1 | 26.4±13.3 | 15.6±4.9 | 51.6±24.8 | | |
| NOTEARS | 1.1±1.3 | 3.7±4.9 | 8.5±3.9 | 20.2±15.3 | 9.4±4.9 | 3.9±2.8 | 18.3±16.3 | 15.1±10.3 | 2.1±1.1 | 4.0±4.7 | 0.4±0.9 | 1.9±2.3 | 17.4±4.1 | 49.8±22.3 | 0.4±0.8 | 0.5±2.7 | 14.2±2.9 | 39.8±18.9 |
| GOLEM | 0.4±0.5 | 1.7±2.9 | 8.6±4.2 | 20.0±16.9 | 9.7±5.2 | 3.8±2.5 | 17.6±15.9 | 15.7±10.1 | 2.3±4.9 | 3.5±3.9 | 0.6±1.5 | 1.9±2.3 | 17.4±4.1 | 49.8±22.3 | 0.4±0.8 | 0.5±2.7 | 14.2±2.9 | 39.8±18.9 |
| NoCurl | 1.5±1.4 | 5.7±4.3 | 54.8±6.2 | 6.4±3.9 | 10.1±5.2 | 5.2±3.1 | 14.1±6.5 | 11.3±11.7 | 2.4±2.1 | 1.7±3.3 | 0.5±1.2 | 2.1±4.3 | 19.4±5.9 | 57.2±22.6 | 0.8±1.4 | 3.2±5.3 | 14.6±3.7 | 39.0±20.1 |
| DAGMA | 0.0±0.0 | 0.0±0.0 | 7.8±3.8 | 19.3±16.6 | 9.6±4.9 | 4.4±2.5 | 14.1±6.5 | 11.3±10.4 | 1.1±1.2 | 1.7±3.3 | 0.3±0.9 | 1.2±3.6 | 15.4±2.8 | 45.7±17.8 | 0.0±0.0 | 0.0±0.0 | 13.8±2.8 | 38.9±17.6 |

Table 21: MLP Setting, for GRP-2 graphs of 20 nodes.

| 20 nodes | Vanilla model | | Latent confounders | | Measurement error | | Autoregressive | | Heterogeneous | | Unfaithful | | Scale-variant | | Missing | | Mechanism violation | |
|---|---|---|---|---|---|---|---|---|---|---|---|---|---|---|---|---|---|---|
| | SHD↓ | SID↓ | SHD↓ | SID↓ | SHD↓ | SID↓ | SHD↓ | SID↓ | SHD↓ | SID↓ | SHD↓ | SID↓ | SHD↓ | SID↓ | SHD↓ | SID↓ | SHD↓ | SID↓ |
| Random | 105.3±6.9 | 260.3±24.1 | 108.1±7.4 | 243.0±23.1 | 100.8±8.0 | 245.7±23.6 | 107.3±8.6 | 257.2±18.4 | 107.3±8.6 | 257.2±18.4 | 106.2±7.7 | 272.8±15.6 | 103.4±8.6 | 255.7±20.8 | 104.7±9.5 | 257.4±21.3 | 105.8±2.9 | 266.9±20.3 |
| PC | 12.1±6.0 | 27.5±25.8 | 31.4±4.3 | 42.3±21.0 | 13.9±5.0 | 20.6±18.3 | 13.9±3.5 | 29.8±24.1 | 23.5±4.8 | 31.5±22.0 | 13.6±4.6 | 27.5±16.1 | 12.1±5.0 | 27.5±25.8 | 11.5±4.2 | 25.2±21.9 | 2.7±3.5 | 7.3±9.9 |
| GES | 18.2±7.5 | 26.0±22.0 | 72.3±10.9 | 26.4±17.2 | 19.5±5.7 | 25.9±11.5 | 20.0±5.2 | 32.8±21.9 | 51.5±16.1 | 29.1±19.1 | 17.0±5.5 | 23.3±14.7 | 18.2±7.5 | 26.0±22.0 | 24.2±26.7 | 4.4±3.3 | 8.2±9.1 | |
| CAM | 3.7±2.2 | 8.6±1.7.4 | 44.6±3.4 | 33.9±15.1 | 19.9±4.4 | 48.7±30.1 | 17.9±5.6 | 35.0±25.9 | 13.0±8.4 | 36.9±22.5 | 7.8±4.6 | 16.2±11.8 | 3.7±2.2 | 8.6±17.4 | 3.3±2.5 | 10.4±10.5 | 10.8±4.6 | 39.9±23.5 |
| NOTEARS-MLP | 3.4±2.1 | 5.5±4.4 | 13.9±2.7 | 41.3±16.4 | 15.0±3.0 | 34.9±18.8 | 23.2±9.5 | 38.8±18.1 | 11.9±3.4 | 21.0±12.3 | 6.6±4.5 | 7.5±4.9 | 14.5±3.3 | 43.4±18.8 | 3.4±1.9 | 4.2±4.3 | 4.1±3.1 | 14.9±14.9 |
| GraN-DAG | 9.1±4.4 | 23.6±16.8 | 11.3±3.7 | 35.0±23.0 | 13.3±4.0 | 31.9±19.7 | 12.8±3.6 | 20.5±15.1 | 8.3±2.7 | 20.5±15.1 | 9.1±3.1 | 22.3±15.3 | 12.1±3.5 | 34.6±20.8 | 12.8±3.3 | 30.5±16.4 | 12.2±4.6 | 28.4±17.9 |
| DAGMA | 2.6±1.2 | 3.4±2.2 | 11.8±2.1 | 33.7±14.2 | 12.2±3.1 | 28.5±16.3 | 13.1±3.9 | 24.5±8.9 | 8.3±2.7 | 15.5±9.6 | 4.8±2.3 | 6.8±3.7 | 12.2±3.5 | 38.5±7.9 | 2.9±1.1 | 3.8±1.8 | 3.8±2.3 | 11.6±7.1 |

## F  TABLE RESULTS FOR COMBINED MISSPECIFIED SCENARIOS

We consider two (confounding and heterogeneity), three (confounding, measurement error, and heterogeneity) and four (confounding, measurement error, heterogeneity, and autoregression) combined misspecified scenarios. The results in Table 22 and Table 23 show that under combined misspecified scenarios, the performance of various methods is worse compared to single misspecified scenario. However, differentiable causal discovery still achieves optimal or competitive performance.

Table 22: Linear Setting with two (confounding and heterogeneity), three (confounding, measurement error, and heterogeneity) and four (confounding, measurement error, heterogeneity, and autoregression) combined misspecified scenarios, for ER-2 graphs of 10 nodes.

| 10 nodes | Vanilla model SHD↓ | SID↓ | Latent confounders SHD↓ | SID↓ | Measurement error SHD↓ | SID↓ | Heterogeneous SHD↓ | SID↓ | Autoregressive SHD↓ | SID↓ | Two combined scenarios SHD↓ | SID↓ | Three combined scenarios SHD↓ | SID↓ | Four combined scenarios SHD↓ | SID↓ |
|---|---|---|---|---|---|---|---|---|---|---|---|---|---|---|---|---|
| Random | 29.1±1.1 | 68.4±2.7 | 25.1±4.7 | 65.2±5.7 | 24.3±3.9 | 65.6±6.6 | 26.2±4.3 | 70.4±2.3 | 27.5±3.6 | 65.8±4.2 | 27.1±1.3 | 63.4±4.5 | 25.7±4.2 | 61.2±7.2 | 28.1±2.3 | 69.2±3.5 |
| PC | 12.4±3.1 | 40.9±13.4 | 18.1±4.7 | 58.1±15.6 | 19.4±4.1 | 48.0±13.1 | 13.8±2.6 | 47.5±10.3 | 14.5±2.0 | 44.8±9.5 | 19.5±3.2 | 61.0±8.3 | 21.8±4.4 | 55.3±13.5 | 21.6±4.0 | 59.3±10.0 |
| GES | 13.8±7.8 | 32.0±13.6 | 25.9±7.7 | 42.6±14.0 | 20.2±4.8 | 46.2±16.7 | 15.5±6.1 | 35.2±12.2 | 20.8±5.5 | 49.7±11.5 | 28.1±3.4 | 41.2±12.8 | 27.4±1.7 | 62.0±8.9 | 24.7±5.0 | 50.2±11.1 |
| DirectLiNGAM | 19.6±3.3 | 46.1±10.6 | 20.4±5.0 | 42.0±6.0 | 17.6±2.4 | 48.8±12.4 | 16.3±3.9 | 34.7±9.9 | 19.7±4.2 | 50.4±8.4 | 22.4±2.4 | 46.7±12.7 | 18.0±3.1 | 43.7±6.3 | 18.9±3.2 | 38.9±4.8 |
| Var-SortnRegress | 11.2±3.5 | 8.4±8.5 | 17.6±5.8 | 12.6±9.9 | 19.6±2.8 | **11.4±8.7** | 17.9±3.3 | 8.6±9.3 | 18.8±2.4 | **16.5±10.6** | 21.5±3.9 | 11.5±10.8 | 20.7±3.3 | **13.0±9.9** | 20.3±3.3 | 23.7±9.3 |
| $R^2$-SortnRegress | 20.2±4.8 | 32.4±14.0 | 25.7±4.1 | 37.6±13.0 | 25.6±6.0 | 39.2±16.0 | 26.0±5.4 | 37.0±14.4 | 25.6±4.9 | 38.8±19.0 | 25.5±3.6 | 31.9±17.6 | 26.8±3.8 | 48.3±10.4 | 26.8±5.2 | 48.1±12.1 |
| NOTEARS | 1.5±1.6 | 1.8±4.2 | 8.5±3.9 | 9.5±8.1 | 12.5±2.0 | 19.6±8.6 | **5.5±2.7** | **5.4±5.1** | **12.2±3.6** | 27.5±14.2 | 12.3±3.4 | 33.1±5.7 | **14.1±3.5** | 25.2±6.8 | 18.6±3.9 | 31.8±10.1 |
| GOLEM | 1.4±1.4 | **0.4±1.2** | **6.7±2.8** | 14.2±9.8 | 17.8±2.5 | 43.1±13.3 | 6.5±4.5 | 9.8±8.1 | 16.6±4.0 | 34.9±16.9 | **10.3±1.7** | 24.3±5.0 | 18.6±3.7 | 46.2±9.1 | 19.7±0.5 | 72.7±6.3 |
| NoCurl | 2.0±1.8 | 5.1±5.8 | 9.1±4.2 | **5.4±3.9** | **11.8±1.8** | **17.9±8.4** | 6.6±2.9 | 5.5±5.7 | 14.8±2.5 | **17.5±10.8** | 13.4±3.4 | **7.4±6.9** | 15.4±4.3 | **19.8±7.0** | 20.2±3.5 | **21.3±8.2** |
| DAGMA | **1.2±1.2** | 3.3±5.3 | 8.4±3.9 | 8.8±7.7 | 12.6±2.5 | 18.5±8.6 | **5.5±2.3** | 12.0±8.2 | **12.2±3.6** | 28.4±15.3 | 12.0±3.8 | 12.3±8.9 | 14.9±4.4 | 20.4±6.6 | **17.9±3.1** | 50.2±14.2 |

Table 23: MLP Setting with two (confounding and heterogeneity), three (confounding, measurement error, and heterogeneity) and four (confounding, measurement error, heterogeneity, and autoregression) combined misspecified scenarios, for ER-2 graphs of 10 nodes.

| 10 nodes | Vanilla model SHD↓ | SID↓ | Latent confounders SHD↓ | SID↓ | Measurement error SHD↓ | SID↓ | Heterogeneous SHD↓ | SID↓ | Autoregressive SHD↓ | SID↓ | Two combined scenarios SHD↓ | SID↓ | Three combined scenarios SHD↓ | SID↓ | Four combined scenarios SHD↓ | SID↓ |
|---|---|---|---|---|---|---|---|---|---|---|---|---|---|---|---|---|
| Random | 27.5±2.6 | 62.6±7.3 | 29.4±1.1 | 62.8±9.4 | 29.3±1.2 | 69.4±3.1 | 28.9±1.8 | 70.4±2.8 | 27.9±2.3 | 72.2±5.5 | 28.4±2.0 | 67.2±4.5 | 24.2±4.1 | 66.2±6.3 | 29.6±2.3 | 68.5±5.6 |
| PC | 16.7±3.2 | 52.0±9.5 | 18.3±3.5 | 54.1±9.3 | 17.9±5.0 | 47.4±15.5 | 20.7±3.5 | 53.9±13.1 | 16.2±4.7 | 49.2±13.1 | 22.9±2.6 | 63.7±9.6 | 24.7±3.2 | 64.4±11.5 | 22.6±3.2 | 65.1±14.3 |
| GES | 21.9±4.4 | 46.9±8.6 | 28.2±7.5 | 53.1±18.6 | 20.2±5.4 | 54.7±14.3 | 27.2±4.9 | 48.6±10.9 | 18.9±5.2 | 43.2±15.5 | 28.6±2.9 | 53.0±8.4 | 28.5±3.5 | 62.9±6.3 | 24.6±2.3 | 61.6±6.7 |
| CAM | 12.4±3.6 | 39.3±16.5 | 16.6±4.2 | 42.4±17.7 | 19.7±4.7 | 59.6±12.0 | 19.9±3.6 | 49.1±7.2 | 16.0±3.4 | 46.0±16.6 | 23.4±3.1 | 60.9±6.1 | 25.4±3.3 | 53.8±13.4 | 24.4±1.9 | 63.4±8.8 |
| NOTEARS-MLP | 8.1±2.7 | 22.2±10.6 | 11.7±5.5 | 33.3±17.0 | 18.5±3.7 | 47.1±11.9 | 14.5±2.2 | 34.3±10.4 | 15.7±4.3 | 39.8±9.6 | 18.5±2.3 | 57.4±11.7 | 19.6±0.7 | 63.4±10.2 | 22.5±3.3 | 61.0±9.2 |
| GraN-DAG | 13.3±3.6 | 32.9±12.3 | 12.9±3.2 | 40.2±12.7 | 16.6±2.0 | 45.0±10.5 | 14.3±2.5 | 38.5±7.7 | 15.5±2.3 | 46.0±9.5 | 16.0±2.5 | 52.9±11.2 | 20.4±1.6 | 63.8±7.2 | 20.3±2.8 | 62.2±10.0 |
| DAGMA | **6.2±1.7** | **18.2±8.7** | **9.3±4.3** | **27.8±10.8** | **14.1±2.6** | **39.2±8.7** | **12.7±2.9** | **31.6±7.2** | **13.6±2.1** | **41.2±3.6** | **15.3±2.7** | **48.2±10.4** | **18.4±0.9** | **51.6±10.4** | **19.6±2.5** | **60.8±7.3** |

## G  TABLE RESULTS FOR NON-GAUSSIAN NOISE

We consider the vanilla model with exponential noise. The results in Table 24 and Table 25 show that differentiable causal discovery still achieve optimal or competitive performance when model assumptions are violated.

Table 24: Linear Setting with exponential noise, for ER-2 graphs of 10 nodes.

| 10 nodes | Vanilla model SHD↓ | SID↓ | Latent confounders SHD↓ | SID↓ | Measurement error SHD↓ | SID↓ | Autoregressive SHD↓ | SID↓ | Heterogeneous SHD↓ | SID↓ | Unfaithful SHD↓ | SID↓ | Scale-variant SHD↓ | SID↓ | Missing SHD↓ | SID↓ | Mechanism violation SHD↓ | SID↓ |
|---|---|---|---|---|---|---|---|---|---|---|---|---|---|---|---|---|---|---|
| Random | 29.2±1.4 | 64.0±7.8 | 27.5±1.9 | 69.2±3.0 | 25.7±2.6 | 65.3±4.8 | 25.1±2.8 | 65.7±8.5 | 26.5±3.0 | 64.2±7.6 | 24.7±3.1 | 71.0±2.4 | 28.1±1.6 | 65.7±5.4 | 26.6±3.4 | 66.1±5.4 | 27.9±3.0 | 67.2±7.2 |
| PC | 11.7±2.3 | 38.7±7.3 | 20.1±2.6 | 58.6±8.7 | 20.5±3.3 | 46.8±15.0 | 17.9±2.2 | 50.2±8.8 | 20.7±4.3 | 58.4±14.5 | 14.0±3.1 | 44.6±13.5 | 11.7±2.3 | 38.7±7.3 | 14.4±4.8 | 42.2±15.3 | 17.9±3.6 | 54.4±12.2 |
| GES | 19.4±8.1 | 41.1±19.7 | 24.8±6.8 | 41.6±16.0 | 24.4±4.4 | 56.7±14.5 | 22.3±3.8 | 37.0±11.9 | 26.4±5.0 | 45.3±17.3 | 18.3±5.8 | 38.2±14.5 | 18.9±8.3 | 37.9±20.1 | 17.3±6.5 | 34.3±19.7 | 19.6±4.9 | 42.5±12.9 |
| DirectLiNGAM | **0.0±0.0** | **0.0±0.0** | 13.6±5.9 | 20.1±15.4 | 15.0±3.7 | 21.1±9.5 | 12.0±3.4 | 18.1±12.2 | 9.1±4.3 | 10.3±9.2 | 1.4±2.2 | 3.9±4.9 | **4.9±2.5** | **18.3±9.0** | **0.0±0.0** | **0.0±0.0** | 16.5±4.3 | 45.9±10.1 |
| Var-SortnRegress | 6.8±4.5 | 9.7±7.8 | 17.0±5.2 | 10.0±6.6 | 18.4±4.0 | 12.2±9.1 | 16.2±2.7 | 10.2±7.9 | 22.1±2.1 | 10.9±6.9 | 8.7±6.1 | 15.0±12.0 | 26.9±5.5 | 58.2±8.3 | 5.7±3.8 | 10.6±11.3 | 19.4±5.0 | 28.8±7.9 |
| $R^2$-SortnRegress | 15.7±6.7 | 30.4±12.2 | 27.9±4.1 | 35.9±16.1 | 26.4±5.9 | 44.5±14.9 | 23.1±5.0 | 37.3±13.9 | 26.6±3.4 | 32.6±14.3 | 20.9±8.3 | 34.3±15.0 | 15.7±6.7 | 30.4±12.2 | 15.8±7.1 | 27.0±10.0 | 29.0±4.2 | 55.8±11.0 |
| NOTEARS | 1.5±1.7 | 6.7±7.7 | 10.7±3.7 | 14.0±7.2 | 13.4±3.5 | 15.8±6.7 | 14.4±5.2 | 18.5±12.1 | **6.6±2.7** | 13.2±8.3 | **0.1±0.3** | **0.7±2.1** | 18.9±1.5 | **65.3±8.7** | 0.9±1.4 | 5.1±7.0 | **14.1±4.6** | 29.0±6.2 |
| GOLEM | 2.3±2.7 | 6.6±8.4 | 10.5±3.4 | 13.9±7.1 | 14.3±4.1 | 18.0±7.3 | 13.6±5.5 | 20.1±13.3 | 6.7±2.7 | 13.5±8.6 | 0.2±0.4 | 0.8±2.7 | 18.6±2.2 | 67.7±10.6 | 0.6±1.7 | 4.5±6.6 | 14.5±4.4 | 28.9±5.7 |
| NoCurl | 5.4±3.6 | 11.5±8.5 | 14.6±4.0 | 14.4±8.7 | 16.7±4.6 | **14.3±8.6** | 18.1±6.2 | **16.8±13.7** | 11.9±4.7 | 9.0±8.7 | 8.6±4.3 | 9.2±12.1 | 24.6±6.1 | 65.6±13.9 | 8.0±2.7 | 7.1±7.7 | 22.2±4.4 | 27.9±8.5 |
| DAGMA | **0.0±0.0** | **0.0±0.0** | 10.1±2.9 | 13.8±6.9 | 13.2±2.4 | 24.6±9.3 | 12.9±4.7 | 31.0±10.8 | 6.8±2.9 | **4.7±5.3** | **0.1±0.3** | **0.7±2.1** | 18.2±4.3 | 67.5±10.4 | **0.3±0.9** | **0.6±1.8** | 14.6±4.7 | 28.8±7.3 |

Table 25: MLP Setting with exponential noise, for ER-2 graphs of 10 nodes.

| 10 nodes | Vanilla model SHD↓ | SID↓ | Latent confounders SHD↓ | SID↓ | Measurement error SHD↓ | SID↓ | Autoregressive SHD↓ | SID↓ | Heterogeneous SHD↓ | SID↓ | Unfaithful SHD↓ | SID↓ | Scale-variant SHD↓ | SID↓ | Missing SHD↓ | SID↓ | Mechanism violation SHD↓ | SID↓ |
|---|---|---|---|---|---|---|---|---|---|---|---|---|---|---|---|---|---|---|
| Random | 27.9±2.2 | 62.6±6.6 | 26.3±2.5 | 62.6±5.9 | 28.0±1.3 | 66.9±6.9 | 25.0±3.2 | 68.1±3.2 | 23.8±1.1 | 67.6±4.3 | 26.1±4.0 | 60.3±5.7 | 30.0±1.0 | 69.9±2.6 | 25.4±5.0 | 63.2±8.4 | 26.7±4.8 | 64.7±9.2 |
| PC | 17.9±3.6 | 54.4±12.2 | 17.2±4.7 | 57.3±14.1 | 18.4±5.1 | 49.2±9.7 | 17.0±3.9 | 55.5±11.5 | 23.2±2.5 | 62.5±11.7 | 17.9±2.4 | 48.2±13.7 | 17.9±3.6 | 54.4±12.2 | 17.0±3.9 | 53.2±10.2 | 11.7±2.3 | 38.7±7.3 |
| GES | 19.6±4.9 | 42.5±12.9 | 22.8±7.0 | 42.8±18.0 | 19.6±4.7 | 46.6±12.4 | 17.1±4.8 | 47.0±16.2 | 30.0±2.3 | 58.5±8.4 | 23.6±6.0 | 44.1±12.3 | 19.6±4.9 | 42.2±12.5 | 20.7±4.3 | 44.2±14.5 | 19.4±8.1 | 41.1±19.7 |
| CAM | 11.2±2.6 | 40.3±17.3 | 14.1±4.9 | 35.5±14.1 | 21.2±3.8 | 61.6±11.2 | 13.7±3.5 | 42.3±16.4 | 23.3±5.3 | 63.5±12.6 | 12.5±5.2 | 35.7±21.4 | **11.2±2.6** | **40.3±17.3** | 12.3±2.1 | 41.6±20.6 | 21.2±7.6 | 62.8±13.1 |
| NOTEARS-MLP | 6.6±2.3 | 17.1±11.0 | 13.6±5.3 | 38.7±12.5 | 16.2±5.1 | 31.8±11.2 | 15.0±5.7 | 34.3±17.7 | 12.6±3.2 | 25.6±10.5 | 11.2±2.4 | 21.6±8.5 | 17.2±1.7 | 61.7±9.6 | 6.5±2.5 | 16.5±11.9 | 10.4±4.2 | 43.9±17.5 |
| GraN-DAG | 12.2±3.0 | 38.6±15.1 | 15.1±2.9 | 51.5±10.1 | 16.4±1.7 | 47.3±7.2 | 14.9±2.8 | 48.1±12.8 | 13.6±3.9 | 33.8±15.1 | 14.1±2.0 | 42.0±9.6 | 17.3±1.9 | 54.6±12.4 | 12.6±3.1 | 40.5±13.1 | 14.7±3.7 | 47.4±12.3 |
| DAGMA | **4.5±2.5** | **13.5±9.3** | **12.1±5.2** | **32.0±9.7** | **14.3±5.0** | **28.2±9.3** | **12.2±5.6** | **28.5±12.5** | **10.8±3.0** | **21.8±8.6** | **9.2±1.7** | **19.5±7.6** | 15.5±1.6 | 52.0±9.7 | **5.3±2.4** | **15.2±11.8** | **8.9±3.1** | **38.3±14.6** |

## H SUMMARY OF THE MOST COMPETITIVE METHODS

Table 26: Summary of performances of the most competitive methods under linear setting. The reported results are the mean and standard deviation of the metrics over 10 repetitions across different graph types, vanilla and model assumption violation scenarios.

| Method | $d$ | SHD | SID |
|--------|-----|-----|-----|
| NOTEARS | 10 | 8.51±5.92 | 23.88±23.32 |
|         | 20 | 22.66±14.27 | 129.96±109.48 |
|         | 50 | 61.07±39.27 | 945.01±774.12 |
| GOLEM | 10 | 9.02±6.90 | 30.67±27.39 |
|       | 20 | 21.79±14.19 | 155.60±126.41 |
|       | 50 | 55.57±35.00 | 1000.95±842.80 |
| NoCurl | 10 | 9.39±7.06 | 23.59±23.92 |
|        | 20 | 32.22±23.51 | **99.43±99.13** |
|        | 50 | 128.97±113.22 | 914.22±699.51 |
| DAGMA | 10 | **8.17±6.12** | **22.42±22.72** |
|       | 20 | **20.98±14.10** | **127.31±112.55** |
|       | 50 | **55.19±37.87** | **882.76±853.40** |

Table 27: Summary of performances of the most competitive methods under nonlinear setting. The reported results are the mean and standard deviation of the metrics over 10 repetitions across different graph types, vanilla and model assumption violation scenarios.

| Method | $d$ | SHD | SID |
|--------|-----|-----|-----|
| CAM | 10 | **8.21±5.17** | **22.66±16.98** |
|     | 20 | **22.23±12.36** | **117.37±79.80** |
|     | 50 | **61.92±35.50** | **696.47±463.94** |
| NOTEARS-MLP | 10 | 12.23±3.63 | 44.68±10.27 |
|             | 20 | 28.86±7.74 | 217.09±78.45 |
|             | 50 | 73.40±18.41 | 1271.24±560.81 |
| GraN-DAG | 10 | **10.44±5.11** | **38.19±16.78** |
|          | 20 | 30.46±8.23 | 214.86±75.62 |
|          | 50 | 85.34±11.81 | 1453.35±575.73 |
| DAGMA | 10 | 12.64±4.66 | 46.99±17.88 |
|       | 20 | **28.28±8.83** | **211.26±85.84** |
|       | 50 | **72.07±20.15** | 1254.57±583.11 |

Table 28: Summary of performances of the most competitive methods under MLP setting. The reported results are the mean and standard deviation of the metrics over 10 repetitions across different graph types, vanilla and model assumption violation scenarios.

| Method | $d$ | SHD | SID |
|--------|-----|-----|-----|
| CAM | 10 | 15.67±2.87 | 44.70±6.53 |
|     | 20 | 34.88±9.23 | 179.14±29.38 |
|     | 50 | 84.89±26.26 | 904.99±134.22 |
| NOTEARS-MLP | 10 | **12.93±4.24** | **34.63±12.57** |
|             | 20 | **26.97±8.02** | **132.86±52.22** |
|             | 50 | **70.08±23.80** | **740.91±239.08** |

# I  TABLE RESULTS ON REAL-WORLD DATA

We test the performance of 12 benchmark methods on the real-world Sachs (Sachs et al., 2005) dataset. Sachs is a bioinformatics dataset used to study the expression levels of various proteins and phospholipids in human cells, and it is a commonly used benchmark in the causal discovery field. We conduct experiments based on 7466 samples. The true graph structure of the Sachs dataset contains 11 nodes and 17 edges, and it is widely accepted by the biological research community.

Table 29: Results on Sachs dataset.

| Method | SHD | SID |
|---|---|---|
| Random | 33 | 56 |
| PC | 22 | 49 |
| GES | 30 | 47 |
| DirectLiNGAM | 14 | 50 |
| Var-SortnRegress | 19 | 49 |
| $R^2$-SortnRegress | 22 | 51 |
| NOTEARS | 17 | 48 |
| GOLEM | 15 | 58 |
| NoCurl | 16 | 50 |
| CAM | 15 | 51 |
| NOTEARS-MLP | 14 | 46 |
| GraN-DAG | 15 | 45 |
| DAGMA | **12** | **42** |

The results in Table 29 show that, represented by DAGMA, differentiable causal discovery achieves optimal performance on the real-world Sachs dataset. Considering that Sachs is also regarded as a real-world heterogeneous dataset (Mooij et al., 2020), the results on both Sachs and synthetic datasets further indicate that differentiable causal discovery performs better under model assumption violations.

## J FIGURE RESULTS ACROSS NODES, GRAPH TYPES, AND GRAPH DENSITIES

This section presents a comprehensive analysis of the figure results across varying numbers of nodes, graph types, and graph densities, in Figure 2, 3, 4, 5, 6, 7, and 8.

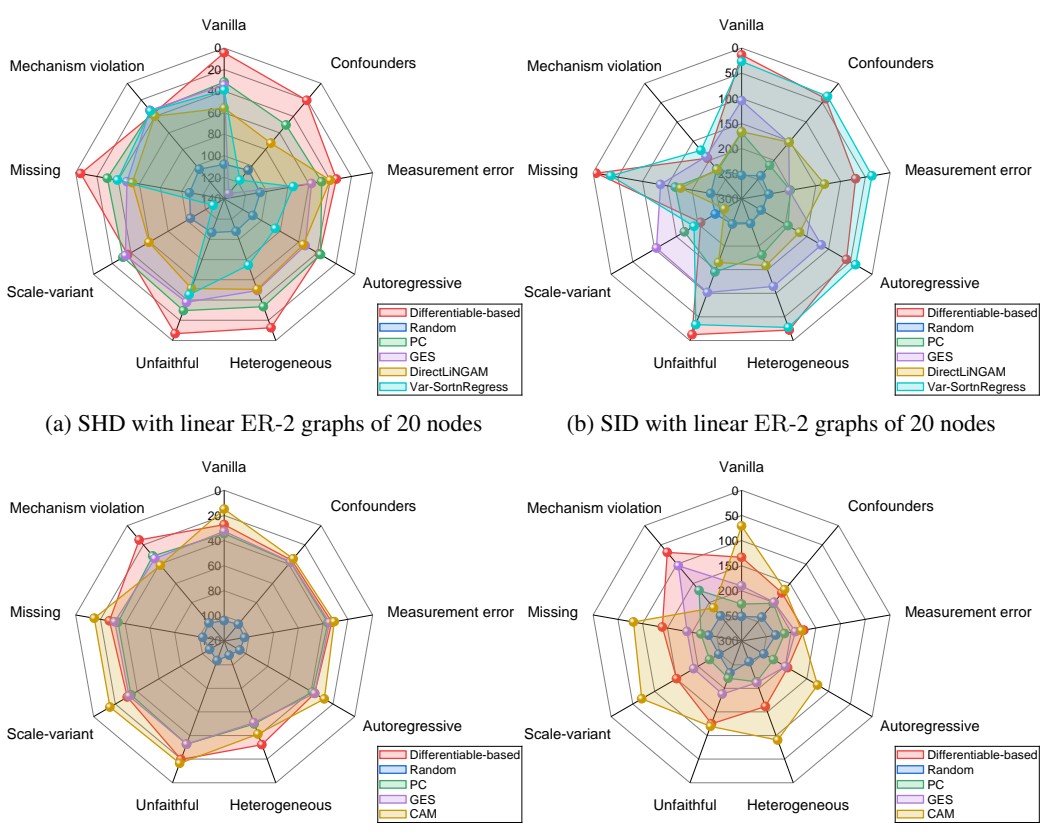

(a) SHD with linear ER-2 graphs of 20 nodes      (b) SID with linear ER-2 graphs of 20 nodes

(c) SHD with nonlinear ER-2 graphs of 20 nodes      (d) SID with nonlinear ER-2 graphs of 20 nodes

Figure 2: Experimental results under the linear and nonlinear ER-2 graphs of 20 nodes. SHD (the lower the better) and SID (the lower the better) are evaluated over 10 trials. For the differentiable causal discovery method, we present only the optimal results. As the nonlinear settings in Figure 2c and Figure 2d are more favorable to CAM, we conduct a more reasonable evaluation of CAM and differentiable causal discovery under the MLP setting (Section 4.1.1).

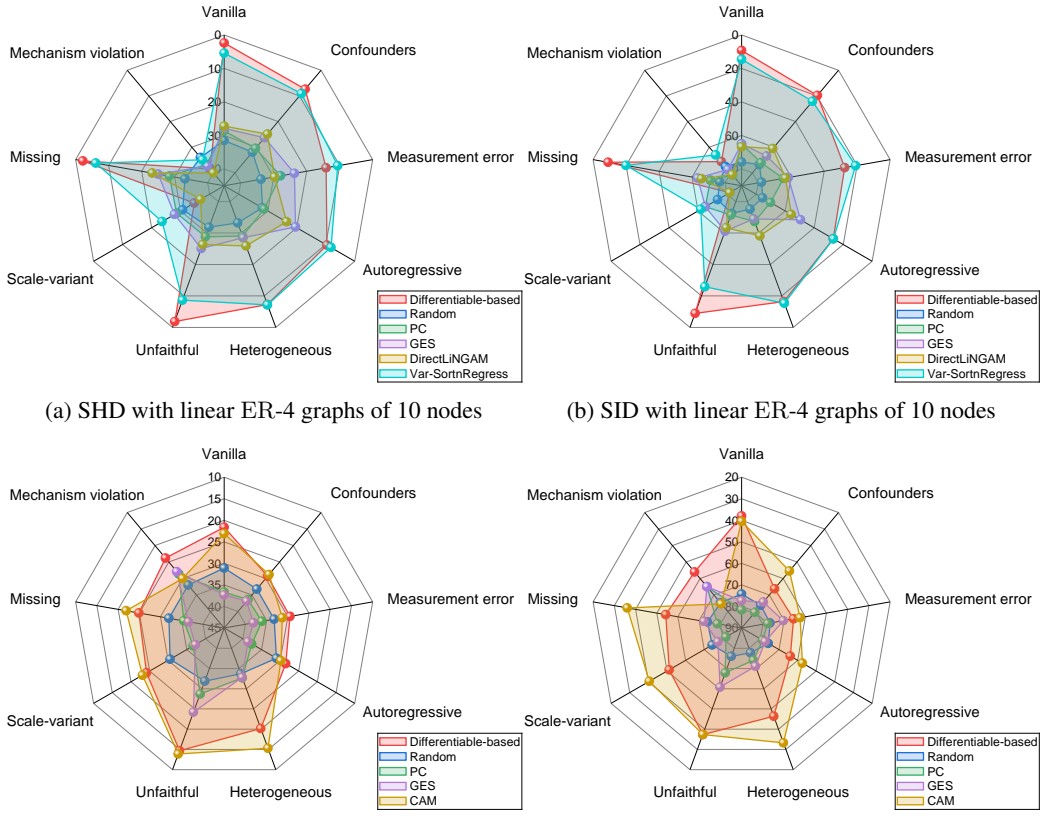

(a) SHD with linear ER-4 graphs of 10 nodes   (b) SID with linear ER-4 graphs of 10 nodes

(c) SHD with nonlinear ER-4 graphs of 10 nodes   (d) SID with nonlinear ER-4 graphs of 10 nodes

Figure 3: Experimental results under the linear and nonlinear ER-4 graphs of 10 nodes. SHD (the lower the better) and SID (the lower the better) are evaluated over 10 trials. For the differentiable causal discovery method, we present only the optimal results. As the nonlinear settings in Figure 3c and Figure 3d are more favorable to CAM, we conduct a more reasonable evaluation of CAM and differentiable causal discovery under the MLP setting (Section 4.1.1).

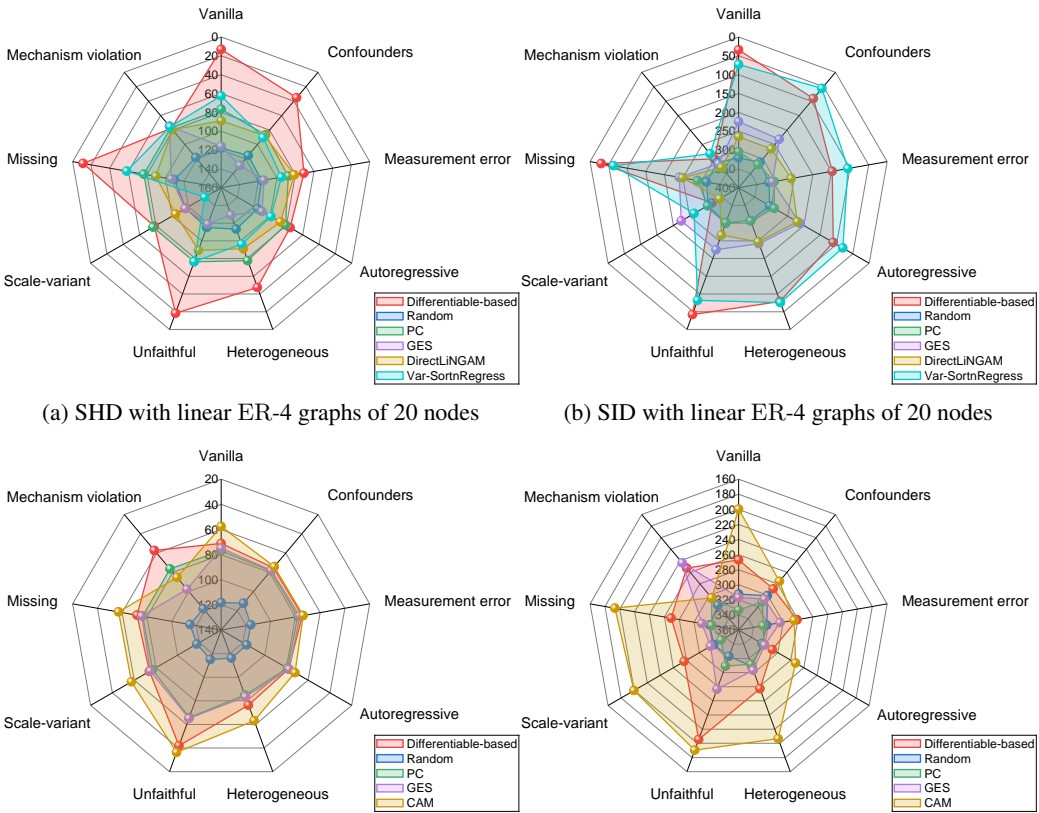

(a) SHD with linear ER-4 graphs of 20 nodes (b) SID with linear ER-4 graphs of 20 nodes

(c) SHD with nonlinear ER-4 graphs of 20 nodes (d) SID with nonlinear ER-4 graphs of 20 nodes

Figure 4: Experimental results under the linear and nonlinear ER-4 graphs of 20 nodes. SHD (the lower the better) and SID (the lower the better) are evaluated over 10 trials. For the differentiable causal discovery method, we present only the optimal results. As the nonlinear settings in Figure 4c and Figure 4d are more favorable to CAM, we conduct a more reasonable evaluation of CAM and differentiable causal discovery under the MLP setting (Section 4.1.1).

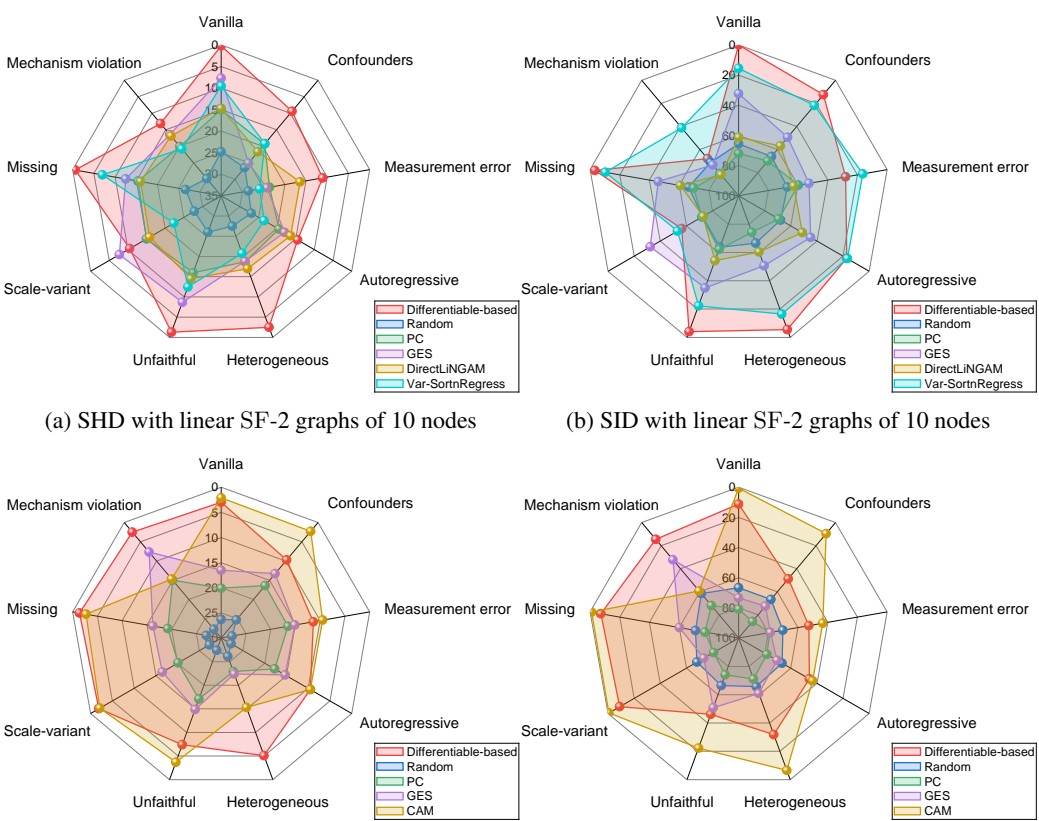

(a) SHD with linear SF-2 graphs of 10 nodes     (b) SID with linear SF-2 graphs of 10 nodes

(c) SHD with nonlinear SF-2 graphs of 10 nodes     (d) SID with nonlinear SF-2 graphs of 10 nodes

Figure 5: Experimental results under the linear and nonlinear SF-2 graphs of 10 nodes. SHD (the lower the better) and SID (the lower the better) are evaluated over 10 trials. For the differentiable causal discovery method, we present only the optimal results. As the nonlinear settings in Figure 5c and Figure 5d are more favorable to CAM, we conduct a more reasonable evaluation of CAM and differentiable causal discovery under the MLP setting (Section 4.1.1).

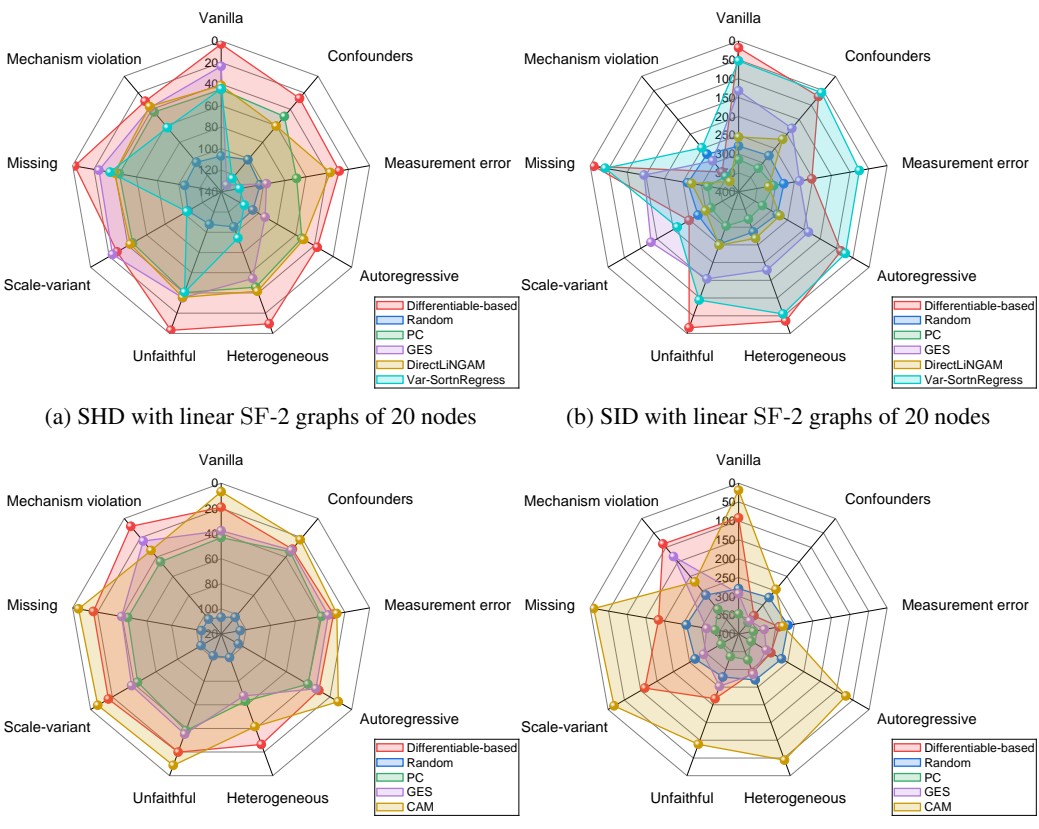

(a) SHD with linear SF-2 graphs of 20 nodes

(b) SID with linear SF-2 graphs of 20 nodes

(c) SHD with nonlinear SF-2 graphs of 20 nodes

(d) SID with nonlinear SF-2 graphs of 20 nodes

Figure 6: Experimental results under the linear and nonlinear SF-2 graphs of 20 nodes. SHD (the lower the better) and SID (the lower the better) are evaluated over 10 trials. For the differentiable causal discovery method, we present only the optimal results. As the nonlinear settings in Figure 6c and Figure 6d are more favorable to CAM, we conduct a more reasonable evaluation of CAM and differentiable causal discovery under the MLP setting (Section 4.1.1).

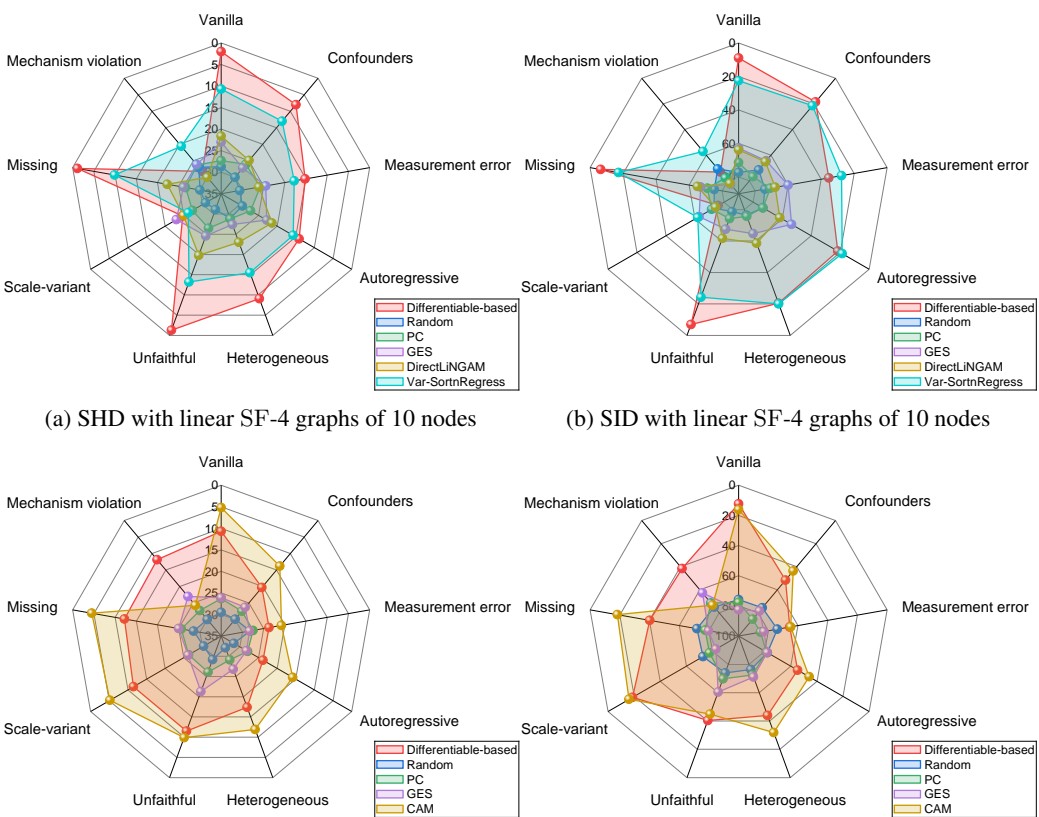

(a) SHD with linear SF-4 graphs of 10 nodes      (b) SID with linear SF-4 graphs of 10 nodes

(c) SHD with nonlinear SF-4 graphs of 10 nodes      (d) SID with nonlinear SF-4 graphs of 10 nodes

Figure 7: Experimental results under the linear and nonlinear SF-4 graphs of 10 nodes. SHD (the lower the better) and SID (the lower the better) are evaluated over 10 trials. For the differentiable causal discovery method, we present only the optimal results. As the nonlinear settings in Figure 7c and Figure 7d are more favorable to CAM, we conduct a more reasonable evaluation of CAM and differentiable causal discovery under the MLP setting (Section 4.1.1).

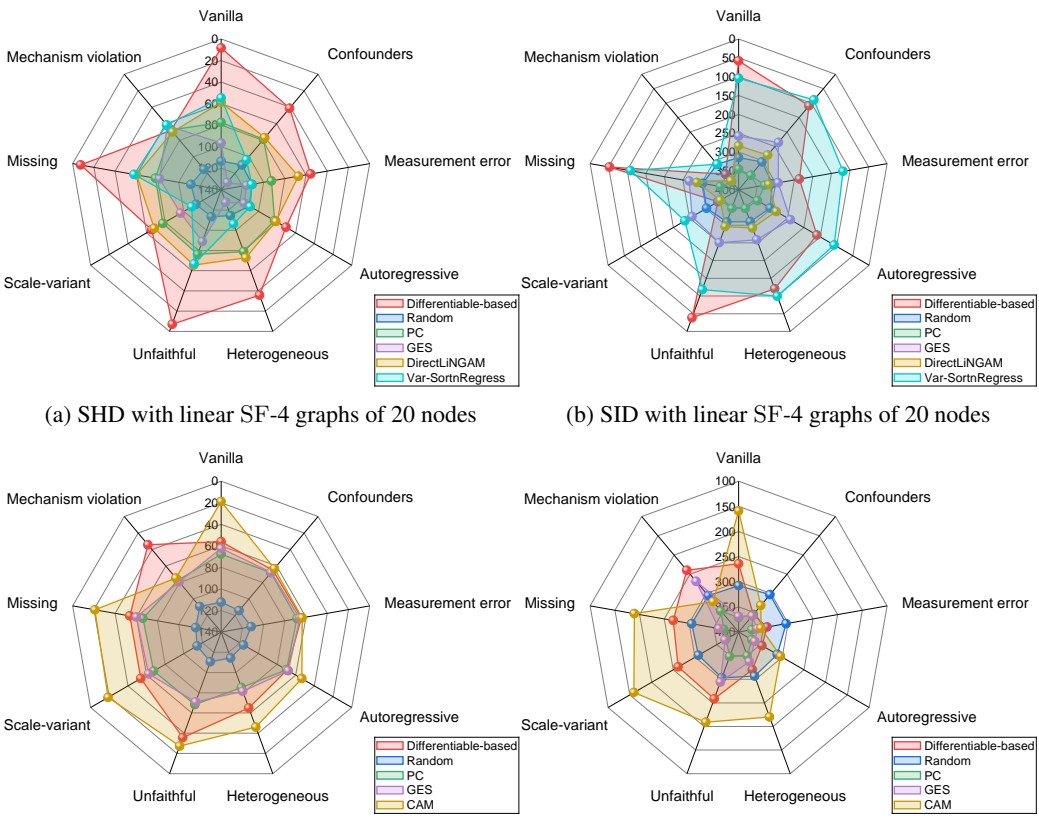

(a) SHD with linear SF-4 graphs of 20 nodes

(b) SID with linear SF-4 graphs of 20 nodes

(c) SHD with nonlinear SF-4 graphs of 20 nodes

(d) SID with nonlinear SF-4 graphs of 20 nodes

Figure 8: Experimental results under the linear and nonlinear SF-4 graphs of 20 nodes. SHD (the lower the better) and SID (the lower the better) are evaluated over 10 trials. For the differentiable causal discovery method, we present only the optimal results. As the nonlinear settings in Figure 8c and Figure 8d are more favorable to CAM, we conduct a more reasonable evaluation of CAM and differentiable causal discovery under the MLP setting (Section 4.1.1).

## K  TABLE RESULTS FOR EXTREME MEASUREMENT ERROR

Table 30 presents the results in the linear setting under measurement error with $\delta = 10$. The results in Table 30 indicate that when $\delta$ takes larger values, differentiable causal discovery methods fail to demonstrate robust performance. $\delta$ is used to control the variance of $\epsilon_i$ in (7). As $\delta$ increases, the noise ratio correspondingly increases, which leads to the loss of robustness in differentiable methods. Tables 2.1 and 9 present the results in the linear setting under measurement error with $\delta = 0.8$. The results in Tables 2.1 and 9 indicate that when $\delta = 0.8$ takes a smaller value, the noise ratio is correspondingly lower, allowing differentiable methods to demonstrate robust performance.

Table 30: Linear Setting under measurement error with $\delta = 10$, for ER-2 graphs of 10, 20 nodes.

| 10 nodes | Vanilla model | | Measurement error ($\delta = 10$) | |
| --- | --- | --- | --- | --- |
| | SHD↓ | SID↓ | SHD↓ | SID↓ |
| Random | 25.6±3.1 | 57.9±9.5 | 23.1±1.9 | 61.2±7.5 |
| PC | 12.4±3.1 | 40.9±13.4 | **19.2±2.1** | 56.9±9.3 |
| GES | 13.8±7.8 | 32.0±13.6 | 20.2±4.5 | **54.1±11.6** |
| DirectLiNGAM | 19.6±3.3 | 46.1±10.6 | 20.0±1.1 | 61.2±8.2 |
| DAGMA | **1.2±1.2** | **3.3±5.3** | 20.7±1.2 | 58.2±7.6 |
| 20 nodes | SHD↓ | SID↓ | SHD↓ | SID↓ |
| Random | 107.9±7.0 | 253.2±26.3 | 92.7±6.7 | 243.6±19.4 |
| PC | 31.6±6.5 | 168.5±27.6 | 44.5±4.6 | 213.8±24.5 |
| GES | 34.3±24.6 | 104.5±51.1 | 51.2±7.1 | 220.9±27.6 |
| DirectLiNGAM | 55.7±9.1 | 166.4±31.0 | **41.8±1.8** | **210.3±24.3** |
| DAGMA | **5.4±3.9** | **14.2±10.3** | 49.4±4.6 | 227.5±24.9 |

## L  TABLE RESULTS ON SEMI-SYNTHETIC DATA

The semi-synthetic data is generated based on the network structure of the real-world Sachs dataset, using linear and nonlinear vanilla models to create eight datasets with model assumption violations. The results in Table 31 and 32 indicate that differentiable causal discovery methods still achieve optimal or competitive performance in scenarios other than scale variation.

Table 31: Linear Setting, for semi-synthetic data of 11 nodes.

| 11 nodes | Vanilla model SHD↓ | SID↓ | Latent confounders SHD↓ | SID↓ | Measurement error SHD↓ | SID↓ | Autoregressive SHD↓ | SID↓ | Heterogeneous SHD↓ | SID↓ | Unfaithful SHD↓ | SID↓ | Scale-variant SHD↓ | SID↓ | Missing SHD↓ | SID↓ | Mechanism violation SHD↓ | SID↓ |
|---|---|---|---|---|---|---|---|---|---|---|---|---|---|---|---|---|---|---|
| Random | 34.0±2.0 | 44.1±6.9 | 32.7±1.4 | 42.9±5.3 | 36.2±3.1 | 47.2±5.8 | 33.7±4.2 | 46.8±5.7 | 38.4±3.7 | 51.6±4.3 | 36.8±3.7 | 50.3±3.9 | 33.5±1.8 | 43.2±6.5 | 39.1±2.8 | 57.3±8.2 | 35.9±4.8 | 56.7±7.4 |
| PC | 10.7±3.8 | 38.6±11.0 | 16.2±2.5 | 44.9±4.1 | 15.6±2.2 | 38.8±11.2 | 14.7±1.7 | 42.2±5.5 | 13.1±2.4 | 43.4±7.8 | 14.7±1.6 | 50.3±5.3 | 10.7±3.8 | 38.6±11.0 | 9.6±2.2 | 34.1±8.5 | 17.5±2.3 | 46.3±4.1 |
| GES | 9.4±2.7 | 28.9±6.3 | 28.7±7.4 | 33.8±12.0 | 17.0±3.5 | 29.2±14.5 | 14.5±3.9 | 22.5±14.5 | 15.5±3.9 | 27.4±9.2 | 13.7±3.4 | 35.5±8.4 | **9.3±2.7** | **28.7±6.2** | 7.9±2.9 | 26.5±8.7 | 19.2±3.3 | 35.5±11.4 |
| DirectLiNGAM | 12.8±4.2 | 34.5±11.5 | 18.9±4.3 | 39.6±6.9 | 14.8±3.2 | 41.0±6.0 | 12.2±2.5 | 35.6±11.2 | 14.5±4.4 | 42.5±9.7 | 11.3±3.7 | 35.7±8.2 | 14.3±4.0 | 43.6±8.5 | 12.3±4.1 | 37.1±7.5 | 16.8±2.8 | 40.9±6.1 |
| Var-SortnRegress | 3.8±3.4 | 7.8±8.3 | 22.6±7.0 | 12.7±7.2 | 13.9±1.7 | **11.3±8.3** | 12.6±3.7 | **8.5±9.2** | 11.3±4.6 | 8.2±8.2 | 5.9±1.9 | 15.0±5.7 | 13.1±4.6 | 38.5±6.9 | 2.9±2.0 | 7.7±6.8 | 18.0±2.6 | 23.4±7.1 |
| $R^2$-SortnRegress | 17.1±5.6 | 37.1±7.1 | 36.3±7.0 | 38.8±5.0 | 20.3±2.6 | 36.3±9.6 | 19.6±4.4 | 38.3±5.0 | 21.6±3.1 | 37.8±7.1 | 22.4±3.0 | 42.9±2.3 | 17.1±5.6 | 37.1±7.1 | 18.1±6.5 | 40.4±7.6 | 22.4±2.0 | 39.5±5.7 |
| NOTEARS | 0.5±0.7 | 5.1±6.3 | 9.3±1.6 | 29.0±6.2 | **8.9±2.4** | **18.3±7.5** | 11.5±4.4 | 11.3±8.8 | 3.4±1.9 | 10.5±7.1 | 0.4±0.9 | 3.2±6.5 | 12.1±3.6 | 44.3±9.2 | 1.1±1.6 | 5.2±7.5 | 14.6±2.4 | 31.7±5.0 |
| GOLEM | 0.7±0.3 | 6.8±5.4 | 9.3±2.6 | 26.7±3.1 | 11.3±4.9 | 25.7±5.9 | 15.0±2.8 | 37.7±5.4 | 5.7±5.2 | **2.7±3.8** | 0.7±0.9 | 2.3±3.3 | 13.0±1.4 | 46.3±3.4 | 1.5±0.8 | 7.2±4.3 | 17.0±0.8 | 52.3±0.5 |
| NoCurl | 0.3±0.6 | 2.8±5.9 | 12.0±3.9 | **9.8±7.2** | 9.4±2.3 | **18.3±8.0** | 11.7±4.3 | **9.2±7.5** | 3.1±2.3 | 6.1±5.1 | 0.5±0.9 | 4.6±7.1 | 12.2±2.4 | **43.0±8.4** | **0.5±0.7** | 4.1±5.4 | 22.8±3.5 | **22.8±6.1** |
| DAGMA | **0.2±0.4** | **2.8±5.6** | **8.9±2.8** | 25.9±10.3 | **8.9±2.4** | 19.1±8.3 | **10.9±3.9** | 12.9±10.7 | **2.9±1.9** | 6.3±6.0 | **0.1±0.3** | **1.4±4.2** | **11.5±2.8** | 43.7±10.1 | 0.6±1.0 | **2.4±3.7** | **13.9±1.9** | 28.0±6.4 |

Table 32: MLP Setting, for semi-synthetic data of 11 nodes.

| 11 nodes | Vanilla model SHD↓ | SID↓ | Latent confounders SHD↓ | SID↓ | Measurement error SHD↓ | SID↓ | Autoregressive SHD↓ | SID↓ | Heterogeneous SHD↓ | SID↓ | Unfaithful SHD↓ | SID↓ | Scale-variant SHD↓ | SID↓ | Missing SHD↓ | SID↓ | Mechanism violation SHD↓ | SID↓ |
|---|---|---|---|---|---|---|---|---|---|---|---|---|---|---|---|---|---|---|
| Random | 32.9±3.2 | 44.0±7.9 | 30.8±2.7 | 41.9±5.8 | 37.5±6.4 | 58.9±8.2 | 35.2±2.9 | 47.3±4.8 | 31.6±2.9 | 46.8±4.7 | 33.9±1.5 | 54.7±5.6 | 36.2±2.8 | 57.3±4.6 | 38.7±6.3 | 51.7±5.4 | 31.4±2.3 | 45.9±5.1 |
| PC | 17.5±2.3 | 46.3±4.1 | 18.8±3.9 | 48.6±4.7 | 18.4±1.9 | 37.1±10.7 | 17.1±2.1 | 44.4±4.8 | 20.9±2.9 | 44.3±6.1 | 18.2±2.6 | 45.7±5.4 | 17.5±2.3 | 46.3±4.1 | 16.8±1.5 | 44.6±4.8 | 10.7±3.8 | 38.6±11.0 |
| GES | 19.2±3.3 | 35.5±11.4 | 23.1±5.6 | 42.9±11.2 | 19.6±3.2 | 45.2±10.3 | 17.6±4.5 | 34.0±11.8 | 27.5±4.0 | 35.2±5.8 | 21.1±3.2 | 40.9±9.5 | 19.5±3.1 | 35.5±11.7 | 19.6±2.0 | 37.0±10.4 | 9.4±2.7 | 28.9±6.3 |
| CAM | 9.4±3.4 | 13.5±10.7 | 15.7±4.1 | 39.1±7.2 | 21.9±5.5 | 42.3±11.4 | 13.0±3.8 | 35.7±12.6 | 18.1±3.5 | 28.6±12.4 | 12.3±2.7 | 20.1±8.3 | **9.4±3.4** | **13.5±10.7** | 9.3±3.6 | 12.6±9.8 | 13.4±1.7 | 43.3±6.4 |
| NOTEARS-MLP | **6.8±2.9** | **10.1±7.5** | 10.4±1.6 | 38.2±7.3 | 15.4±1.2 | 46.5±3.9 | 15.9±4.1 | 39.9±5.2 | 13.9±2.9 | **20.7±9.1** | 10.0±2.9 | **17.3±8.3** | 16.5±2.3 | 46.5±9.8 | **7.3±2.4** | **7.2±5.8** | 5.2±1.7 | **22.2±5.7** |
| GraN-DAG | 9.1±2.6 | 33.8±10.0 | **10.0±2.6** | 40.9±8.8 | **13.0±2.1** | 42.0±3.9 | **10.5±2.5** | **33.3±7.5** | **9.1±1.8** | 32.6±7.7 | **10.0±1.5** | 28.1±4.3 | 13.1±1.5 | 47.1±6.7 | 9.8±2.7 | 35.7±10.1 | 10.6±2.5 | 40.0±7.3 |
| DAGMA | 8.7±2.7 | 11.1±5.1 | 11.9±1.7 | 41.0±5.6 | 15.8±2.6 | **34.7±3.8** | 16.0±1.5 | 40.5±8.9 | 16.2±2.7 | 34.7±8.4 | 13.2±2.4 | 21.5±8.5 | 16.7±2.2 | 43.5±4.8 | 9.0±2.8 | 12.1±5.9 | **5.0±1.4** | 22.6±4.2 |

