# OpenReview forum: "THE ROBUSTNESS OF DIFFERENTIABLE CAUSAL DISCOVERY IN MISSPECIFIED SCENARIOS"
_ICLR.cc/2025/Conference — ICLR 2025 Poster_

### Official Review · Reviewer_KM7a · 2024-10-25

**Soundness:** 1
**Presentation:** 3
**Contribution:** 2
**Rating:** 3
**Confidence:** 4

**Summary:**

The present paper provides an extensive benchmark on causal discovery with a focus on differentiable algorithms in misspecified scenarios. This is an interesting effort to better understand the behavior of these methods from an empirical perspective, as the previous benchmark in similar misspecified settings (Montagna et al., 2023) almost did not cover any of the differentiable methods tested in this paper.

**Strengths:**

The misspecifications and the differentiable algorithms subject to testing are well chosen, and the experimental setting is clearly explained. Section 4.1.2. provides interesting theoretical explanations about the empirical performance of differentiable-based methods in the case of a few misspecified scenarios.

**Weaknesses:**

## Main points
1. I don’t understand why the linear vanilla model is defined with Gaussian noise terms, which are notably non-identifiable. This may be an interesting choice in the light that non-standardized SCMs may still be identified in fact by differentiable causal discovery methods, even in these setting - e.g. exploiting equal variance effects, as studied in Peters et al., 2012 (Identifiability of Gaussian structural equation models with equal error variances). Yet, it is limiting that the only linear setting is that with Gaussian noise, due to the very well-known identifiability limitations affecting almost every algorithm in the list (with the exception of PC and GES).
2. An additional, strong limitation of this work, is that it limits the benchmarking to non-standardized data: this is problematic specifically when testing occurs on differentiable-based algorithms, which notoriously degrade their performance when variance does not increase in the causal order. The authors’ experimental design choice, then, undermines most of the conclusion that one might draw from the experimental outcomes: looking at the linear Gaussian experiments (non-identifiable), the competitive performance of differential-based methods is clearly symptomatic that they are using the variance-induced *shortcuts* in non-standardized data (which is not surprising, as the authors seem to be aware given that they cite Reisach et al., 2021 and Ng et al., 2024 as references) (see the discussion on noise ratio in section 4.2.1 to understand what I mean by *shortcuts*). In light of the good performance of differentiable-based methods on vanilla linear data, all the conclusions that one can derive from these experiments boil down to the following question: given a misspecified scenario, does the noise ratio (discussed in section 4.2.1) ensure that the optimal solution for a differentiable based methods is the correct one? But this is not testing these algorithms’ robustness, it is more testing the properties of the misspecified datasets. Hence, conclusions of *counter-intuitive robustness* (L19) of differentiable based methods in the misspecified settings are wrong (in the sense that they are not surprising if the noise ratio allows it, as in the vanilla linear case). Moreover, they are partial, given that benchmarking occurs on a very limited class of SCMs, those with cumulating variance along the causal order, which we know to enable causal discovery with differentiable based methods. Testing should be performed on the class of standardized causal models, to gain meaningful conclusions about robustness.
3. The SHD outcomes of Tables 2 and 3 are odd/off: e.g. in reference to the R2-sortnregress line of table 2, 50 nodes, you have SHD values of 388, 100, 346, 552, 549, 442, … How is this possible? In SHD, each error (edge addition, removal, or direction flip w.r.t. ground truth) counts as one, or counts as two if the direction flip is counted twice (because we have both a false positive and a false negative): in the latter case, for a fully connected graph with 50 nodes and 2 edges per nodes, the maximum achievable SHD is 200. Hence, I believe something fishy is going on with the SHD values proposed in the table (SHD beyond the possible bound appears all over Table 2 and Table 3). Could you explain this?
4. In Table 2 the latent confounding columns SHD values have a weird behavior: it looks like all methods simply output an empty graph, i.e. for ER-2 graphs with 10 nodes, an empty graph gives you SHD 20, for 20 nodes you get SHD 40, and 50 nodes you get SHD 100: these are exactly the values you get for almost any run, which is suspicious, also in the light of results from Montagna et al., 2023, which already tested the nonlinear confounded setting, and whose result disagree with yours (in their paper there is some robustness displayed by some methods under confounded setting, while no algorithms seem to give the results agreeing with what the authors show). Comments on this point are needed.
5. L343: authors say that on nonlinear data differentiable CD achieves optimal (what do you mean by optimal?) or competitive performance, whereas this is very bad assuming that SHD is bounded by the number of edges, which should be the case according to the description of SHD in section 3.4. One example (which generalizes to other nonlinear cases): on the vanilla model, 10 nodes, the differentiable methods achieve SHD around 10, on 20 nodes SHD around 30, on 50 nodes SHD between 70 and 85, which are all very bad. Hence, the analysis of the authors is incorrect.
6. Authors only present pointwise SHD, without any deviation from multiple experiments or confidence interval, which should instead be provided

## Minor points
1. Reading Tables 2 and 3 is hard, as there are many lines packed in a small space, in a small font. Consider moving it to the appendix?
2. In the introduction, I believe that authors are unnecessarily overselling the value of differentiable causal discovery methods, or at least they don’t seem objective: they define differentiable causal discovery a *ground breaking* *advancement* (L47), whereas in my direct and undirect experience (readings) they are not groundbreaking in the sense they don’t perform generally better than traditional methods or many other approaches to causal discovery. Later, they talk about their *transformative potential in the realm of causal inference* (L53), which is also unsupported by evidence, to the best of my knowledge.
3. Again in the introduction, they say that causal sufficiency and absence of measurement error are *generally imperative* (L54) and they are *fundamental prerequisites for guaranteeing validity and reliablility of the causal inference* (L56-57). They are not imperative, nor fundamental prerequisites: these assumptions are not necessary for identifiability, as methods for causal discovery under confounding effects and measurement error exist, as the authors themselves mention.

   In reference to this and the previous point: using expressions like *immense potential, imperative need, etc.,* makes unnecessarily strong statements, I would advise against it.
4. L134: the citation to Lachapelle, 2019 (GRAN-DAG paper) when introducing identifiability is wrong. They just propose an algorithm and no identifiability statement. I would remove it, the citations coming after are those that are on point.
5. The paper claims that Montagna et al., 2023 misspecifications are limited in scope, yet 6/9 scenarios covered here (e.g. those in the graphs of Figure 1) are covered there.

**Questions:**

1. Why did the authors limit their theoretical analysis of section 4.2.1 to measurement error and unfaithful datasets? I believe that in the light of the theoretical analysis in Ng, 2024 (Structure Learning with Continuous Optimization: A Sober Look and Beyond), it would be of great interest if this theoretical study was carried over all the scenarios. This is not a criticism to the paper, but a suggestion of what I believe would be a good contribution in connection to this work.
2. L64: the authors say *true mechanisms remain unclear when applied to real data*. This sentence is obscure to me, could they clarify?

---

> ### Author Response · Authors · 2024-11-21
> **Response to Reviewer KM7a (Part I)**
>
> ### Main points
>
> > **W1: Linear Gaussian model with equal error variances is identifiable**
>
> **A:** Thanks for comment. We agree that linear model with Gaussian noise is non-identifiable. It is worth clarifying that the linear vanilla model in the main text is defined using **Gaussian noise with equal noise variance (lines 205-206)**. This setup is chosen because linear Gaussian with equal noise variance **satisfies identifiability** (Peters et al., 2012), aligning with the assumptions of most linear methods in Table 1, except DirectLiNGAM, and facilitating a fair comparison among these methods.
>
> Additionally, we have supplemented experiments on the linear vanilla model with **non-Gaussian noise** in Table 17 and Table 18 (Appendix G). Experimental results show that under non-Gaussian noise, differentiable causal discovery still demonstrates optimal or competitive performance in scenarios except scale variation. Our supplementary experiments confirm that non-Gaussian noise does not affect our main conclusions.
>
> > **W2.1: Clarification on standardized data**
>
> **A:** Actually, **standardized data scenario**, **noted as scale variation** in our work, is one of the eight tested misspecified scenarios in the original version of the manuscript. Thus, we have already conducted experiments on standardized data. Specifically, the **scale-variant model** described in the main text (line 230) refers to the **standardized data** scenario as discussed in Reisach et al. (2021). Furthermore, in the abstract (lines 17–20), we have explicitly stated that differentiable methods exhibit robustness in scenarios other than scale variation.
>
> > **W2.2/Q1: Discuss noise ratio**
>
> **A:** The noise ratio (Ng et al., 2024) can **only provide theoretical guarantees for the performance of linear differentiable methods on linear data**. However, it is **not applicable in the four tested scenarios**, i.e., latent confounding, autoregression, heterogeneity, and mechanism violation. Additionally, it does not extend theoretical guarantees to **nonlinear differentiable methods**. Below are specific limitations of the noise ratio in these scenarios:
>
> - Latent confounding: Not all variables in the data are observable, violating the assumption of availability of all variables.
> - Autoregression and heterogeneity: The assumption of independent and identically distributed (i.i.d.) data is violated, making the noise ratio inapplicable.
> - Mechanism violation: Data is generated by nonlinear models, which lies outside the scope of noise ratio theory.
>
> Given these limitations, **the noise ratio theory cannot be applied to these scenarios.** Therefore, our work, which experimentally investigates the robustness of linear differentiable methods in these challenging settings, is both meaningful and valuable. Additionally, the noise ratio also fails to provide theoretical guarantees for nonlinear differentiable methods, emphasizing the necessity of benchmarking these methods empirically.
>
> For scenarios such as measurement error, unfaithfulness, missing data, and scale variation, the noise ratio can theoretically support the performance analysis of linear differentiable methods. In the previous version of the manuscript, we have just analyzed the measurement error and unfaithfulness scenarios in Section 4.1.2. In the latest version, we have also added an analysis of the noise ratio in the context of missing data (lines 481-484). **Establishing a connection between the noise ratio and the performance of linear differentiable methods** under model assumption violations is one of the solid contributions of our work. Even though, empirical evaluations remain indispensable to validate theoretical insights.
>
> In **real-world application scenarios**, the theoretical assumptions underlying the noise ratio are often not satisfied, making it **not applicable** for analyzing the performance of linear differentiable methods. The model assumption violation scenarios are commonly encountered in practice, and evaluating the performance of various methods under these conditions offers insights into their real-world applicability. Thus, benchmarking the performance of different methods under model assumption violations is both crucial and meaningful.
>
> > **W2.3: Delete descriptions**
>
> **A:** Considering that the noise ratio can explain the performance of linear differentiable methods in some misspecified scenarios, we have revised the corresponding wording by **deleting counter-intuitive** in the abstract (line 18).

---

> > ### Author Response · Authors · 2024-11-21
> > **Response to Reviewer KM7a (Part II)**
> >
> > > **W3/W5: Computation of SHD**
> >
> > **A:** Thanks for the comment. Actually, the idea that the metric, SHD, is bounded by the number of edges in the ground-truth DAG is not correct. SHD counts the number of edge insertions, deletions, and reversals necessary to transform the estimated DAG into the true DAG.
> >
> > $$SHD=∣Insertions∣+∣Deletions∣+∣Reversals∣$$
> >
> > For example, given a true DAG with 50 nodes and an average degree of 2, there are 100 edges. The maximum number of edges for a DAG with $n$ nodes is $n(n−1)/2$. For a 50-node estimated DAG, the maximum edge number is 1225. It is clear that in this case, the SHD value can far exceed 200. The results we present in the table are reasonable.
> >
> > > **W4: Check latent confounder setting**
> >
> > **A:** Thanks for the suggestion. We have carefully checked the data generation process, and found that the previous open-source code (Montagna et al., 2023) regarding one of the eight considered misspecified scenarios, i.e., latent confounders, had some bugs. Thus, we have corrected the corresponding data generation process and re-conducted the experiments accordingly. The updated results, presented in Tables 2~Table 21, further facilitate the conclusions on the robustness of differentiable methods.
> >
> > > **W5: Clarify SHD and explain optimality**
> >
> > **A:** Since we have explained the concept of SHD in the above answer to **W3/W5**, the results we present in the tables are reasonable. Optimality refers to the results with the smallest metrics. Competitive results refer to the performance that is either the second or close to the optimal one. In the nonlinear setting, the vanilla model adopts a Gaussian process mechanism, which aligns closely with CAM's assumption of SEM, thereby favoring CAM in the evaluation. We provide a further discussion of CAM's performance compared to differentiable causal discovery in Section 4.1.1 (Table 4). Overall, differentiable causal discovery methods are either optimal or competitive.
> >
> > > **W6: Add standard deviation**
> >
> > **A:** Thanks for the suggestion. In the previous version, we only reported the mean of the metrics over 10 trials. In the new manuscript, we have added the standard deviation of the metrics over 10 trials in Table 2~Table 21. The inclusion of standard deviations enhancing the understanding of the robustness of our results.
> >
> > ### Minor points
> >
> > > **W1: Improve presentation**
> >
> > **A:** Thanks for the comment. Since Table 2 and Table 3 show the main results of all the benchmark methods on linear and nonlinear ER-2 graphs with 10, 20, and 50 nodes, they are closely placed after the description of the results. The data in these tables are crucial for understanding the main findings and conclusions of the paper, and thus they are retained in the main text for easy reference and direct association with the discussion in the paper. It would better improve the readability and convenience for readers. However, since the font is a bit small, we sincerely suggest the readers use zoom-in and zoom-out operation when reading the PDF file of the manuscript.
> >
> > > **W2/W3: Improve descriptions**
> >
> > **A:** Thanks for the valuable suggestion to improve the presentation. The description has been revised thoroughly to ensure a more accurate and objective evaluation of these methods. The sections that contained the somewhat inaccurate expressions have been amended and are marked in the main body of the text (lines 47-54, 88-89).
> >
> > > **W4: Delete citation**
> >
> > **A:** Deleted with many thanks.
> >
> > > **W5: Clarify Scenarios**
> >
> > **A:** Thanks for the comment. We have added four new scenarios that were not considered in the previous work by Montagna et al. (2023): heterogeneity, scale variation, missing data, and mechanism violation (lines 218-221). We adopt four of the six scenarios from the previous work: latent confounders, measurement error, autoregressive, and unfaithfulness. We do not include the PNL and LiNGAM models, because their vanilla models rely solely on nonlinear Gaussian processes, which are unfair for evaluating linear methods. In our study, we considered both linear and nonlinear vanilla models, facilitating a more fair evaluation of both linear and nonlinear methods.
> >
> > > **Q2: True mechanisms in real-world datasets**
> >
> > **A:** The true mechanism refers to the physical process of real-world data generation. In practical applications, causal discovery algorithms are typically based on causal assumptions. However, due to the lack of knowledge about the underlying physical mechanism of real-world data, we cannot ensure that the causal assumptions of the algorithms are satisfied in real-world scenarios. Thus, true mechanisms remain unclear when applying causal discovery algorithms to real data.
> >
> >
> >
> > **Overall, we hope our responses address your concerns, and sincerely appreciate higher positive evaluation scores.**

---

> > > ### Comment · Reviewer_KM7a · 2024-11-22
> > >
> > > I thank the authors for addressing several concerns and points that were confusing me. Still, I believe some important weaknesses and points worth discussion remain.
> > >
> > > ### Major points
> > >
> > > - **W2.1: Clarification on standardized data**: the wording I use in the review is imprecise, particularly saying that the authors *limit* the benchmarking to non-standardized data. I apologize; what I meant is that on all the tested scenarios, but one, data are non-standardized. In this light, I stand by the point that I made: how can you ensure that in all the non-standardized misspecified scenarios, the good performance you notice is due not to some robustness property of the algorithms, but instead to varsortability or other artifacts that we know methods for differentiable causal discovery exploit? For example, say you consider the *measurement error scenario*. To ensure robustness, you should have *standardized* data, generated under measurement error.
> > > - **W4**. Could the authors be more specific about the bug? Have they updated the text? I don't see blue notes in the updated version of the manuscript in the section where they comment on results on confounded scenarios, yet the results have drastically changed.
> > >
> > > ### Minor points
> > >
> > > - **W1**. I am aware of the zooming option. Still, presenting with a font that is not even close to being readable on printed paper is a poor presentation choice. I would suggest the following two options:
> > >     - Avoid presenting *all* results, and limit yourselves to e.g. 20 nodes (something that is representative of what happens at different scales). Use the additional space you gain from that to better space the columns, and hence increase the font size
> > >     - Use plots, which in any case better convey the message.
> > >
> > > ### Questions
> > > **Q2.** As it is currently phrased, in the sentence "the true mechanisms remain unclear when applied to real data," the subject of *applied* is *the true mechanisms*. Instead, for the rephrasing you proposed in the rebuttal, it is clear that the subject of *applied* is *the causal discovery algorithms*. I suggest sticking to the "rebuttal" version.

---

> ### Author Response · Authors · 2024-11-23
> **Response to Reviewer KM7a**
>
> ### Main points
>
> > **W2.1: Clarification on standardized data**
>
> **A:** Thanks for the comment. Actually, the idea that the performance of linear differentiable causal discovery methods is determined by the varsortability in the data, rather than the robustness of the algorithm itself, is not correct.
>
> The noise ratio theory from Ng et al. (2024) has already proven that **the performance of linear differentiable methods is determined by the least squares score function used by the algorithm itself, not by the varsortability in the data**. The noise ratio theory states that, due to the use of the least squares score function, linear differentiable methods return the optimal solution as long as the noise ratio variation is sufficiently small. The recent work by Deng et al. (2024) [1] further demonstrated that linear differentiable methods can achieve scale invariance when the correct scoring function is employed.
>
> In our work, we have established a connection between the performance of linear differentiable methods under misspecified scenarios and the noise ratio theory. Therefore, the robust performance of linear differentiable methods in misspecified scenarios is determined by the scoring function of the algorithm itself, not by the varsortability in the data.
>
> As the previous work (Ng et al., 2024) has already shown that linear differentiable methods fail to demonstrate robustness on standardized data due to their use of the least-squares scoring function, it is unnecessary to conduct additional standardized data experiments for each misspecified scenario. For example, in scenarios involving measurement error and scale variation, linear differentiable methods simultaneously violate two model assumptions, whereas other types of methods violate only one. This makes the evaluation of differentiable methods under such conditions unfair.
>
> Based on these analysis, our conclusion that differentiable causal discovery methods exhibit robust performance in scenarios other than scale variation is valid.
>
> [1] Deng C, Bello K, Ravikumar P, et al. Likelihood-based Differentiable Structure Learning. arXiv preprint arXiv:2410.06163, 2024.
>
> > **W4: Check latent confounder setting**
>
> **A:** In the open-source code provided by Montagna et al. (2023), available at https://github.com/francescomontagna/causally/blob/main/causally/scm/scm.py, the erroneous code is found at line 115:
>
> ```python
> graph_order = topological_order(self.adjacency)
> ```
>
> We have corrected the code as follows:
>
> ```python
> if "confounded" in list(self.scm_context.assumptions):
>     graph_order = topological_order(adjacency)
> else:
>     graph_order = topological_order(self.adjacency)
> ```
>
> It can be seen that in the confounded scenario, the input to the `topological_order()` function should be `adjacency` instead of `self.adjacency`.
>
> In the previous version of the manuscript, the results for confounding scenarios were incorrect, leading to an erroneous interpretation of these results. In the latest version of the manuscript, we have removed the incorrect interpretation of the confounding results and marked the corrected text with blue annotations (lines 314-316).
>
> ### Minor points
>
> > **W1: Improve presentation**
>
> **A:** Thanks for the suggestion. In the new manuscript, we present the results for 10 nodes cases in the main text (Table 2.1, 2.2, 3.1, 3.2, 4.1 and 4.2), while the other results are included in the Appendix E for reference. Additional figure results are provided in Appendix J for reference.
>
> > **Q2: Improve descriptions**
>
> **A:** Modified with many thanks (line 61).
>
>
>
> **Overall, thank you for your valuable comments. We hope the detailed clarifications provided above adequately address your concerns. If you have any further questions, please feel free to reach out, and we would be glad to provide additional explanations. We kindly invite you to reconsider your evaluation in light of these responses.**

---

> > ### Comment · Reviewer_KM7a · 2024-11-24
> >
> > I thank the authors for engaging in this discussion, which I hope proves beneficial to the paper. I will focus on the point relative to data standardization.
> >
> > Considering the noise ratio theory mentioned by the authors, let’s focus on the experiments on linear data, as the theory primarily applies to this setting. The following observations hold:
> >
> > 1. The authors' experiments show a correlation between non-standardized data (high varsortability) and good performance of differentiable methods (low noise ratio)—as seen in the *vanilla* SHD results. Conversely, standardized data correlate with poor performance (high noise ratio)—evident in the *scale-variant* results.
> > 2. Reisach et al., 2021 (or 2024) experiments provide the same empirical evidence.
> >
> > Now, consider the measurement error scenario, for which authors claim robustness of differentiable methods. According to theory, this is explained by the use of synthetic data with low noise ratios. If data with high noise ratios were sampled, conclusions could be reversed while still aligning with empirical evidence. In light of these considerations, my point is that the authors' experimental design, in particular the synthetic data sampling, may be unintentionally biased in favor of differentiable causal discovery. In absence of experiments with measurement error (or any other misspecified scenarios) *and* standardization, we can not exclude this possibility, as we know of the strong correlation between SHD and non-standardization. Thus, the experiments do not sufficiently support sharp claims of robustness under measurement error, as those made in this paper.
> >
> > For other scenarios (e.g., autoregressive), where no clear connection to noise ratio theory exists, I would also urge caution. Good performance may stem from the specific properties of the synthetic data that enhance the correlation between standardization and inference accuracy. The authors sampled their data from unstandardized SCMs with sometimes good varsortability (often better than random, better than PC and GES, sometimes with best SHD). It is known that this correlates with good performance of differentiable causal discovery (as discussed in the beginning of the response). This supports that the authors' experimental design could be favorable to differentiable causal discovery. To exclude this, further experiments combining each misspecified scenario with standardization (e.g *autoregressive + standardization)* are necessary to derive reliable conclusions.
> >
> > After the discussion, I am more convinced of the value of the paper's empirical evaluation, but unfortunately, in light of all these considerations, I still believe that the experimental setting is too limited to support the authors' conclusions.

---

> > > ### Author Response · Authors · 2024-11-28
> > > **Response to Reviewer KM7a**
> > >
> > > We would like to express our sincere gratitude to the reviewer for taking the time to read and review our manuscript, and for providing valuable and constructive feedback. Your comments have been incredibly helpful in guiding our revisions and improving the quality of our work.
> > >
> > > In order to express the conclusion more objectively, we have added new experiments (Table 30). The results in Table 30 indicate that differentiable methods do not always perform well or exhibit robustness. In the new manuscript, we have confined the robustness conclusions to commonly used misspecified scenarios to convey our findings more objectively and accurately (lines 19, 106, 311, 533).
> > >
> > > We hope that these revisions and clarifications adequately address your concerns. Should you have any further questions or need additional information, please feel free to reach out, and we would be happy to provide further details or make any necessary adjustments. We would appreciate it if you could reconsider your rating.

---

> ### Author Response · Authors · 2024-11-25
> **Response to Reviewer KM7a**
>
> ### Main points
>
> > **W2.1: Clarification on standardized data**
>
> **A:** Thanks for the comment. **It is worth clarifying that previous work by Ng et al. (2024) has already demonstrated that high varsortability in the data does not necessarily reflect the good performance of differentiable methods**. The **noise ratio theory** proposed by Ng et al. (2024) provides a better explanation for the performance of differentiable methods, but **not varsortability**. Therefore, subsequent analyses will use the noise ratio theory instead of varsortability.
>
> The noise ratio theory proves that because differentiable methods use a least-squares scoring function, they can return the optimal solution for data with low noise ratio, leading to the robust performance. This **robustness is attributed to the least-squares scoring function employed by the method itself rather than the varsortability properties of the data**.
>
> We agree with the reviewer that **differentiable causal discovery does not always demonstrate robustness across all measurement error synthetic data configurations (lines 533-536)**. Thus, we aim to highlight that under **commonly used synthetic data** configurations for measurement error, differentiable methods can exhibit robust performance. In measurement error scenario considered by Montagna et al. (2023), the observed variables are:
>
> $$\tilde{X}_i=X_i+\epsilon_i, \forall i=1, \ldots, d,$$
>
> where $X_i=f_i(X_{p a(X_i)}) + U_i$, $f_i$ is a linear mechanism, $U_i \sim N(0,1)$, $\epsilon_i \sim N(0,\delta*\operatorname{Var}(X_i))$.
>
> In the main text, we adopt the same synthetic data configuration ($\delta = 0.8$) as Montagna et al. (2023). Even if this configuration might unintentionally favor differentiable causal discovery, it does not undermine our conclusion about the robustness of differentiable methods under commonly used misspecified synthetic data.
>
> **Indeed, differentiable methods do not always perform well and robustly**. We have added **new experiments** under measurement error scenarios with $\delta = 10$. The results in Table 30 (Appendix K) show that differentiable methods fail to demonstrate robust performance in measurement error scenarios with larger $\delta$, as the noise ratio correspondingly increases with $\delta$. **The noise ratio theory indicates that differentiable methods perform robustly on low noise ratio data ($\delta = 0.8$) but fail to perform well when the noise ratio exceeds the robustness threshold**, as observed in high noise ratio data ($\delta = 10$).
>
> **In the scenario combining both measurement error and standardized data, it is not possible to fairly evaluate various methods**. In such cases, differentiable methods violate these two model assumptions, while other methods violate only one, i.e., measurement error, since other methods will not be influenced by standardized data. Reasonable robustness conclusions should be drawn under conditions where all methods simultaneously violate a single model assumption.
>
> Based on the above analysis, our conclusion in the paper regarding the robustness of differentiable methods under commonly used misspecified scenarios (except for scale variation) is valid.
>
>
>
> **Overall, thank you for your insightful comments. We agree with the reviewer that differentiable methods do not always perform robustness. In the new manuscript, we have restricted the robustness conclusions to commonly used misspecified scenarios to convey our findings more objectively and accurately (lines 19, 106, 311, 533). We hope the explanations provided above effectively address your concerns. If you have any additional questions or require further details, please feel free to contact us. We kindly invite you to reconsider your evaluation in light of the explanations shared in our response.**

---

> > ### Comment · Reviewer_KM7a · 2024-12-01
> >
> > I fear the provided experiments in Table 30 are not convincing: in that case, you control the noise ratio by extreme values of the measurement error. What I intend, is to consider datasets with the same "common" measurement error values you already have in the main paper, at different noise-ratio values. This is what would clear out the doubt whether the observed results are due only to a synthetic data design that is favorable to differentiable-based causal discovery, or not.
> >
> > In light of all the discussions we had, I decided to retain my score.

---

> > > ### Author Response · Authors · 2024-12-02
> > > **Response to Reviewer KM7a**
> > >
> > > ### Main points
> > >
> > > > **W2.1: Clarification on standardized data**
> > >
> > > **A:** Thanks for the comment. It is worth noting that the idea that **the good performance of the differentiable methods is due to the specific attributes of the synthetic data is not correct**. Based on this incorrect idea, **the suggested further experiment of combining each misspecified scenario with standardization (e.g., measurement error+ standardization) to eliminate the impact of varsortability is also not correct**.
> > >
> > > The key reason is that noise ratio theory proposed by Ng et al. (2024) demonstrates that the good performance of differentiable methods is **due to the least squares scoring function used by the method itself**, which leads to robustness for low noise ratio data, and the good performance **is not because of the varsortability property of the data**. The introduction of Deng et al. (2024) [1] also clearly points out this view, and through the selection of an appropriate scoring function, achieved scale invariance for differentiable methods.
> > >
> > > The noise ratio theory also indicates that in the standardized scenario, the noise ratio exceeds the threshold of robustness for differentiable methods, leading to a decline in performance (see experiments with different values of noise ratio, δ = 0.8 and δ = 10). Therefore, **the experiment in the measurement error and standardization combined scenario is essentially not about excluding the effect of varsortability on performance, but rather about exceeding the robustness threshold of the differentiable methods.** The experiment of combining each misspecified scenario with standardization (e.g., measurement error+ standardization) to eliminate the impact of varsortability is not correct. We need to analyze the performance of differentiable methods using the noise ratio theory, rather than using the incorrect varsortability.
> > >
> > > **Overall, thank you for your comments. We hope our responses address your concerns. If you have any additional questions or require further details, please feel free to contact us. We kindly invite you to reconsider your evaluation in light of the explanations shared in our response.**
> > >
> > > [1] Deng C, Bello K, Ravikumar P, et al. Likelihood-based Differentiable Structure Learning. arXiv preprint arXiv:2410.06163, accepted in NIPS 2024.

---

### Official Review · Reviewer_gPsY · 2024-10-28

**Soundness:** 3
**Presentation:** 4
**Contribution:** 3
**Rating:** 8
**Confidence:** 3

**Summary:**

This paper provides a comprehensive study of the performance of various causal discovery algorithms under eight types of model assumption violations. Notably, the differentiable causal discovery algorithm demonstrates the greatest robustness across different metrics. The paper also includes a degree of theoretical discussion regarding the performance of differentiable causal discovery methods.

**Strengths:**

This paper presents an extensive empirical study of twelve leading causal discovery methods across eight scenarios involving model assumption violations. The benchmark provides strong evidence for the robustness of differentiable DAG learning algorithms.

**Weaknesses:**

* The paper does not include experiments on real-world data. Since the focus is on evaluating various DAG learning algorithms under assumption violations, real-world data would be a natural and valuable addition. Including real-data applications could strengthen the study’s relevance.

* Some algorithms are challenging to train and may require longer training times, which could partly explain their strong performance. A comparison of running times across algorithms would help clarify this aspect.

* Including the standard error of each result would improve the analysis by indicating the stability of each algorithm across different datasets.

**Questions:**

* Recent work [1] has shown that methods like NOTEARS and DAGMA can achieve scale invariance if the correct loss function is applied and optimization reaches the global optimum. In this paper, differentiable causal discovery performs poorly on scaled datasets; however, this recent work [1] suggests that such issues can be addressed. I believe this insight adds valuable perspective on differentiable causal discovery and warrants an appropriate discussion in the paper.



[1] Deng, C., Bello, K., Ravikumar, P. and Aragam, B., 2024. Likelihood-based Differentiable Structure Learning. arXiv preprint arXiv:2410.06163.

---

> ### Author Response · Authors · 2024-11-21
> **Response to Reviewer gPsY**
>
> > **W1: Results on real-world datasets**
>
> **A:** To investigate the performance on real-world datasets, we have incorporated an analysis of **the real-world Sachs dataset**, a recognized benchmark in causal discovery, especially within bioinformatics. Our findings, detailed in Table 28 (Appendix I), showcase the performance of 12 causal discovery methods on this real-world dataset. DAGMA, a exemplified differentiable causal discovery method, outperforms others on the Sachs dataset. This outcome is significant as it empirically validates the effectiveness of differentiable causal discovery methods in real-world scenarios, which are often characterized by complexity. The consistent superiority of these methods across both synthetic and real-world datasets underscores their robustness amidst violations of model assumptions. Notably, Montagna et al. (2023) did not incorporate real-world data in their previous benchmarks. Our additional experiments bridge this gap and lead to more dependable conclusions.
>
> > **W2: Runtime of benchmark methods**
>
> **A:** Thanks for the suggestion, which improves the fair justification on the 12 benchmark methods. To address this concern, we have newly added the runtime results of benchmark methods across different model assumption violation scenarios, as detailed in Table 8 (Appendix D). The results indicate that differentiable causal discovery methods, specifically DAGMA, NOTEARS-MLP, and NoCurl, not only achieve superior performance but also with almost negligible runtime costs. As shown in the Table 8, DAGMA has a runtime of 2.41±0.32 seconds for 10 nodes and 3.19±0.48 seconds for 20 nodes. Similarly, NOTEARS-MLP has a runtime of 4.70±0.82 seconds for 10 nodes and 5.84±0.97 seconds for 20 nodes. NoCurl's runtime is slightly higher but still relatively low, at 5.68±0.49 seconds for 10 nodes and 10.29±1.84 seconds for 20 nodes.
>
> > **W3: Stability of benchmark methods**
>
> **A:** To highlight the stability of different methods, we have newly added standard deviation results. All results are now presented in the format of mean ± standard deviation over 10 trials, **as shown in Table 2.1~Table 27**. The results indicate that differentiable causal discovery demonstrates relatively stable performance under different model assumption violations.
>
> > **Q1: Discussion on scale invariance**
>
> **A:** Thanks for the comment. In the revised manuscript, we have discussed and cited the latest works (Deng et al., 2024) in **Summary and Implications for Practice (Section 4.2, lines 514-519), Introduction (line 60) and Related Work (lines 1239-1240)**. The recent work by Deng et al. (2024) shows that for linear differentiable methods, scale invariance can be achieved by appropriately choosing the loss function. This further reinforces our conclusion regarding the robustness of differentiable methods. In our benchmarks, the results in Table 4.2, Table 20 and Table 24 indicate that the performance of nonlinear differentiable methods under scale variation remains challenging and warrants further investigation.
>
>
>
> **Overall, we hope our responses address your concerns, and sincerely appreciate higher positive evaluation scores.**

---

> > ### Comment · Reviewer_gPsY · 2024-11-25
> > **Thanks**
> >
> > I appreciate the authors' thorough and detailed rebuttals. I feel that most of my concerns have been adequately addressed. To reflect my positive impression of the response, I have decided to increase my score.

---

> > > ### Author Response · Authors · 2024-11-26
> > > **Response to Reviewer gPsY**
> > >
> > > Many thanks for the kind response and the acknowledgement of our responses.

---

### Official Review · Reviewer_VU3C · 2024-11-04

**Soundness:** 3
**Presentation:** 3
**Contribution:** 2
**Rating:** 6
**Confidence:** 5

**Summary:**

Causal discovery algorithms frequently depend on unverifiable causal assumptions, which are often challenging to fulfill in real-world data. This study conducts an extensive empirical evaluation of the performance of various mainstream causal discovery algorithms across eight types of model assumption violations. The work finds that differentiable causal discovery methods demonstrate unexpected robustness in challenging conditions. Furthermore, they offer theoretical insights into the observed performance of differentiable causal discovery methods.

**Strengths:**

1. This paper addresses one of the most critical challenges in causal discovery: the applicability of causal discovery methods to real-world data. Overall, the paper is well-structured and clearly written.

2. The experiments are extensive, covering eight types of model assumption violations, twelve causal discovery methods, and varying graph sizes.

**Weaknesses:**

1. In the synthetic experiments, each type of model assumption violation is considered in isolation. However, in real-world datasets, multiple issues may arise simultaneously. How do the causal discovery methods perform when these issues are combined, e.g., confounding effects + measurement errors + heterogeneity?

2. Additionally, if you were to recommend the single most reliable causal discovery method for real-world datasets, what would it be?

3. Although this paper aims to address real-world problems, no real-world datasets can be found in the study.

**Questions:**

(See weaknesses above)

---

> ### Author Response · Authors · 2024-11-21
> **Response to Reviewer VU3C**
>
> > **W1: Add experiments on scenarios with combined model assumption violation**
>
> **A:** We have conducted new experiments under combined misspecified scenarios where confounding effects, measurement errors and heterogeneity are present simultaneously. The new experimental results in Table 22 and Table 23 (Appendix F) show that under combined misspecified scenarios, the performance of various methods is worse compared to single misspecified scenario. However, differentiable causal discovery still achieves optimal or competitive performance. Experiments involving other scenarios with combined model assumption violations are currently under active development, and the results will be included in the updated version of the manuscript within 48 hours.
>
> > **W2: Algorithm recommendation**
>
> **A:** We have added summarized results of the most competitive methods under misspecified scenarios in Table 26~Table 28. The experimental findings indicate that DAGMA exhibits optimal performance in linear settings. When the vanilla model is nonlinear Gaussian process, CAM achieves the best performance, and DAGMA performs competitively. However, the experiments in Section 4.1.1 show that NOTEARS-MLP outperforms CAM in broader nonlinear scenarios. Considering that DAGMA performs better than NOTEARS-MLP in vanilla scenarios, DAGMA is more reliable under misspecified scenarios. The newly added experiments on the real-world Sachs dataset, presented in Table 29 (Appendix I), also demonstrate the superiority of DAGMA. Since the misspecified scenarios considered here are commonly encountered in real-world data, the present experimental results suggest that DAGMA is the most reliable method for applications on real-world datasets.
>
> > **W3: Results on real-world datasets**
>
> **A:** We have supplemented experiments on the **real-world Sachs dataset**, a well-established benchmark in the research field of causal discovery, particularly within the bioinformatics community. The results, as presented in Table 29 (Appendix I), demonstrate the performance of 12 benchmark methods on this real-world dataset. The results in Table 29 indicate that differentiable causal discovery methods achieve the best performance on the Sachs dataset. This is significant as it provides empirical evidence of the effectiveness of these methods in a context that closely mirrors real-world complexities. The consistent performance of differentiable causal discovery methods across both synthetic and real-world datasets further substantiates their robustness under model assumption violations. It is worth noting that the previous benchmark by Montagna et al. (2023) did not include real-world data. Our newly added experiments fill in the blank and yield more reliable conclusions.
>
>
>
> **Overall, we hope our responses address your concerns, and sincerely appreciate higher positive evaluation scores.**

---

> ### Author Response · Authors · 2024-11-22
> **New response to Weakness 1**
>
> > **W1: New experiments on scenarios with combined model assumption violation**
>
> **A:** We have added new experiments under two (confounding, heterogeneity) and four (confounding, measurement error, heterogeneity, and autoregression) combined misspecified scenarios. The new experimental results shown in **Tables 22** and **Table 23** (Appendix F) indicate that as the number of combined misspecified scenarios increases, the performance of all methods declines. However, differentiable causal discovery still demonstrates optimal or competitive performance.

---

> ### Author Response · Authors · 2024-11-28
> **Response to Reviewer VU3C**
>
> We thank the reviewer for the time in reading and reviewing our manuscript and for their valuable comments, which are helpful for us to further improve our paper.
>
> We have made every effort to address all opportunities for improvement and questions by providing detailed clarifications and requested results.
>
> **Overall, thank you for your valuable comments. We hope the detailed clarifications provided above adequately address your concerns. If you have any further questions, please feel free to reach out, and we would be glad to provide additional explanations. We kindly invite you to reconsider your evaluation in light of these responses.**

---

> ### Comment · Reviewer_VU3C · 2024-12-02
>
> I appreciated the responses from authors, which addressed my concerns. The experimental results are really impressive, together with some theoretical analysis in particular cases. Therefore, I would like to maintain my positive score with higher confidence.

---

> ### Author Response · Authors · 2024-12-02
> **Response to Reviewer VU3C**
>
> Thank you for all these insightful comments and the kind feedback.

---

### Official Review · Reviewer_KWup · 2024-11-07

**Soundness:** 3
**Presentation:** 2
**Contribution:** 3
**Rating:** 5
**Confidence:** 4

**Summary:**

This work aims to empirically study the behavior of causal discovery algorithms, with a particular focus on optimization based methods. The authors approach this problem by generating a set of graph structures using erdos renyi and scale free graph generation algorithms, and then considering a set of scenarios of misspecification with respect to the generated data. These results are compared across a large representative set of causal discovery algorithms. Results are mixed overall, but generally show preferable performance for optimization and additive noise models.

**Strengths:**

This paper is addressing a very important, and generally understudied, problem within causal discovery. Because causal discovery relies on a set of untestable assumptions in order to provide the accompanying theoretical guarantees on structure recovery (and as a result entailed causal estimands), it's critical to have an understanding of the relative merit of these algorithms under violations of assumptions. The authors do a nice job of compiling a large set of candidate algorithms, and considering multiple misspecification scenarios. The paper also presents a substantial number of empirical results both in the main text and the supplement. While the paper contains little theoretical content, its direction represents work that is valuable to the community, allowing practitioners to reason over the behavior of current state of the art outside of the context of work that is advocating for a specific approach.

**Weaknesses:**

* Erdos Renyi and scale free networks are both fairly unrepresentative of the structures encountered in real world settings. It would have been preferable to either have a much wider range of graph generation algorithms, or used a semi-synthetic approach where the structure of known, real world networks, are used and synthetic data drawn using those implied conditional independencies to augment the current results.
* Overall, I would have liked to have seen more discussions on the implications of the findings of the experiments in practice. The authors do a commendable job of trying to assess a number of scenarios, but the accompanying discussion is limited and there, to my reading, isn't a discussion summarizing the findings or broader implications for practice. Including discussion and analysis along these lines would substantially improve the value to the community.
* It isn't clear how significance is being computed here. The simulations appear to only use 10 iterations per setting. There also aren't any uncertainty bounds given. See below for more questions on this front.

**Questions:**

Following up from above:
* Are the presented results the mean over 10 trials?
* Is significance being computed? If so how?
* It seems strange that many algorithms seem to have the same value for SHD and SID in the latent confounder setting. Can the authors provide some intuition for this?
* The degree of these graphs is fairly limited. Did the authors try denser settings? If so can these results be described as well?

---

> ### Author Response · Authors · 2024-11-21
> **Response to Reviewer KWup**
>
> > **W1: Add experiments on wider range of graphs**
>
> **A:** In the new manuscript, additional experiments on **Gaussian Random Partition (GRP) graphs** and the **real-world Sachs dataset** are introduced. The results for GRP graphs are shown in Table 20 and Table 21 (Appendix E), and the results for Sachs are shown in Table 29 (Appendix I). The core conclusions remain consistent: differentiable causal discovery methods still demonstrate robust performance. For the semi-synthetic dataset, we will consider it as future work in the conclusion (lines 536-538). We note that most differentiable causal discovery studies (NOTEARS (Zheng et al., 2018), GOLEM (Ng et al., 2020), NOTEARS-MLP (Zheng et al., 2020), DAGMA (Bello et al., 2022), etc.) have also not considered semi-synthetic dataset, so we believe the current experiments are reliable.
>
> > **W2: Summary and discuss implications for practice**
>
> **A:** In the new manuscript, we summarize the findings in Section 4.2 and discuss their practical implications. We have added summarized results on misspecified scenarios (Table 26~Table 28), real-world data results (Table 29), semi-synthetic data results (Table 31 and 32) and runtime comparisons of the benchmark methods (Table 8). The experimental findings indicate that differentiable causal discovery methods, exemplified by DAGMA, have the potential to achieve optimal or competitive performance on real-world data with negligible computational overhead.
>
> > **W3/Q1/Q2: Add standard deviation to show significance**
>
> **A:** In the previous version, we only reported the mean of the metrics. Now we have provided both the mean and the standard deviation of the metrics over 10 trials in Table 2.1~Table 28. The inclusion of standard deviations enhancing the understanding of the robustness of our results.
>
> > **Q3: Check SHD and SID in latent confounder setting**
>
> **A:** We have carefully checked the data generation process under latent confounders, and re-conducted the experiments accordingly. In fact, bugs have been fixed in the open-source code of the previous work (Montagna et al., 2023) by us in the new version. The updated results, presented in Tables 2.1~Table 28, further reinforce the conclusions on the robustness of differentiable methods.
>
> > **Q4: Denser graphs**
>
> **A:** Most studies in the literature, including NOTEARS (Zheng et al., 2018), GOLEM (Ng et al., 2020), NOTEARS-MLP (Zheng et al., 2020), GraN-DAG (Lachapelle et al., 2019), and DAGMA (Bello et al., 2022), typically evaluate graphs with a maximum average node degree of 4. Our work reports results for graphs with average degree of 2, and the Appendix E presents results for average degree of 4. To further test our findings, we conducted additional experiments on ER graphs with an average degree of 6. As shown in Table 18 and 19 (Appendix E), the results consistently demonstrate the robustness of differentiable causal discovery, thereby facilitating our main conclusions.
>
>
>
> **Overall, we hope our responses address your concerns, and sincerely appreciate higher positive evaluation scores.**

---

> > ### Author Response · Authors · 2024-11-26
> > **Response to Reviewer KWup**
> >
> > > **W1: New experiments on semi-synthetic data**
> >
> > **A:** In the new manuscript, we have added new experiments on **semi-synthetic data**. The semi-synthetic data is generated based on the network structure of the real-world Sachs dataset, using both linear and nonlinear vanilla models to create eight datasets with model assumption violations. The results in Tables 31 and 32 (Appendix L) show that differentiable causal discovery methods achieve optimal or competitive performance in scenarios other than scale variation.

---

> ### Author Response · Authors · 2024-11-28
> **Response to Reviewer KWup**
>
> We thank the reviewer for the time in reading and reviewing our manuscript and for their valuable comments, which are helpful for us to further improve our paper.
>
> We have made every effort to address all opportunities for improvement and questions by providing detailed clarifications and requested results.
>
> **Overall, thank you for your valuable comments. We hope the detailed clarifications provided above adequately address your concerns. If you have any further questions, please feel free to reach out, and we would be glad to provide additional explanations. We kindly invite you to reconsider your evaluation in light of these responses.**

---

### Meta-Review · Area_Chair_NynF · 2024-12-23

**Metareview:**

This work executes a comprehensive evaluation of differentiable causal discovery methods. Given the popularity of these methods, this is a welcome addition to the literature. A few minor concerns from the reviewers have been addressed in a revision, and although a consensus was not reached during the discussion, I found the authors' response and revision convincing. I recommend the paper for acceptance as a poster.

**Additional Comments On Reviewer Discussion:**

One reviewer expressed concerns about the experimental setting, however, the authors pointed out that recent work by Ng et al and Deng et al has addressed these concerns both theoretically and empirically. Two of the other reviewers expressed satisfaction with the response and recommended acceptance.

---

### Decision · Program_Chairs · 2025-01-22

Accept (Poster)